# Spatially targeted chemokine exocytosis guides transmigration at lymphatic endothelial multicellular junctions

Inam Liaqat [iD][1], Ida Hilska [iD][1], Maria Saario[1], Emma Jakobsson[1], Marko Crivaro[2], Johan Peränen [iD][3] & Kari Vaahtomeri [iD][1,4 ✉]

## Abstract

**Migrating cells preferentially breach and integrate epithelial and endothelial monolayers at multicellular vertices. These sites are amenable to forces produced by the migrating cell and subsequent opening of the junctions. However, the cues that guide migrating cells to these entry portals, and eventually drive the transmigration process, are poorly understood. Here, we show that lymphatic endothelium multicellular junctions are the preferred sites of dendritic cell transmigration in both primary cell co-cultures and in mouse dermal explants. Dendritic cell guidance to multicellular junctions was dependent on the dendritic cell receptor CCR7, whose ligand, lymphatic endothelial chemokine CCL21, was exocytosed at multicellular junctions. Characterization of lymphatic endothelial secretory routes indicated Golgi-derived RAB6+ vesicles and RAB3+/27+ dense core secretory granules as intracellular CCL21 storage vesicles. Of these, RAB6+ vesicles trafficked CCL21 to the multicellular junctions, which were enriched with RAB6 docking factor ELKS (ERC1). Importantly, inhibition of RAB6 vesicle exocytosis attenuated dendritic cell transmigration. These data exemplify how spatially-restricted exocytosis of guidance cues helps to determine where dendritic cells transmigrate.**

**Keywords** Chemokine CCL21; Targeted Exocytosis; Lymphatic Endothelium; Multicellular Junctions; Transmigration
**Subject Categories** Immunology; Membranes & Trafficking; Vascular Biology & Angiogenesis

## Introduction

Endothelium and epithelium form barriers that separate compartments. These barriers control fluid balance and pathogen invasion but, at the same time, allow the transmigration of leukocytes. Earlier studies have shown that the transmigration events are not randomly distributed on all blood endothelial junctions, rather there are preferred sites, coined as transmigration hot spots. The limited number of transmigration sites in the barrier is presumed to allow monolayer integrity and, thus, minimize the non-desired leakage and pathogen invasion.

The paracellular leukocyte transmigration across blood endothelium can be simplified as a sequence of events: (1) Attractants, such as chemokines, trigger the adhesion of leukocytes on endothelium, (2) upon adhesion, leukocytes migrate on the plane of the endothelium (also referred as "crawling"), and (3) opening of the junction allows the transmigration across the endothelium (Muller, 2016; Vestweber, 2015). However, how leukocytes identify the transmigration sites, is poorly understood. At least, three mutually non-exclusive possibilities have been reported: (i) leukocytes use protrusions to "palpate" the endothelium (Carman et al, 2007; Martinelli et al, 2014; Shulman et al, 2011; Vaahtomeri et al, 2017), possibly, for the identification of the path of least resistance (Martinelli et al, 2014). Accordingly, blood endothelial tricellular junctions, which have discontinuities in adherence junction proteins (Burns et al, 1997), have been identified as the preferred sites of transmigration (Burns et al, 1997; Gorina et al, 2014; Wang et al, 2006). (ii) Endothelial adhesion protein ICAM1 and membrane protrusions are enriched at the site of transmigration (Arts et al, 2021; Gronloh et al, 2023; Sumagin and Sarelius, 2010). It is conceivable that these present a landmark, resulting in the arrest of leukocytes at the transmigration permissive site. (iii) Leukocytes deposit chemokines and membrane fractions and, also, modify the basement membrane at the transmigration site (Girbl et al, 2018; Mydel et al, 2008; Wang et al, 2006). These traces may represent a memory of a successful entry point and, thus, guide the transmigration of subsequent neutrophils (Lim et al, 2015; Mydel et al, 2008; Wang et al, 2006). Altogether, the current evidence supports a model where the migrating leukocytes identify the transmigration sites based on their unique features. However, it is less clear what guides the leukocytes to the transmigration site and drives the transmigration.

Also, lymphatic endothelium presents a transmigration barrier for a variety of leukocytes, such as antigen-presenting dendritic cells (DCs), neutrophils, T cells, monocytes, and macrophages. The

[1]Translational Cancer Medicine Research Program, University of Helsinki, Biomedicum Helsinki, Haartmaninkatu 8, 00290 Helsinki, Finland. [2]Light Microscopy Unit, Institute of Biotechnology, HiLIFE, University of Helsinki, FI-00014 Helsinki, Finland. [3]Institute of Biotechnology, HiLIFE, University of Helsinki, FI-00014 Helsinki, Finland. [4]Wihuri Research Institute, Biomedicum Helsinki, Haartmaninkatu 8, 00290 Helsinki, Finland. ✉E-mail: kari.vaahtomeri@helsinki.fi

best-characterized leukocyte sub-type in this context are DCs, which breach the lymphatic endothelium of afferent lymphatic vessels on their way into the lymph nodes, where DCs present the antigens and, thus, activate the T cells (Arasa et al, 2021a; Jackson, 2019). In a context-dependent manner, DCs transmigrate across the lymphatic endothelial junctions, which display either discontinuous button-like or continuous zipper-like conformation of the cell junctions (Arasa et al, 2021b; Baluk et al, 2007; Pflicke and Sixt, 2009; Yao et al, 2012), the latter resembling blood endothelial junctions. The entry of DCs into the lymphatic vessels in vivo is dependent on the DC receptor CCR7 and its corresponding ligand chemokine CCL21 (Ohl et al, 2004; Tal et al, 2011). CCL21 is expressed by lymphatic endothelial cells (LEC), resulting in high concentration CCL21 deposits at the lymphatic vessel basement membrane and a decaying interstitial gradient, which guides the DC approach from the interstitium to the vicinity of lymphatic capillaries (Tal et al, 2011; Vaahtomeri et al, 2017; Vaahtomeri et al, 2021; Weber et al, 2013). However, it is not known whether CCL21 contributes to further guiding the DCs at the plane of the lymphatic endothelium, thus, enabling the identification of the transmigration sites.

Here, we ask the question, what determines the site of DC transmigration across the lymphatic endothelium. In these studies, we have used primary cell culture and explant models of transmigration (Pflicke and Sixt, 2009; Vaahtomeri et al, 2017). Our results show that lymphatic endothelial chemokine CCL21 exocytosis at the multicellular junctions drives the transmigration. These results suggest a novel cell biology concept of spatially targeted guidance cue exocytosis, as one of the key determinants of transmigration site.

## Results

### Chemokine CCL21 promotes dendritic cell transmigration at multicellular junctions

To investigate the preferred locations of DC transmigration across the lymphatic endothelium, we used our primary cell culture model, where lymphatic endothelial monolayer displays continuous zipper-like junctions similar to downstream lymphatic capillaries and collectors in vivo (Figs. 1A and EV1A), and lymphatic capillary tips upon inflammatory response (Yao et al, 2012). In this model, lymphatic endothelial expression of CCL21-mCherry drives the DC transmigration (Vaahtomeri et al, 2017). Here, we stained lymphatic endothelial cell junctions with fluorophore-coupled non-blocking VE-cadherin antibody and loaded activated DCs on the monolayer. At first, the DCs displayed a lamellar leading edge and migrated on the monolayer, followed by a 3–15-min arrest at the transmigration site, re-polarization towards the monolayer, and transmigration (Fig. 1A and Movie EV1). A great majority of the transmigration events were observed at multicellular junctions (84%) and only a minority of DCs traversed across the bicellular junctions (16%; Fig. 1A,B and Movie EV1).

To determine whether DCs transmigrate lymphatic endothelial multicellular junctions also in the tissue context, we used well-established mouse ear pinna dermal explants, where LPS-activated DCs approach and enter lymphatic capillaries and collectors (Arasa et al, 2021b; Pflicke and Sixt, 2009; Russo et al, 2016; Vaahtomeri

et al, 2017; Weber et al, 2013). Here, we stained lymphatic endothelial junctions with α-CD31-FITC and labeled DCs to visualize the transmigration process. For imaging, we selected pre-collector-like segments, which resided on the surface layers of the exposed dermis. In these segments, LEC morphology varied from elongated LECs displaying straight junctions to lymphatic capillary-like LEC morphology, displaying complex LEC shapes. All of the LECs of the imaged vessel segments had continuous CD31 junctions. In line with Arasa et al (Arasa et al, 2021b), LPS-activated DCs transmigrated the lymphatic pre-collectors upon 1–3-h interstitial migration and, subsequent, dwelling on the pre-collector vessel segments. Similar to the primary cell culture, a large percentage of DCs (48.5%) transmigrated the lymphatic endothelium at the multicellular junctions. 24.2% of the transmigration occurred near the multicellular junctions, i.e., <5 μm away, and 27.3% occurred at a 5–18 μm distance from the closest multicellular junction (Fig. 1C,D; Appendix Fig. S1A–C, Movies EV2–5). Altogether, these results show that the lymphatic endothelial multicellular junctions present the preferred site of transmigration, similar to blood endothelium (Burns et al, 1997; Gorina et al, 2014; Wang et al, 2006).

Approach and entry of DCs to lymphatic vessels are dependent on lymphatic endothelial chemokine CCL21 in vivo and in dermal explants (Ohl et al, 2004; Tal et al, 2011; Weber et al, 2013). Peri-lymphatic vessel CCL21 enables the DC arrest at the lymphatic vessel endothelium in a CCR7-dependent manner (Tal et al, 2011), and CCL21-CCR7 signaling is required for DC transmigration across lymphatic endothelium in vivo, in dermal explants, and in primary cell culture (Johnson and Jackson, 2010; Pflicke and Sixt, 2009; Tal et al, 2011; Vaahtomeri et al, 2017). Nonetheless, it is not known whether CCL21 contributes to the identification of the transmigration site. To address this with our primary cell culture model, we traced the wild-type and CCR7$^{-/-}$ DC migration paths and the sites of arrest on the lymphatic endothelial monolayer prior to transmigration. The wild-type DCs showed more straight trajectories in comparison to CCR7$^{-/-}$ DCs (Fig. 1E,F and Movie EV6). Moreover, wild-type DCs were more efficient than CCR7$^{-/-}$ DCs in arresting at the multicellular junctions (63% vs. 35%, Fig. 1G), and less prone to subsequent detachment (6% vs. 15%, Fig. EV1B and Movie EV6). Most of the arrested wild-type DCs transmigrated the LEC multicellular junction during the duration of the recorded movie. Comparatively, the arrested CCR7$^{-/-}$ DCs were unable to polarize towards the LEC monolayer and could not transmigrate. The defects of CCR7$^{-/-}$ DCs in reaching and arresting at multicellular junctions were primary to defective CCL21 sensing, since expression of DC markers CD86, CD11b, CD11c, and MHCII; contractile machinery; cell size; and cell morphology were similar in wild-type and CCR7$^{-/-}$ DCs (Fig. EV1E–L). Altogether, these results show that CCL21 promotes the identification of the multicellular junctions, i.e., the site of DC transmigration.

### CCL21 is exocytosed at the multicellular junctions

Since CCR7-CCL21 signaling promoted DCs arrest at multicellular junctions, we hypothesized that the CCL21 presentation was enriched at those sites. We envisioned two non-exclusive alternatives for localized CCL21 presentation: preferential (i) CCL21 exocytosis or (ii) anchoring at the multicellular junctions.

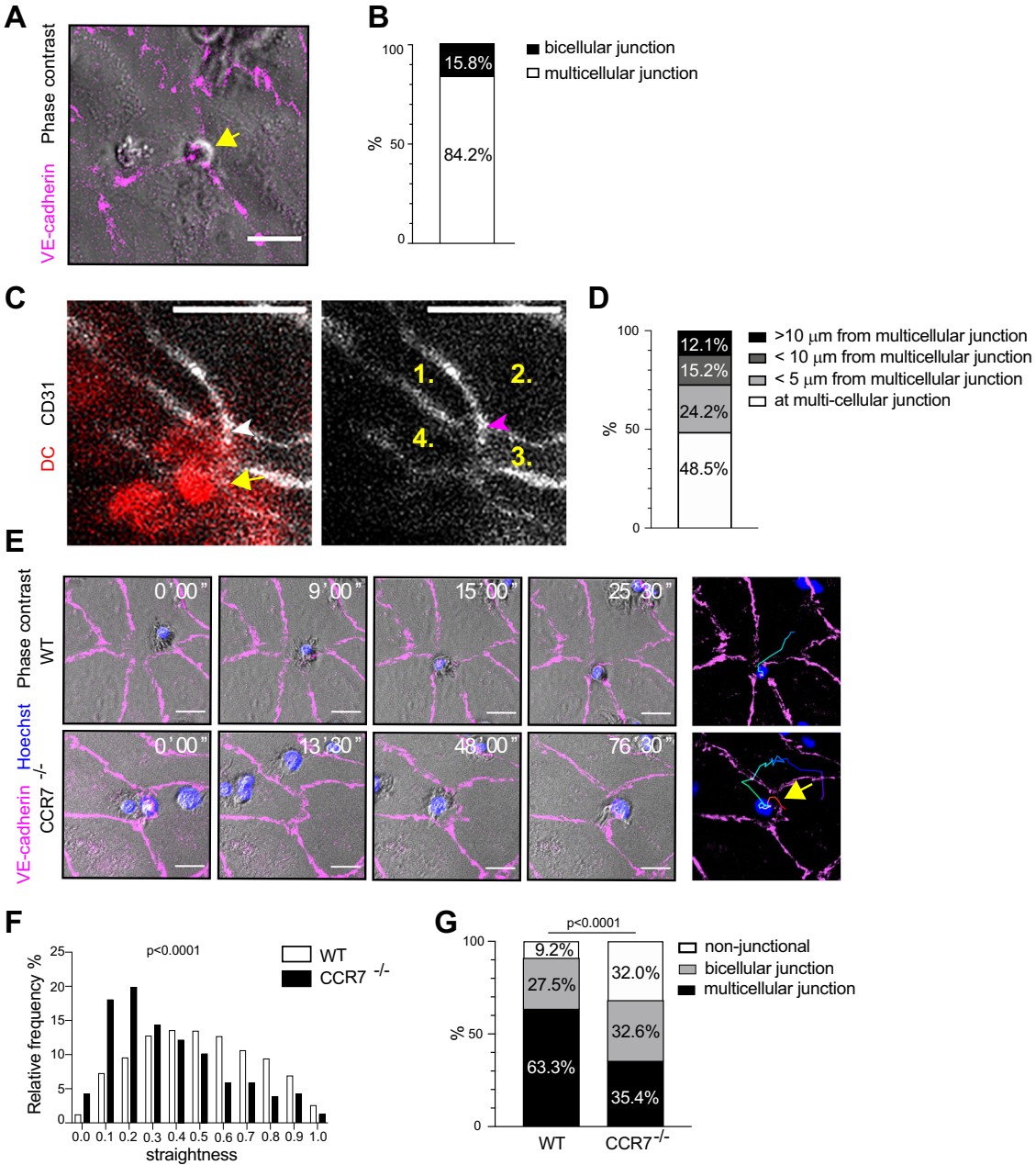

**Figure 1. CCL21 promotes DC transmigration across multicellular lymphatic endothelial junctions.**

(A, B) A capture of a phase contrast/immunofluorescence live recording (Movie EV1) shows a DC transmigrating the VE-cadherin stained (magenta) lymphatic endothelial multicellular junction. (B) Quantification of DC transmigration sites in CCL21-mCherry expressing LEC cultures. Transmigration sites are shown as a percentage of all events from 8 biological replicates in three independent experiments. $n = 361$ transmigration events. (C, D) A capture of a live recording of a mouse ear pinna dermal explant (Movies EV2–5) shows a DC (red), which is starting to transmigrate an α-CD31-FITC stained (gray) lymphatic endothelial multicellular junction. See more examples in Appendix Fig. S1A–C. In (D), the transmigration sites are shown as a percentage of all events ($n = 33$) from 3 independent experiments, representing, altogether, 6 mice. (E–G) Time-lapse series of a wild-type (WT) or CCR7[−/−] DC migration on the CCL21-mCherry expressing LEC cultures prior to transmigration. Stained VE-cadherin junctions are shown in magenta, and nuclei in blue (Hoechst). Also, time-color-coded track of the full DC migration path is shown. See also the corresponding Movie EV6 and Fig. EV1B. Quantification in (F) shows the straightness of wild-type and CCR7[−/−] DC tracks. The data represents 1403 wild-type and 543 CCR7[−/−] tracks. Quantification in (G) shows the sites of wild-type or CCR7[−/−] DCs arrest on CCL21-mCherry expressing LEC monolayers prior to transmigration. The data represents 633 wild-type and 396 CCR7[−/−] DC arrest events. In (F, G) data is derived from 6 wild-type and 5 CCR7[−/−] biological replicates and, altogether, three independent experiments. Data information: The yellow arrow in (A) indicates a transmigrating DC. In (C), yellow arrow indicates DC cell body, white arrowhead the DC leading edge, and magenta arrowhead the multicellular junction. In (E) the CCR7[−/−] DC detachment is indicated with a yellow arrow. The p-value in (F) and (G) were calculated using the Chi-squared test. Scale bars are 20 μm in (A), (C) and (E). Source data are available online for this figure.

To first address the contribution of the CCL21 anchoring, we expressed anchoring incapable CCL21ΔC-mCherry mutant (Fig. EV1C) (Vaahtomeri et al, 2017). The truncated CCL21 mutant supported DC transmigration at multicellular junctions in a comparable manner to full-length CCL21-mCherry (compare Fig. EV1D to Fig. 1B). Expression of endogenous CCL21 is downregulated upon cell culture (Wick et al, 2007), but still low levels are detected in primary LEC cultures (Johnson and Jackson, 2010). To exclude the contribution of the endogenous CCL21 in patterning of the multicellular junction, we used siCCL21, which efficiently silenced endogenous human CCL21, but did not affect the expression levels of overexpressed mouse CCL21ΔC-mCherry (Fig. EV1M,N). Upon silencing of endogenous hCCL21, CCL21ΔC-mCherry and full-length CCL21-mCherry promoted DC transmigration at the LEC multicellular junctions in a comparable manner, thus, indicating that CCL21 anchoring is not required for DC transmigration at LEC multicellular junctions (Fig. EV1O).

Thus, we turned our focus on CCL21 exocytoses. Earlier studies have indicated that CCL21 localizes to Golgi and storage vesicles in LECs (Johnson and Jackson, 2010; Vaahtomeri et al, 2017; Weber et al, 2013). Here, to study the spatial distribution and exocytosis of CCL21, we used the CCL21ΔC-mCherry construct, which allows a clear view of CCL21 storage vesicles, since it does not stick to the cell culture substrate (Vaahtomeri et al, 2017). Spinning disc confocal microscopy of fixed LEC monolayers showed high-intensity CCL21ΔC-mCherry vesicles at the perinuclear region, whereas lower-intensity CCL21ΔC-mCherry vesicles were observed in the vicinity of the LEC-LEC junctions, especially at the multicellular junctions (Fig. 2A). Next, we used live epifluorescence microscopy to investigate whether CCL21 is exocytosed at the LEC junctions. We observed exocytosis of CCL21ΔC-mCherry+ vesicles at the entire length of LEC junctions (Fig. 2B,C). However, exocytosis events were most abundant at 0–5 μm distance from the multicellular junction and decayed as a function of distance (Fig. 2C and Movie EV7).

Earlier we showed that DCs induce Ca-flux in LECs and, on the other hand, CCL21ΔC-mCherry exocytosis was, in part, calcium-sensitive (Vaahtomeri et al, 2017). To investigate whether CCL21 vesicles at multicellular junctions were sensitive to DC-LEC interactions we monitored CCL21ΔC-mCherry exocytosis events live at multicellular junctions in the presence or absence of an arrested DC. Recordings indicated that CCL21ΔC-mCherry exocytosis was not affected by DC-LEC contact (Movie EV8). To corroborate this data, we treated CCL21ΔC-mCherry expressing monolayers with calcium chelator BAPTA-AM, to blunt conditional secretion events. The calcium chelation did not have an effect on the number of CCL21ΔC-mCherry exocytosis events at the LEC junctions (Fig. EV1P). Although we cannot fully exclude conditional CCL21 exocytosis at some point of transmigration, these results, altogether, suggest that CCL21 exocytosis at primary LEC multicellular and bicellular junctions is constitutive.

## CCL21 containing RAB6+ vesicles localize to the multicellular junctions

Earlier reports have shown that in LECs CCL21 localizes to Golgi and exocytic vesicles, which are trafficked along the microtubules (Johnson and Jackson, 2010; Vaahtomeri et al, 2017; Weber et al, 2013). Here, we investigated the molecular mechanisms controlling

CCL21 delivery to multicellular junctions and, thus, conducted colocalization analyses with RAB-GTPases, which are specific markers of distinct sub-classes of intracellular endomembrane vesicles (Zhen and Stenmark, 2015). To this end, we selected a panel of RAB-GTPases known to decorate exocytic vesicles or endosomes. The selected RAB-GTPases were sub-cloned as EGFP-fusions to a lentiviral vector and the LECs were co-transduced with the EGFP-RAB-GTPase and CCL21ΔC-mCherry constructs. Semi-automated colocalization analyses of high-resolution images indicated a robust colocalization of CCL21ΔC-mCherry with EGFP-RAB6A (30% of all the CCL21ΔC-mCherry vesicles), especially at multicellular junctions, whereas RAB3D (57%), RAB27A (48%) and RAB37 (52%) colocalized with CCL21ΔC-mCherry in the perinuclear region (Fig. 2D–F). Other tested EGFP-RAB fusions showed either moderate (RAB1A 4.6%, RAB2A 6.3%, RAB5C 1.9%, RAB7A 11.4%, RAB8A 1.5%, RAB10 1.6%, RAB11A 2.5%) or no colocalization (RAB13, and RAB35) with the CCL21ΔC-mCherry (Fig. EV2A–G). In addition to vesicles, EGFP-RAB8A localized also to the plasma membrane and displayed high-intensity patches at the multicellular junctions (Fig. EV2H,I). To capture the EGFP-RAB8A colocalization with CCL21ΔC-mCherry in these high-intensity areas, we conducted manual analysis, which indicated that EGFP-RAB8A colocalized with 7% of CCL21ΔC-mCherry vesicles in the whole LEC area. The colocalization pattern of EGFP-RAB8A resembled EGFP-RAB6 (Fig. EV2J,K). Colocalization analysis with full-length CCL21-mCherry and EGFP-RAB3D, -RAB6A, -RAB10, -RAB13, -RAB27A, or -RAB37 recapitulated these results (Fig. EV3A–E).

To address, whether the identified vesicle populations represent more general LEC secretory routes, we analyzed the colocalization of the EGFP-RAB3D, -6A, and -27A with both endogenous chemokine CCL2, also known as monocyte chemoattractant protein 1 (MCP1), and a general secretory marker composed of a signaling peptide (SP) fused to mCherry. Here, the LECs were treated with TNF-α for 24 h to induce the CCL2 expression. Similar to CCL21ΔC-mCherry, both endogenous CCL2 and SP-mCherry showed robust colocalization with EGFP-RAB6A+ vesicles at the multicellular junctions (Fig. EV4A). In the perinuclear region, EGFP-RAB3D and EGFP-RAB27A colocalized with SP-mCherry but less with CCL2 and some of the CCL2 puncta did not colocalize with any of the tested EGFP-RAB-GTPases (Fig. EV4A). Considering similar EGFP-RAB signatures of CCL21ΔC-mCherry+, and CCL2+ vesicles, we addressed whether these two chemokines colocalized. Indeed, TNF-α treated LECs displayed a robust colocalization of CCL21ΔC-mCherry and endogenous CCL2 both at the LEC junctions and perinuclear area (Fig. 2G). Altogether, we have identified 4 RAB-GTPases (RAB6A, RAB3D, RAB27A, and RAB37) that mark LEC secretory vesicle populations.

Next, we evaluated the expression levels of the identified RAB-GTPases in dermal lymphatic vessel LECs in vivo. Here, we capitalized on a previously published single-cell sequencing dataset (Data ref: Gene Expression Omnibus, GSE201916) (Petkova et al, 2023). Amongst the chemokine colocalizing RAB-GTPases, RAB6A, RAB3D, and RAB27A were expressed in all LEC sub-types, including capillary and collector LECs, whereas RAB37 was not expressed in dermal LECs in vivo (Fig. EV4B).

These results encouraged us to analyze the colocalization of endogenous RAB6 with the LEC secretory cargo. Similar to EGFP-RAB6, endogenous RAB6 showed near complete colocalization with both CCL21ΔC-mCherry and endogenous CCL2 at the

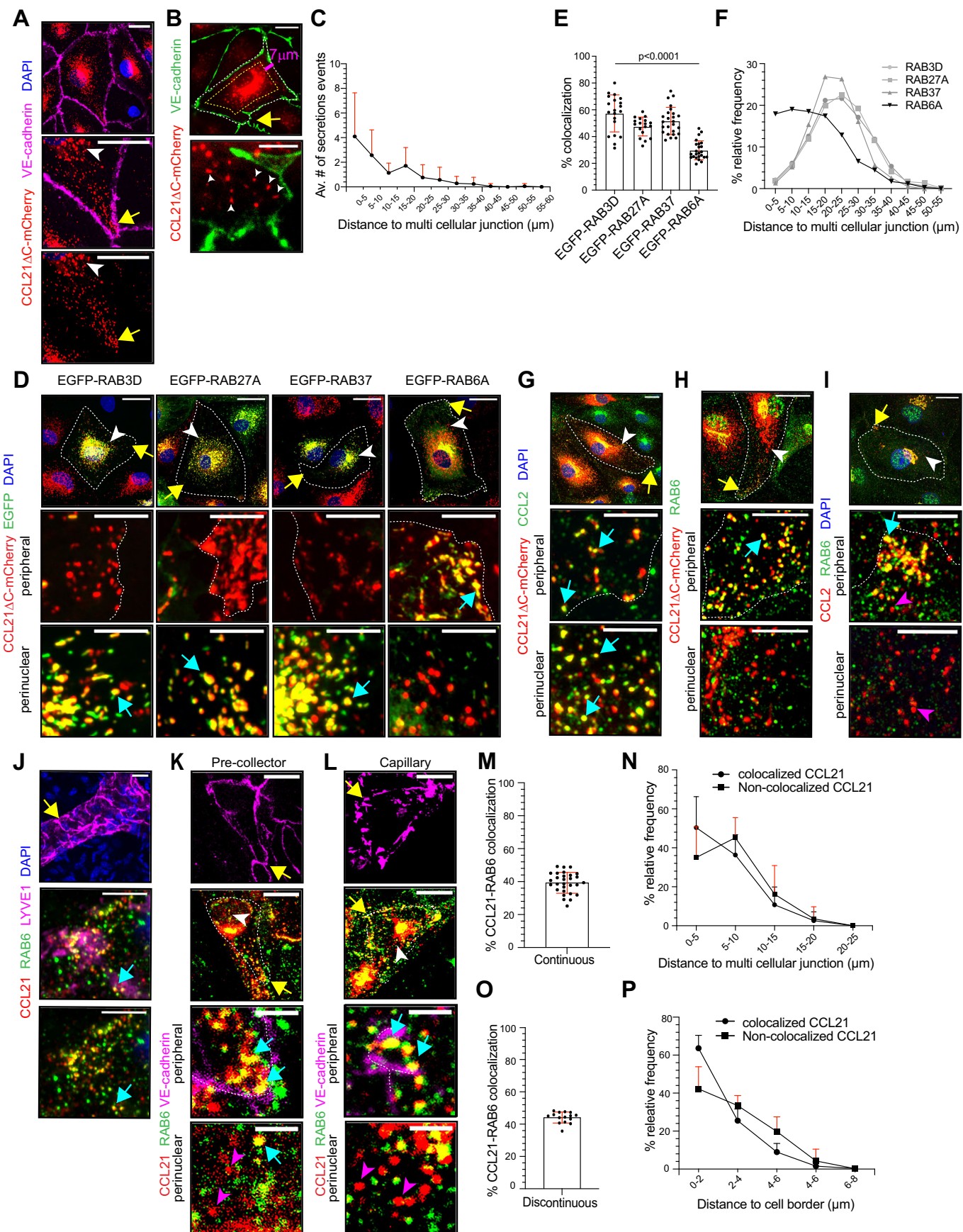

◄

**Figure 2.   RAB-GTPase identity of chemokine CCL21 containing vesicles in LECs.**

Figure 2 is shown with alternative colors (non-red and -green) in Appendix Fig. S5. (A) LEC monolayer expressing CCL21ΔC-mCherry (red) and stained for VE-cadherin (magenta) and nuclei (DAPI, blue). The data in (A) represents at least $n = 3$ independent experiments. (B, C) A capture of immunofluorescence live recording (Movie EV7) shows primary LEC monolayer expressing CCL21ΔC-mCherry (red) and stained for VE-cadherin (green). Exocytosis events were analyzed within a 7 μm wide region. (C) A histogram showing exocytosis events (mean number of  secretions/cell + SD) at the LEC junctions, as a function of distance (in μm) from the nearest multicellular junction. The data in (B and C) represents 241 secretion events from $n = 22$ LECs in 5 biological replicates across two independent experiments. (D–F) LECs expressing chemokine CCL21ΔC-mCherry (red) and the indicated EGFP-tagged RAB-GTPase (green). The nuclei are stained with DAPI (blue). The images are representative of $n = 3$ biological replicates in three independent experiments. Quantification in (E) shows percentage of CCL21ΔC-mCherry+ vesicle colocalization with the indicated EGFP-RAB GTPases in the whole LEC area. The dot plot shows the mean percentage ± SD. Each data point represents a single analyzed cell (EGFP-RAB3D ($n = 20$); EGFP-RAB27A ($n = 18$); EGFP-RAB37 ($n = 25$) and EGFP-RAB6A ($n = 24$)), from a total of 3 biological replicates in three independent experiments. The histogram (F) shows the distribution (mean percentage) of the colocalized vesicles as a function of distance from a multicellular junction. The number of samples was the same as in (E). (G) A TNF-α-treated LEC, expressing chemokine CCL21ΔC-mCherry (red) and, stained for endogenous CCL2 (green), and nuclei (DAPI, blue). The images represent $n = 2$ independent experiments. (H) A LEC expressing chemokine CCL21ΔC-mCherry (red) and stained for endogenous RAB6 (green). The images represent $n = 2$ independent experiments. (I) A TNFα-treated LEC monolayer stained for endogenous CCL2 (red), RAB6 (green), and nuclei (DAPI, blue). The images represent $n = 2$ independent experiments. (J) Mouse-ear pinna dermis stained for CCL21 (red), RAB6 (green), LYVE1 (magenta), and nuclei (DAPI, blue). For clarity, the overview image shows only staining of LYVE1+ and nuclei. The images are representative of 3 mice. (K) Representative images show a mouse ear pinna dermis lymphatic pre-collector (continuous junctions, quantified in (M, N) and (L) capillary (discontinuous junctions, quantified in (O, P) stained for CCL21 (red), RAB6 (green), VE-cadherin (magenta). The overview images show VE-cadherin-only or CCL21 and RAB6. The zoom-in images show images of peripheral and perinuclear areas of the LEC. Figures 2K and L are shown with more examples in Appendix Fig. S2A–F. (M) Quantification of CCL21 vesicle colocalization (mean percentage ± SD) with RAB6, in LECs showing continuous junctions in mouse ear pinna dermis in vivo. Each data point represents a single analyzed cell. (N) The histogram shows the distribution (mean percentage + SD) of CCL21 and RAB6 colocalized vesicles and non-colocalized CCL21 vesicles, as a function of distance from the multicellular junction (for LECs displaying continuous junctions). In (K) and (M, N), $n = 29$ cells representing 6 mice. (O) Quantification of CCL21 vesicle colocalization (mean percentage ± SD) with RAB6, in LECs showing discontinuous junctions in mouse ear pinna dermis in vivo. Each data point represents a single analyzed cell. (P) The histogram shows the distribution (mean percentage + SD) of CCL21 and RAB6 colocalized vesicles and non-colocalized CCL21 vesicles, as a function of distance from the VE-cadherin stained cell border (for LECs displaying discontinuous junctions). In (L) and (O, P), $n = 15$ cells representing 3 mice. Data information: In (A), (D), (G–I), and (J–L), the yellow arrow indicates the peripheral, and the white arrowheads the perinuclear area shown in the zoom-in image. The cell borders in (D), (G–I), and (K, L) are marked with white dotted lines. The cyan arrow shows an example of colocalization and the magenta arrowheads examples of non-colocalizing vesicles. In (B), the yellow arrow indicates LEC multicellular junction (shown in the zoom-in image and in Movie EV7). The vesicles that were exocytosed are marked with white arrowheads. The *p*-value in (E) was calculated using one one-way ANOVA test. Scale bars in (A, B), (D), and (G–J) are 20 μm in the overview images and 5 μm in the zoom-in images. In (K, L) the scale bars in overview images are 10 μm and 2 μm in the zoom-in images. Source data are available online for this figure.

multicellular junctions and Golgi of cultured human primary LECs (Fig. 2H,I). The detected RAB6 staining was specific, as the siRAB6-mediated silencing resulted in a loss of RAB6 signal (Fig. EV5G). Next, we investigated CCL21-RAB6 colocalization in vivo in mouse ear pinna dermis. In pre-collector lymphatic vessels, where LECs display continuous VE-cadherin junctions and elongated cell morphology, 39% of CCL21 vesicles colocalized with RAB6 (Fig. 2J,K; Appendix Fig. S2A–D). In line with our primary cell culture data, RAB6 + CCL21 vesicles were most abundant in the vicinity of the multicellular junctions, whereas RAB6 negative CCL21 vesicles were more abundant further away from the multicellular junctions (Fig. 2N). To address whether CCL21-RAB6 colocalization also occurred in lymphatic capillary LECs, which have discontinuous VE-cadherin junctions, we analyzed the LECs in the tip segments of blind-ended lymphatic vessels. Also here, 44% of CCL21 vesicles colocalized with RAB6 (Fig. 2L,O; Appendix Fig. S2E,F), and most of these colocalizing vesicles were found at the LEC VE-cadherin junctions (Fig. 2L,P). To conclude, CCL21 and CCL2 chemokines are stored in RAB6 secretory vesicles at LEC junctions.

## CCL21 localizes to RAB6A+ Golgi-derived vesicles and RAB3D/27A+ dense core secretory granules

To corroborate the RAB-GTPase signature analyses, we investigated the identity of the CCL21 storage vesicles with transmission electron tomography. Upon, immunogold-labeling of mCherry we detected CCL21-mCherry and CCL21ΔC-mCherry in three types of vesicles: low-electron density vesicles, dense core granules, and multivesicular bodies (Fig. 3A,B, see Appendix Fig. S3A for a

staining control). Next, to identify the RAB-GTPase signature of each of the vesicle types, we overexpressed and immunogold-labeled EGFP-RAB6A as a marker of multicellular junction vesicles or EGFP-RAB27A as a marker of perinuclear vesicles. EGFP-RAB6A decorated the low-electron-density vesicles, whereas EGFP-RAB27A was exclusively specific for the dense core vesicles (Fig. 3C, see Appendix Fig. S3B for a staining control). Neither EGFP-RAB6A nor EGFP-RAB27A localized on multivesicular bodies (Fig. 3C). Multivesicular bodies are late endosomes that mature into lysosomes or fuse with the plasma membrane to release the exosomes (Huotari and Helenius, 2011). Accordingly, we observed some colocalization of CCL21ΔC-mCherry and EGFP-RAB7A, a marker of late endosomes, in our initial colocalization screen (Fig. EV2B,E). To confirm the late endosome localization, we analyzed CCL21ΔC-mCherry colocalization with two endogenous markers of late endosomes RAB7 and LAMP3 (CD63) and a marker of lysosomes LAMP1. All of these markers colocalized with CCL21ΔC-mCherry (5.6–7.5%, Fig. 3D–H), albeit to a much lower extent than the EGFP-RAB6A, RAB3D, RAB27A, and RAB37. These results indicate that some of the CCL21ΔC-mCherry is funneled to the lysosomal route.

Previously, RAB27A, RAB3D, and RAB37 have been indicated as markers of dense-core secretory granules in COS cells (Tsuboi and Fukuda, 2006). Considering that EGFP-RAB27A decorated dense core granules in LECs (Fig. 3C) and displayed similar perinuclear localization with EGFP-RAB3D and EGFP-RAB37 (Fig. 2D,F), we hypothesized that all of these three RAB-GTPases mark the same vesicle type. To test this, we tagged RAB27A with mCherry and co-expressed it with EGFP-RAB3D. Indeed, the mCherry-RAB27A and EGFP-RAB3D showed almost

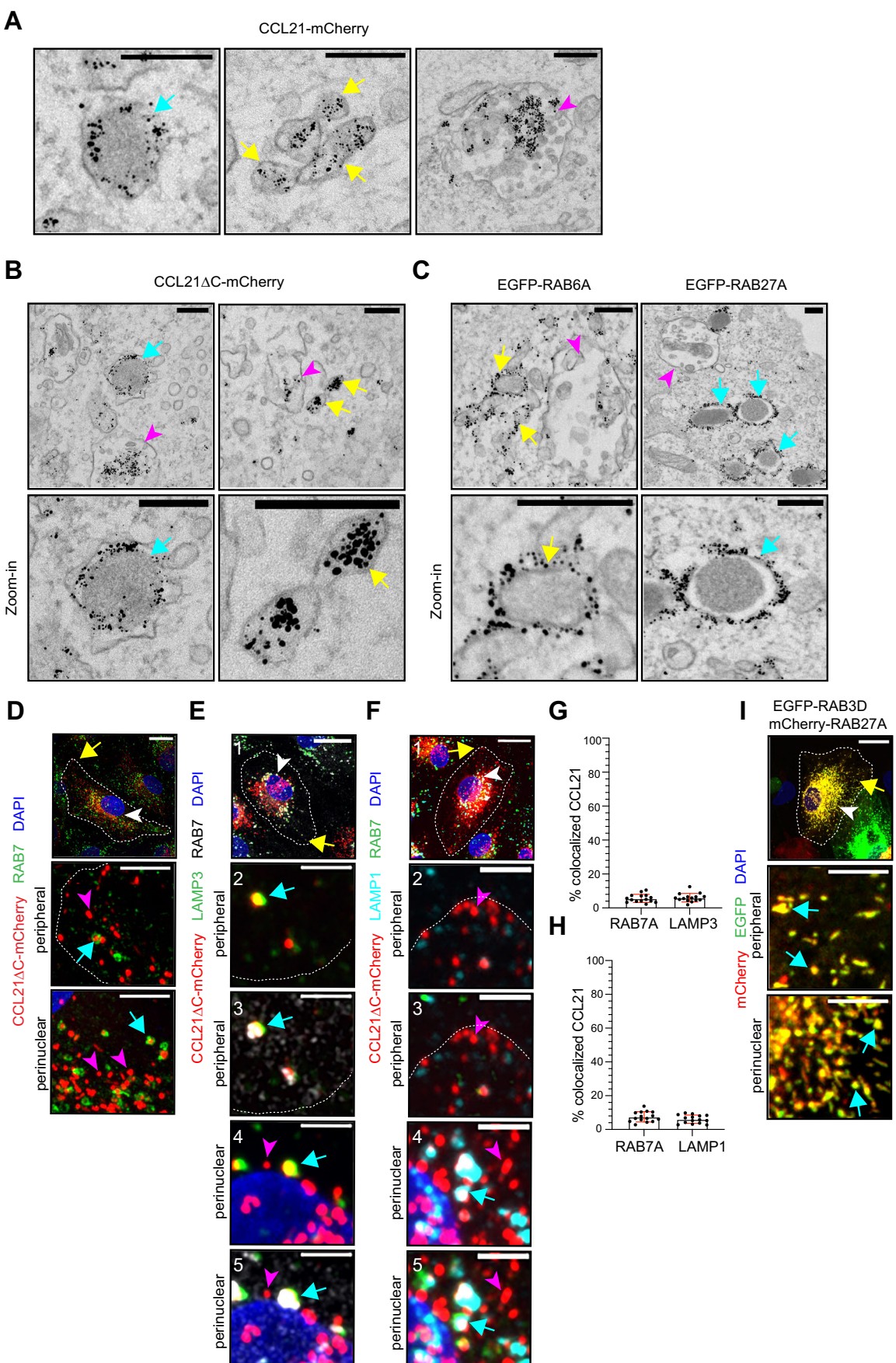

**Figure 3.  RAB27A-RAB3D vesicles represent dense-core secretory granules in LECs.**

Figure 3 is shown with alternative colors (non-red and -green) in Appendix Fig. S6. (A–C) Transmission electron micrograph of anti-RFP or anti-EGFP immunogold labeled (A) CCL21-mCherry, (B) CCL21ΔC-mCherry, (C) EGFP-RAB6A, or EGFP-RAB27A expressing LECs. The images represent $n = 2$ independent experiments. Electron micrographs with control labeling are shown in Appendix Fig. S3A,B. (D) Shows a LEC expressing CCL21ΔC-mCherry (red) and stained for endogenous RAB7 (green). The images represent $n = 2$ independent experiments. (E) LEC expressing CCL21ΔC-mCherry (red) and stained for endogenous LAMP3 (CD63; green) and RAB7 (gray). The nuclei are stained with DAPI (blue). The zoom-in images, show either CCL21ΔC-mCherry and LAMP3 channels (images 2 and 4) or RAB7 together with CCL21ΔC-mCherry, and LAMP3 channels (images 3 and 5). The images represent $n = 3$ independent experiments. Quantification shown in (G). (F) Shows LEC expressing CCL21ΔC-mCherry (red) and stained for endogenous LAMP1 (cyan), RAB7 (green), and nuclei (DAPI; blue). The zoom-in images, show either CCL21ΔC-mCherry and LAMP1 (images 2 and 4) or also with RAB7 (images 3 and 5). The images represent $n = 3$ independent experiments. Quantification is shown in (H). (G, H) Quantification of CCL21ΔC-mCherry vesicle colocalization with endogenous RAB7A or LAMP3 in (G) and RAB7A or LAMP1 in (H). The dot plots show the mean percentage ± SD. Each data point represents a single analyzed cell with $n = 15$. The data represents 5 biological replicates in three independent experiments. (I) Shows expression of EGFP-RAB3D (green) with mCherry-RAB27A (red) in TNF-α treated LECs. The images represent $n = 3$ independent experiments. Data information: In (A–C), cyan arrows indicate dense core granules, yellow arrows low-electron density vesicles, and magenta arrowheads multivesicular bodies. In (D–F) and (I), the cell borders are marked with a white dotted line. Yellow arrows and white arrowheads indicate the site of the peripheral and perinuclear areas, respectively, shown in the zoom-in images. Cyan arrows indicate examples of CCL21ΔC-mCherry+ colocalizing vesicles and magenta arrowheads non-colocalizing CCL21ΔC-mCherry+ vesicles. Scale bars are 200 nm in transmission electron micrographs in (A–C); 20 μm in overview image and 5 μm in zoom-in images in (D–F) and (I). Source data are available online for this figure.

complete colocalization (Fig. 3I). Altogether, our approach resulted in the identification of two major pools of CCL21 containing LEC secretory vesicle types: the RAB6 vesicles, which have been reported to represent direct Golgi-to-plasma membrane secretory vesicles (Fourriere et al, 2019; Grigoriev et al, 2007; Miserey-Lenkei et al, 2010), and RAB3/27+ vesicles, which represent dense core secretory granules (Tsuboi and Fukuda, 2006).

## Microtubules target LEC multicellular junctions

The multicellular junction localization of chemokine containing RAB6+ vesicles allowed us to investigate the molecular mechanisms of chemokine delivery and exocytosis at the site of transmigration. First, we used live microscopy to monitor the trafficking of the EGFP-RAB6 vesicles. The tracing of EGFP-RAB6 vesicles in LECs showed direct tracks between the perinuclear area and multicellular junctions (Fig. 4A), which is in line with previous reports, indicating RAB6+ vesicles in the direct trafficking of secretory cargo from Golgi to the site of exocytosis (Fourriere et al, 2019; Grigoriev et al, 2007). At multicellular junctions, EGFP-RAB6 vesicles dwelled for long periods of time and some of them were exocytosed (Fig. 4A and Movie EV9). The EGFP-RAB6 trajectories resembled microtubule tracks and, indeed, previous studies have indicated microtubules in both RAB6+ and CCL21ΔC-mCherry+ secretory vesicle trafficking (Fourriere et al, 2019; Grigoriev et al, 2007; Vaahtomeri et al, 2017). Accordingly, CCL21ΔC-mCherry+ RAB6+ double-positive vesicles were associated with microtubules at the multicellular junctions in LECs (Fig. 4B).

To dissect whether RAB6 trafficking to multicellular junctions is caused by the use of a selected subset of microtubules or more general targeting of multicellular junctions by microtubules, we stained the microtubules in LECs. Interestingly, most of the microtubules connected the perinuclear area and multicellular junctions (Fig. 4C). To investigate the causatives of such a pattern, we transduced LECs with a microtubule plus-end marker EB3-mCherry, which allows tracking of the growing microtubule. Live recording of EB3-mCherry+ microtubule +ends indicated that microtubules grew radially from the perinuclear area towards multi- or bicellular junctions (Movie EV10). Upon, reaching the bicellular junction, microtubule +ends continued to grow parallel to the junction until reaching the multicellular junction, where the microtubule growth was finally arrested (Fig. 4D,E and

Movie EV10). Altogether, these results indicate that most of the microtubule tracks terminate at LEC multicellular junctions. Thus, the RAB6-CCL21 vesicle trafficking to multicellular junctions could be simply explained by plus-end motor protein-mediated trafficking to the end of the microtubule tracks (Grigoriev et al, 2007).

## RAB6 docking factor ELKS determines the multicellular junctions as a hot spot of CCL21 exocytosis

Next, we investigated the molecular mechanisms of RAB6+ CCL21+ vesicle exocytosis at the LEC multicellular junctions. Previous studies have indicated ELKS as a critical docking factor of RAB6+ vesicles (Fourriere et al, 2019; Lansbergen et al, 2006), whereas in neurons ELKS captures and arrests RAB6 vesicles prior to the site of exocytosis (Nyitrai et al, 2020). Here, in LECs, ELKS localized at the immediate vicinity of the LEC junctions. Intriguingly, the ELKS levels were the highest at the multicellular junctions where ELKS formed high-intensity patches (Fig. 5A,B). The junctional ELKS intensity decayed as a function of distance from the multicellular junctions and reached a plateau at about 10 μm distance (Fig. 5B).

ELKS is recruited to the plasma membrane by LL5-β (Lansbergen et al, 2006), which directly interacts with Phosphatidylinositol (3,4,5)-trisphosphate (PIP3) by its C-terminal PH-domain (Lansbergen et al, 2006; Paranavitane et al, 2003). To investigate whether ELKS localization to multicellular junctions in LECs was dependent on phosphoinositide 3-kinase (PI3K) activity, we treated the LECs with PI3K-specific inhibitor PI-103 (Bain et al, 2007). In concentration-dependent manner, 1-h-long treatment resulted in a loss of high-intensity ELKS patches and flattening of the ELKS intensity across the entire length of LEC junctions (Fig. EV5A–C). Altogether, these results showed that RAB6 docking factor ELKS is enriched at LEC multicellular junctions in a PI3K activity-dependent manner.

To dissect whether ELKS is needed for the RAB6+ CCL21+ vesicle capture and/or exocytosis at the multicellular LEC junctions, we silenced ELKS with siRNAs. ELKS knockdown resulted in a significant reduction of ELKS mRNA and ELKS protein levels, as measured by quantitative PCR and immunofluorescence, respectively (Fig. EV5D–F). Live microscopy of siELKS-treated LECs showed almost a complete loss of CCL21ΔC-mCherry exocytosis at the LEC junctions (Fig. 5C,D). Lack of junctional exocytosis was accompanied by a marked accumulation of CCL21ΔC-mCherry

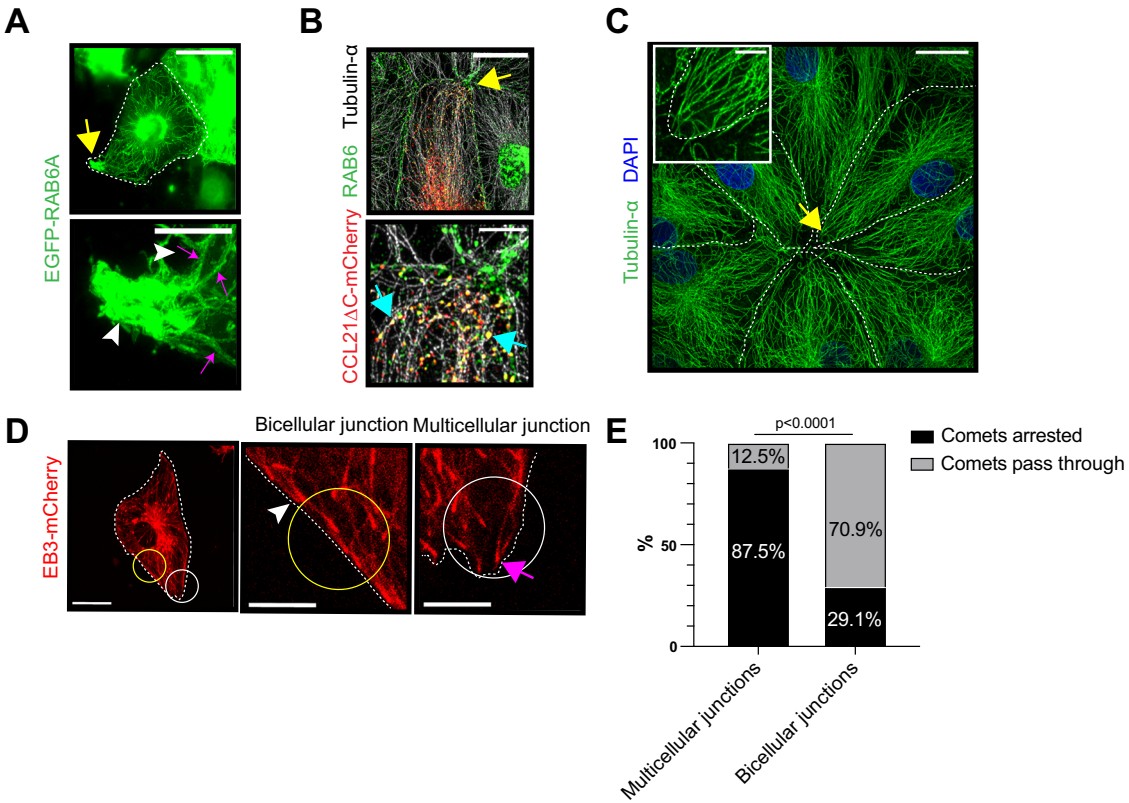

**Figure 4. RAB6 tracks and microtubules target the LEC multicellular junctions.**

Figure 4 is shown with alternative colors (non-red and -green) in Appendix Fig. S7. (A) Maximum projection (in time) from a live recording of EGFP-RAB6A (green) expressing LEC monolayer (Movie EV9). The images represent n = 3 independent experiments. (B) The image shows a LEC monolayer expressing CCL21ΔC-mCherry (red) and stained for endogenous RAB6 (green) and tubulin-α (gray). The images represent n = 2 independent experiments. (C) LEC monolayer stained for tubulin-α (green) and nuclei (DAPI, blue). The images represent n = 3 independent experiments. (D, E) A capture of live recording of LEC monolayer expressing EB3-mCherry (Movie EV10). Yellow and white circles indicate examples of selected areas on bicellular and multicellular junctions, respectively. These indicated areas were used for the quantification shown in (E). (E) A stacked bar graph shows mean percentage of EB3-mCherry comets that arrest or pass through the analyzed area. The data represents 21 cells (consisting of a total of 59 bicellular and 59 multicellular ROIs) from n = 6 biological replicates in three independent experiments. Data information: In (A), (C), and (D) white dotted line indicates LEC boundaries and in (A–C), the yellow arrow indicates the multicellular junction, which is shown in the zoom-in image. In (A), in the zoom-in images, magenta arrows indicate straight tracks of EGFP-RAB6A vesicles, whereas the white arrowheads indicate the dwelling of the EGFP-RAB6 vesicles at the multicellular junction of the LEC (see Movie EV9). In (B), cyan arrows indicate CCL21ΔC-mCherry+ RAB6 vesicles which are associated with microtubules. In (D), The white arrowhead indicates EB3-mCherry (red) positive comet, which will pass through the analyzed area (yellow circle). Whereas the magenta arrow indicates EB3-mCherry comet that arrests inside the white circle at the multicellular junction of the cell (see Movie EV10). In (E), the p-value was calculated using Fisher's exact test. The scale bar in (A) is 50 μm in overview images and 10 μm in zoom-in images and in (B–D), 20 μm in overview images and 5 μm in zoom-in images. Source data are available online for this figure.

vesicles at the multicellular junctions but only a moderate increase in whole LEC CCL21ΔC-mCherry intensity (Fig. 5E–H). Importantly, accumulated CCL21ΔC-mCherry vesicles, within 5 μm reach of multicellular junctions, were positive for endogenous RAB6, whereas RAB6 + CCL21ΔC-mCherry+ double-positive vesicles did not accumulate in the rest of the LEC body (Fig. 5I,J). These results indicate that ELKS-mediated CCL21 exocytosis at multicellular junctions is largely restricted to RAB6 vesicles.

## RAB6+ vesicle-mediated CCL21 exocytosis controls DC transmigration at multicellular junctions

To further investigate the RAB6 dependency of the CCL21 exocytosis at the multicellular junctions, we silenced RAB6 in CCL21ΔC-mCherry expressing monolayers. Analysis of live-microscopy recordings indicated almost a complete loss of

exocytosis events at the LEC multicellular junctions (Fig. 6A,B). In an analogy to siELKS, the siRAB6 treatment also resulted in the accumulation of CCL21ΔC-mCherry vesicles at the multicellular junctions and a more moderate increase in the CCL21ΔC-mCherry intensity in the whole LEC (Fig. 6C–F).

Next, we analyzed endogenous CCL2 levels in ELKS or RAB6 silenced LECs upon TNF-α treatment. Similarly, to CCL21ΔC-mCherry, endogenous CCL2 accumulated at multicellular junctions upon ELKS knockdown, whereas the total CCL2 levels displayed only a moderate statistically non-significant increase (Fig. 6G–J). In contrast, RAB6A silencing did not result in the accumulation of CCL2 at multicellular junctions but, rather, in a modest increase in CCL2 intensity in the whole LEC area (Fig. 6G–J). The difference in siELKS and siRAB6 effect, possibly, reflects the role of RAB6 in kinesin-mediated vesicle trafficking of Golgi-derived vesicles to microtubule +ends (Grigoriev et al, 2007). Thus, chemokine

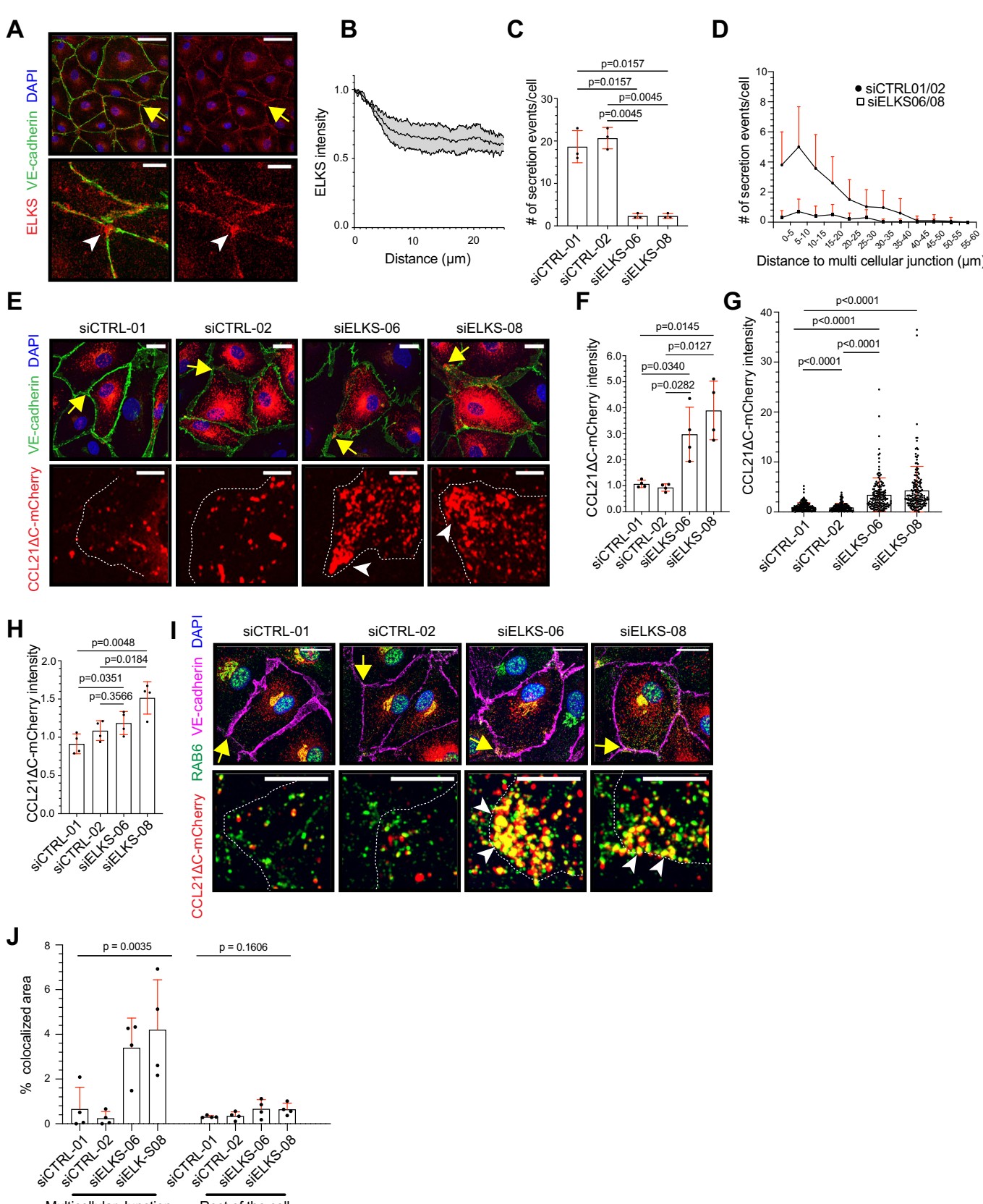

◀

**Figure 5. CCL21 is exocytosed at multicellular junctions in ELKS-dependent manner.**

Figure 5 is shown with alternative colors (non-red and -green) in Appendix Fig. S8. (A, B) LEC monolayer stained for endogenous ELKS (red), VE-cadherin (green), and nuclei (DAPI, blue). (B) A quantification of ELKS staining at the LEC junctions. The graph shows mean ELKS intensity ± SD as a function of distance from the nearest multicellular junction. In (A, B) the data represents a total of 250 junctions in 8 biological replicates and three independent experiments. (C, D) Quantification of the CCL21ΔC-mCherry exocytosis events at the LEC junctions in siControl and siELKS-treated LEC monolayers. The dot plot in (C) shows a mean number of observed exocytosis events/cell ± SD. Whereas, in (D), the histogram shows the distribution of mean number of exocytosis events +SD, at the LEC junctions, as a function of distance from a multicellular junction. The data in (C, D) is derived from 3 independent experiments, representing, altogether, $n = 15$ cells and 5 biological replicates per oligo (siCTRL01, siCTRL02, siELKS06, or siELKS08). In (D) the results from siCTRL01/02 or siELKS06/08 oligo transfected samples are pooled together. (E–H) Shows a LEC monolayer expressing CCL21ΔC-mCherry (red), treated with siControl or siELKS oligos and stained for VE-cadherin (green) and nuclei (DAPI, blue). The dot plots in (F, G) show the mean CCL21ΔC-mCherry intensity ± SD at the multicellular junction measured (F) per experiment and (G) per multicellular junction. Whereas the dot plot in (H) shows the mean CCL21ΔC-mCherry intensity ± SD measured in the whole LEC. Data points represent $n = 4$ independent experiments or $n = 164$ (siControl01), $n = 155$ (siControl02), $n = 162$ (siELKS06), and $n = 171$ (siELKS08) multicellular junctions in (F, G) and 207 (siControl01), 225 (siControl02), 202 (siELKS06), and 216 (siELKS08) LECs in (H), from 6 biological replicates. In (F–H) the results were normalized to the average of controls (set at 1) in each experiment. (I, J) siControl and siELKS transfected LECs expressing CCL21ΔC-mCherry (red) and stained for endogenous RAB6 (green), VE-cadherin (magenta) and nuclei (DAPI, blue). (J) The dot plot shows the mean percentage colocalized area + SD of CCL21ΔC-mCherry and endogenous RAB6 in siControl or siELKS transfected LECs at the multicellular junctions (i.e., ≤5 μm from the multicellular junction), or in the rest of the LEC. Each data point represents independent biological replicates ($n = 4$) from three experiments. Total number of analyzed multicellular junctions and LECs was 81 and 16, respectively, for siControl01, 91 and 18 for siControl02, 96 and 19 for siELKS06, and 92 and 19 for siELKS08. Data information: In (A), (E), and (I) yellow arrows in the overview images indicate the multicellular junctions (shown as a zoom-in), and white arrowheads examples of accumulation. In (E) and (I) white dotted lines mark the cell boundaries in the zoom-in images. In (C), (F) and (H) the p-values were calculated using a parametric T-test with Welch's correction, in (G) with a non-parametric Mann–Whitney's test, and in (J) with an ordinary ANOVA statistical test. The scale bar in (A) is 50 μm in the overview images and 10 μm in zoom-in images, and in (E and I) 20 μm in overview images and 5 μm in zoom-in images. Source data are available online for this figure.

accumulation to the site of exocytosis, i.e., multicellular junctions, may take longer in siRAB6 than siELKS conditions due to the lowered efficiency of vesicle trafficking to microtubule +ends. Accordingly, CCL21ΔC-mCherry, which is expressed for a longer period than CCL2 in our experimental context, accumulated at multicellular junctions upon RAB6 silencing (Fig. 6C–E). Altogether, our data suggests a two-component system, where microtubule targeting of multicellular junctions is causative for the RAB6 vesicle delivery to the site of transmigration, where RAB6+ vesicle docking factor ELKS drives the exocytosis.

To validate the relevance of CCL21 exocytosis at multicellular junctions for DC transmigration, we loaded LPS-activated DCs on CCL21ΔC-mCherry or CCL21-mCherry expressing control or siRAB6 silenced LEC monolayers, and analyzed the number of transmigration events in recorded movies. Here, silencing of lymphatic endothelial RAB6 resulted in a decreased number of transmigration events (by 29.9%) on CCL21ΔC-mCherry expressing monolayers (Fig. 6K, Movie EV11). RAB6 silencing did not affect the number of transmigration events across monolayers expressing full-length CCL21-mCherry (Fig. 6K, Movie EV12). This difference may be due to the accumulation of full-length CCL21-mCherry on the cell culture dish surface, which is not seen in vivo (Vaahtomeri et al, 2017). Thus, we presume that CCL21ΔC-mCherry recapitulates the in vivo conditions better. Altogether, these results show that RAB6 vesicle-mediated exocytosis of CCL21 at the multicellular junctions drives DC transmigration.

## Inhibition of RAB8A attenuates DC transmigration

Next, we searched for a second independent means to blunt CCL21 exocytosis at multicellular junctions. RAB6+ vesicle docking complex is composed of ELKS, RAB8A, and mono-oxygenase MICAL3 (Grigoriev et al, 2011). In our initial colocalization screen, RAB8A localized to the LEC plasma membrane and to some CCL21ΔC-mCherry+ vesicles at the multicellular junction (Fig. EV2A,H–K). Thus, we used RAB8A dominant negative (DN) construct (EGFP-RAB8A-T22N), which has been shown to inhibit RAB6+ vesicle-mediated exocytosis (Grigoriev et al, 2011).

To investigate whether RAB8A DN attenuates exocytosis at LEC junctions, we transduced LECs with CCL21ΔC-mCherry and EGFP (control) or EGFP-RAB8A-DN. Similar to ELKS silencing, EGFP-RAB8A-DN expression resulted in a strong inhibition of CCL21ΔC-mCherry exocytosis at the cell junctions (Fig. 7A), as observed in live microscopy, and a 40% reduction in mCherry signal in the culture media (Fig. 7B). Accordingly, expression of EGFP-RAB8A DN caused the accumulation of CCL21ΔC-mCherry+ vesicles in LECs, especially at the multi-cellular junctions, whereas expression of wild-type EGFP-RAB8A did not affect CCL21ΔC-mCherry levels (Fig. 7C–F). Confirming the RAB6+ route-specific effect of EGFP-RAB8A DN, the accumulated vesicles at the multicellular junctions were double positive for CCL21ΔC-mCherry and endogenous RAB6 (Fig. 7G). To corroborate these results, we also quantified the endogenous CCL2 levels in EGFP, EGFP-RAB8A DN, or wild-type EGFP-RAB8A transduced and TNF-α treated LECs. Similar to CCL21ΔC-mCherry, expression of EGFP-RAB8A DN resulted also in a strong accumulation of endogenous CCL2+ vesicles, especially, at multi-cellular junctions of TNF-α treated LECs (Fig. 7H–K).

Thus, the EGFP-RAB8A DN construct presented a second independent tool, in addition to siRAB6 to inhibit CCL21 exocytosis at the multicellular junctions. To solidify the notion that the CCL21 secretion at the multicellular junctions is required for the DC transmigration, we transduced the LECs with CCL21-mCherry or CCL21ΔC-mCherry together with EGFP or EGFP-RAB8A DN. The transduction efficiency was high, ensuring EGFP or EGFP-RAB8A-DN expression in large majority of LECs (Fig. 7L). LPS-activated DCs were loaded on the confluent monolayers, followed by live microscopy of the co-cultures. The analysis of recorded movies revealed that, in comparison to control co-cultures, EGFP-RAB8A-DN reduced DC transmigration events by 14% in CCL21-mCherry expressing cultures and by 38% in CCL21ΔC-mCherry expressing cultures (Fig. 7M, Movie EV13). Thus, EGFP-RAB8A-DN was more efficient in attenuating DC transmigration than siRAB6, which correlates with higher chemo-kine accumulation, i.e., lack of exocytosis, at the multicellular junctions. In conclusion, these results show that exocytosis of

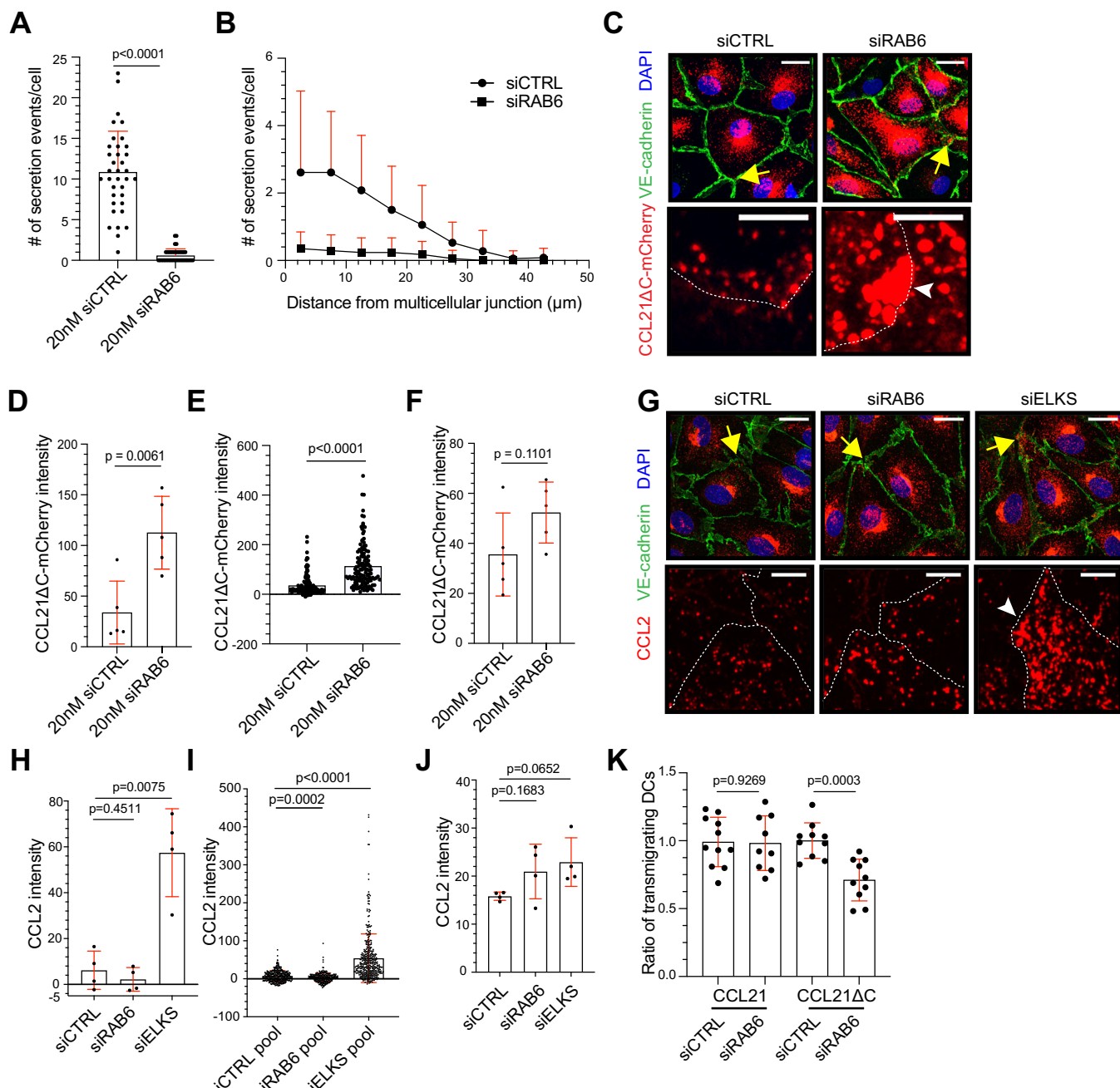

RAB6 + CCL21-mCherry+ double-positive vesicles at the multicellular junctions contributes to DC transmigration across the lymphatic endothelium.

## Discussion

Here, we present the first characterization of lymphatic endothelial exocytosis routes and show that RAB6+ vesicles deliver chemokine CCL21 to the multicellular junction, which represents the preferential site of DC transmigration across the lymphatic endothelium in dermal explants and primary LEC culture.

Importantly, inhibition of the RAB6+ vesicle exocytosis attenuated DC transmigration in primary LEC culture. Moreover, we identified that also monocyte chemoattractant protein CCL2 is trafficked to the multicellular junctions along the same delivery route. This suggests that chemokine exocytosis at multicellular junctions may represent a more general mechanism of leukocyte transmigration guidance. Altogether, our study supports a novel concept of targeted chemokine exocytosis as one of the determinants of the transmigration site.

Similar to DC transmigration across lymphatic endothelial multicellular junctions, as identified here, neutrophils transmigrate across the blood endothelium multicellular junctions and

**Figure 6.   RAB6 is required for the CCL21 exocytosis at multicellular junctions.**

Figure 6 is shown with alternative colors (non-red and -green) in Appendix Fig. S9. (A, B) The dot plot in (A), shows a mean number of observed exocytosis events/cell + SD in siControl and siRAB6 samples. Whereas, in (B), the histogram shows the distribution of the mean number of exocytosis events/cell + SD at LEC junctions, as a function of distance from a multicellular junction. In (A, B), the data points represent $n = 39$ cells in 13 biological replicates from three independent experiments. (C–F) Images of a LEC monolayer expressing CCL21ΔC-mCherry (red), treated with siControl or siRAB6 oligos and stained for VE-cadherin (green) and nuclei (DAPI, blue). The dot plot in (D, E) shows the mean CCL21ΔC-mCherry intensity ± SD measured at the multicellular junctions per (D) biological replicate and (E) per multicellular junction, whereas the dot plot in (F) shows the mean CCL21ΔC-mCherry intensity ± SD measured in the whole LEC. Data points represent $n = 5$ biological replicates or $n = 129$ (siControl) and $n = 129$ (siRAB6) multicellular junctions in (D, E) and 160 (siControl) and 164 (siRAB6) LECs in (F), altogether, in 4 independent experiments. (G–J) Images of TNF-α treated LEC monolayer transfected with siControl, siRAB6, or siELKS oligos, and stained for endogenous CCL2 (red), VE-cadherin (green), and nuclei (DAPI, blue). The dot plot shows the mean CCL2 intensity ± SD, measured at the multicellular junctions, (H) per experiment and (I) per multicellular junction. Whereas, in (J) the dot plot shows the mean CCL2 intensity ± SD, measured in the whole imaged area. Data points represent $n = 4$ independent experiments or $n = 337$ (siControl pool), $n = 268$ (siRAB6 pool), and $n = 384$ (siELKS pool) multicellular junctions from 5 (siControl pool and siRAB6 pool) or 6 (siELKS pool) biological replicates in (H, I), and 37 (siControl pool and siRAB6 pool), and 42 (siELKS pool) images from 6 (siControl pool and siRAB6 pool) or 7 (siELKS pool) biological replicates in (J). (K) Quantification of the mean DC transmigration efficiency ± SD on LEC monolayer transfected with siControl or siRAB6 and transduced with CCL21-mCherry or CCL21 ΔC-mCherry. The results were normalized to the average of control samples (set at 1) in each experiment. The data points represent biological replicates: CCL21-mCherry + siControl $n = 11$ (4359 DCs), CCL21-mCherry + siRAB6 $n = 9$ (4091 DCs), CCL21ΔC-mCherry + siControl $n = 10$ (3319 DCs), CCL21ΔC-mCherry + siRAB6 $n = 10$ (3766 DCs), across three independent experiments. The data is related to Movies EV11 and EV12. Data information: In (C) and (G) yellow arrows indicate the multicellular junction (area shown in the zoom-in images below), and white arrowheads indicate the accumulation. White dotted lines represent the cell borders. In (A), (D), (F), (H), (J), and (K), the p-values were calculated using a parametric T-test with Welch's correction and in (E) and (I) the p-values were calculated using Mann–Whitney's test. The scale bar in (C) and (G) is 20 μm in overview images and 5 μm in zoom-in images. Source data are available online for this figure.

multiciliary cells integrate to Xenopus lung epithelium at multi-cellular vertices (Burns et al, 1997; Gorina et al, 2014; Ventura et al, 2022; Wang et al, 2006). Despite a long-standing interest in factors that determine the site of transmigration, it is poorly understood, what guides leukocytes to these sites and drives the transmigration. Here, by using primary lymphatic endothelial culture as a model system, we show that the lymphatic endothelial chemokine CCL21 guides CCR7+ DCs on the plane of the monolayer and promotes the arrest of the DCs at the multicellular junctions. The identification of chemokine CCL21 exocytosis at multicellular junctions supports a model where chemokine exocytosis creates both a landmark (high CCL21) at a transmigration site and a guidance cue (a gradient in the plane of the LEC monolayer). It is conceivable that, apart from CCL21, also other components contribute to the identification of transmigration sites on lymphatic endothelium. This was evidenced by a considerable number of CCR7$^{-/-}$ DCs that arrested at multicellular junctions (Fig. 1G). These components may include, membrane protrusions (Arts et al, 2021), and leukocyte adhesion-promoting proteins, such as ICAM (Arasa et al, 2021b; Johnson et al, 2006; Johnson and Jackson, 2010; Sumagin and Sarelius, 2010; Teijeira et al, 2013), which are enriched at multicellular vertices and allow leukocytes to exert force on the endothelium (Yeh et al, 2018). Here, we propose a working model where a combination of chemical cues and biomechanics determine the multicellular junctions as preferential leukocyte transmigration sites.

Here, we described localized chemokine exocytosis and DC transmigration in conditions of continuous lymphatic endothelial zipper-like junctions, similar to in vivo in conditions of inflammation (capillaries and collectors) and in homeostasis (downstream capillaries and collectors) (Arasa et al, 2021b; Baluk et al, 2007; Yao et al, 2012). In homeostasis, however, initial lymphatic capillary LECs have distinct discontinuous junctions and oak-leaf-like shape (Baluk et al, 2007), which is assumed to allow an easy entry of leukocytes into the lumen of the vessels. Similar to continuous junctions, the entry across the flap-like openings of initial lymphatic capillaries is CCL21-CCR7-dependent (Pflicke and Sixt, 2009; Tal et al, 2011). Interestingly, CCL21 + RAB6+ double-positive vesicles were enriched at the discontinuous VE-cadherin

junctions in vivo, which suggests that targeted CCL21 exocytosis also guides DCs to transmigrate across discontinuous lymphatic capillary endothelial junctions in homeostasis.

Targeted exocytosis would be expected to result in high local extracellular chemokine concentration. Thus, our finding of CCL21 exocytosis at LEC junctions may, in part, provide an explanation for the observed CCL21 deposits at the lymphatic endothelial basement membrane in the vicinity of DC transmigration sites in vivo and ex vivo (Tal et al, 2011; Vaahtomeri et al, 2017; Vaahtomeri et al, 2021; Weber et al, 2013). It is conceivable that the formation of CCL21 depots at the lymphatic capillaries in vivo is dependent on both the targeted exocytosis and CCL21 anchoring. CCL21 contains a charged carboxy-terminus, which interacts with glycosaminoglycans and basement membrane component collagen IV (Bao et al, 2010; Hirose et al, 2002; Vaahtomeri et al, 2021). In an analogy, neutrophil-expressed chemokine CXCL2 is presented at blood endothelial junctions by binding to a decoy receptor ACKR1 (Girbl et al, 2018). Whether any of the potential CCL21 anchoring-mediating interactions are enriched at lymphatic endothelial multicellular junctions is not known.

Our mini-screen of 13 RAB-GTPases revealed that RAB6+ vesicles deliver chemokines CCL21 and CCL2 to multicellular junctions in primary lymphatic endothelial cultures. Importantly, CCL21 also localized to RAB6A+ vesicles at the lymphatic endothelial multicellular junctions in dermis in vivo. Based on our data, we propose that exocytosis at multicellular junctions depends on two factors: (i) the RAB6A+ vesicle tracks, i.e., microtubules, which target the multicellular junctions, and (ii) RAB6A+ vesicle docking factors ELKS and RAB8A, which are enriched at these sites. Accordingly, inhibition of RAB6+ vesicle fusion to the plasma membrane by silencing of RAB6 or ELKS, or expression of a dominant negative RAB8A resulted in attenuated CCL21 exocytosis and marked accumulation of CCL21/RAB6A+ vesicles specifically at the multicellular junctions. These results are in line with previous studies, which have identified ELKS and RAB6 as determinants of specific sites of exocytosis, i.e., focal adhesions, cell contact-free leading edge, and pre-synaptic boutons (Fourriere et al, 2019; Grigoriev et al, 2007; Nyitrai et al, 2020). The ELKS localization at multicellular junctions was, in part, dependent on

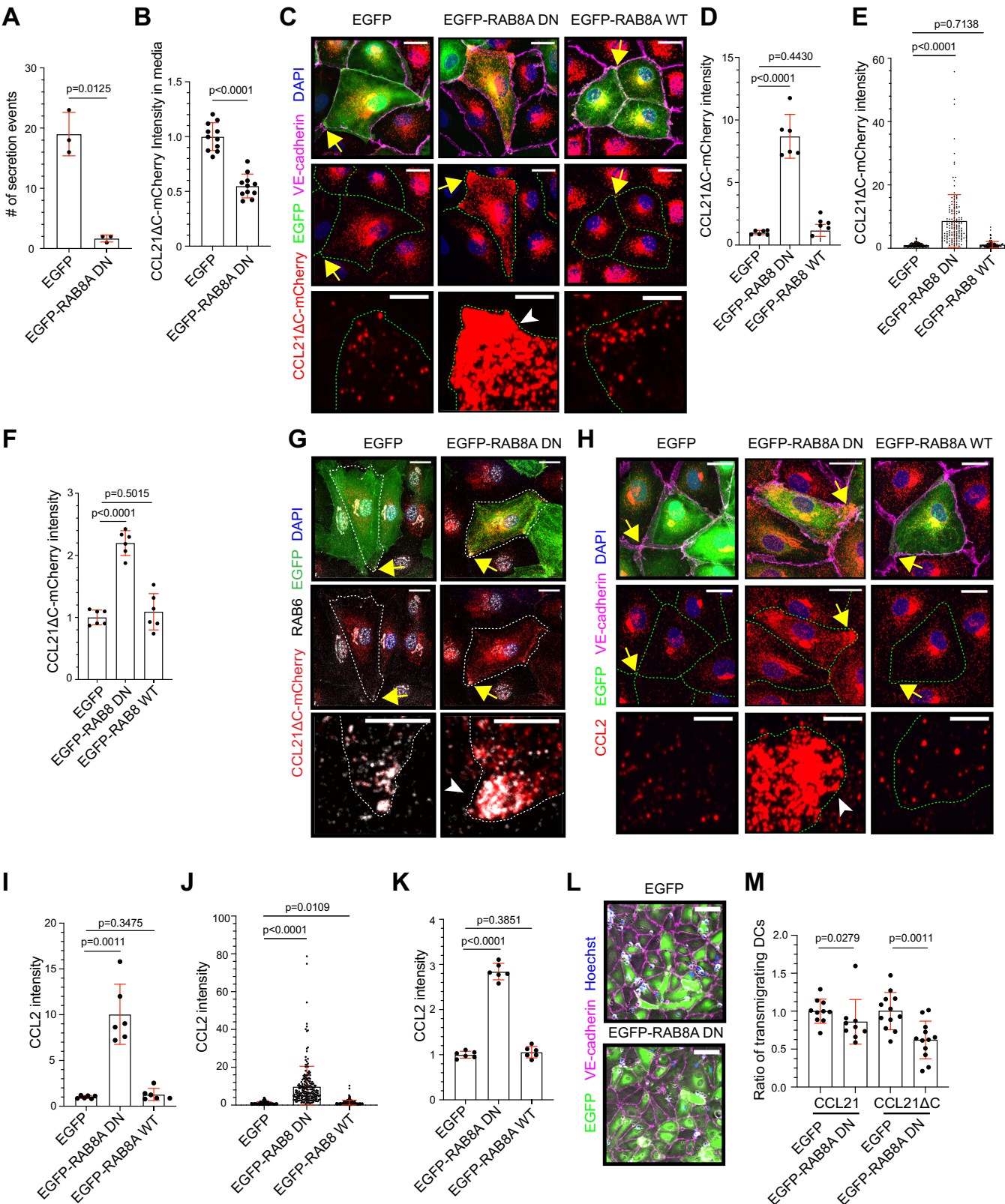

**Figure 7.  Dominant negative RAB8A inhibits CCL21 exocytosis and DC transmigration.**

Figure 7 is shown with alternative colors (non-red and -green) in Appendix Fig. S10. (A) Quantification of the CCL21ΔC-mCherry exocytosis events at the LEC junctions in EGFP or EGFP-RAB8A DN expressing LECs. The dot plot shows the mean number of exocytosis events/cell ± SD. Data points represent $n = 3$ independent experiments, comprising of, altogether, 24 LECs in 12 biological replicates. (B) Quantification of the effect of EGFP-RAB8A DN expression on CCL21ΔC-mCherry intensity in the media. The dot plot shows CCL21ΔC-mCherry mean intensity ± SD, and normalized to the average of controls, which was set at 1 in each experiment. The data points represent $n = 11$ biological replicates in three independent experiments. (C–F) Shows a LEC monolayer expressing CCL21ΔC-mCherry (red) and EGFP, EGFP-RAB8A DN, or EGFP-RAB8A WT (green). LECs were stained for VE-cadherin (magenta) and nuclei (DAPI, blue). Quantification of the mean CCL21ΔC-mCherry intensity ± SD at the multicellular junctions of EGFP-mCherry double-positive LECs (D) per biological replicate and (E) per multicellular junction. Whereas the dot plot in (F) shows the mean CCL21ΔC-mCherry intensity ± SD measured in the whole LEC. The results were normalized to the average of controls, which was set at 1, in each experiment. The data points represent $n = 6$ biological replicates in three independent experiments or $n = 149$ (EGFP), $n = 153$ (EGFP-RAB8A DN), and $n = 147$ (EGFP-RAB8 WT) multicellular junctions in (D, E) and 212 (EGFP), 188 (EGFP-RAB8A DN), and 170 (EGFP-RAB8 WT) LECs in (F). (G) Immunofluorescence images of a LEC monolayer expressing CCL21ΔC-mCherry (red) and either EGFP or EGFP-RAB8A DN (green). The cells were stained for endogenous RAB6 (gray), and nuclei (DAPI, blue). The data represents $n = 4$ biological replicates from 2 independent experiments. (H–K) Shows TNF-α treated LEC monolayer expressing EGFP (control), EGFP-RAB8A DN, or EGFP-RAB8A WT (green). LECs were stained for CCL2 (red), VE-cadherin (magenta), and DAPI (blue). Quantification of the mean CCL2 intensity ± SD at the multicellular junctions (I) per biological replicate and (J) per multicellular junction. Whereas the dot plot in (K) shows mean CCL2 intensity ± SD measured in the whole LEC. The data was normalized to the average of controls (set at 1) in each experiment. The data points represent $n = 6$ biological replicates from 3 independent experiments or $n = 204$ (EGFP), $n = 202$ (EGFP-RAB8A DN), and $n = 192$ (EGFP-RAB8 WT) multicellular junctions in (I, J), and 242 (EGFP), 240 (EGFP-RAB8A DN), and 136 (EGFP-RAB8 WT) LECs in (K). (L, M) A capture of live recording of LEC monolayer expressing CCL21-mCherry and either of EGFP or EGFP-RAB8A DN (green) and stained for VE-cadherin (magenta). The nuclei of dendritic cells (DC) are stained with Hoechst (blue). (M) Quantification of the mean DC transmigration efficiency ± SD on LEC monolayer co-expressing either EGFP (control) or EGFP-RAB8A DN together with either CCL21-mCherry or CCL21 ΔC-mCherry. The results were normalized to the average of control (set at 1) in each experiment. The data points represent $n = 10$ CCL21-mCherry + EGFP (total of 3285 DCs), $n = 9$ CCL21-mCherry + EGFP-RAB8A DN (2552 DCs), $n = 12$ CCL21 ΔC-mCherry + EGFP (7123 DCs), and $n = 12$ CCL21 ΔC-mCherry + EGFP-RAB8A DN (5535 DCs) biological replicates, across 4 independent experiments. The data in (L, M) is related to the Movie EV13. Data information: In mCherry channel-only images in (C), the cell borders of all the EGFP-mCherry, double-positive LECs, are indicated with green dashed lines. Similarly, in (H) the green dashed lines indicate the cell borders of all EGFP-CCL2 double-positive LECs. Cell borders in (G) are shown with white dashed line. In (C), (G), and (H), yellow arrows indicate the multicellular junction shown in the zoom-in images and the white arrowheads the accumulation. In (A, B), (D), (F), (I), (K), (M, CCL21 ΔC-mCherry samples), the *p*-values were calculated using a parametric T-test with Welch's correction, and in (E), (J), and (M, CCL21-mCherry samples) using Mann–Whitney's test. Scale bars in (C, G and H) are 20 μm in overview images and 5 μm in zoom-in images and in (L) 50 μm. Source data are available online for this figure.

PI3-kinase activity. Earlier, ELKS localization at the plasma membrane has been shown to depend on LL5β (Lansbergen et al, 2006), which directly interacts with PIP3 (Paranavitane et al, 2003). At the plasma membrane LL5β represents a scaffold, which recruits and stabilizes interactions with both ELKS and microtubule +ends (Lansbergen et al, 2006). We propose that in LECs, multicellular junctions and the associated protein complexes present a hub that coordinates secretory protein delivery and exocytosis. Further investigations are required to determine whether, for example, PI3K activity controls multicellular junctions in response to acute growth factor stimuli.

In this study, we identified RAB6A-positive Golgi-derived vesicles and RAB3D/27A-positive dense core secretory granules as the major lymphatic endothelial secretory routes of chemokines CCL21 and CCL2. It is conceivable that these routes traffic various other secretory and transmembrane proteins, such as chemokine CX3CL1 (Johnson and Jackson, 2013), which affect lymphatic endothelium-leukocyte interactions. In an analogy, blood endothelial RAB3+ and RAB27A+ Weibel-Palade bodies contain, for example, P-selectin, thus, affecting leukocyte-to-blood endothelium adhesion (Lowenstein et al, 2005). Our studies highlight the spatial differences between these two routes: Whereas RAB6+ vesicles traffic CCL21 to multicellular junctions in primary cell culture and in vivo, RAB27A and RAB3D vesicles are located in the central body in primary LECs. Similarly, in blood endothelium, RAB27+ and RAB3+ Weibel-Palade bodies are secreted at the basal and apical surfaces (Lopes da Silva and Cutler, 2016; McCormack et al, 2017), whereas RAB6 vesicles localize to a leading edge of migrating endothelial cells in culture (Martin et al, 2018). Interestingly, in blood endothelium, both CCL2- and CX3CL1-containing vesicles are found at the site of lymphocyte interaction (Schoppmeyer et al, 2022; Shulman et al, 2011), but the identity of these vesicles is not

yet known. Further, the RAB6+ and RAB3/27 routes may have profound differences in their responsiveness to various cues: whereas RAB6 vesicles mediate constitutive exocytosis (Fourriere et al, 2019; Grigoriev et al, 2007), RAB27+ and RAB3+ secretory granules have been indicated in calcium-dependent conditional secretion, also, in blood endothelial cells (Nass et al, 2021). Previously, we reported that CCL21 exocytosis, in part, is calcium-dependent in primary LECs and that extracellular CCL21 is enriched at the dermal lymphatic vessels upon DC transmigration ex vivo (Vaahtomeri et al, 2017). This conditional arm of CCL21 exocytosis may be accounted for the RAB3/27 positive CCL21 storage granules in primary cell cultures and RAB6 negative CCL21 storage granules in vivo. The RAB6 negative CCL21 vesicles are found also at the LEC junctions in vivo, although, more dispersed throughout the LEC than the RAB6+ positive CCL21+ vesicles, which are more restricted to multicellular junctions and LEC borders (Fig. 2J–P; Appendix Fig. S2A–F). Based on the previous and current results, we propose a model, where RAB6+ vesicle-mediated CCL21 exocytosis at the LEC multicellular junctions and LEC borders forms a constitutive cue that guides DCs to the preferred transmigration sites. Whereas, a conditional burst of RAB6 negative CCL21+ vesicles, upon DC-LEC contact, may mark the arrival of a DC to the lymphatic vessel, thus, slowing down the fast interstitial migration (Pflicke and Sixt, 2009; Tal et al, 2011; Weber et al, 2013), and/or mark the sites of successful transmigration (Tal et al, 2011; Vaahtomeri et al, 2017). Thus, the two mechanisms of CCL21 exocytosis would be cooperating. It is conceivable, that the two routes may also play distinct roles, also, in the exocytosis of large quantities of CCL21 to drive the formation of CCL21 interstitial gradient (Weber et al, 2013), or stabilize LEC–leukocyte interactions observed at peripheral and lymph node lymphatic endothelial cells in vivo (Kedl et al, 2017; Russo et al,

2016), possibly, for the purpose of antigen transfer (Kedl et al, 2017). The spatially controlled exocytosis, as characterized here, is expected to synergize with other reported control mechanisms of CCL21 presentation in space, including anchoring of CCL21 (Bao et al, 2010; Hirose et al, 2001; Vaahtomeri et al, 2021; Yang et al, 2007; Yin et al, 2014), cleavage by DCs (Schumann et al, 2010), tissue fluid and lymph flow (Miteva et al, 2010; Russo et al, 2016), and decoy receptor-mediated sequestration (Friess et al, 2022; Ulvmar et al, 2014). We envision that spatio-temporal control of CCL21 presentation and, possibly, semaphorin-plexin driven DC migration (Takamatsu et al, 2010), guide the DCs to multicellular junctions, where CCL21 together with direct DC-LEC contacts mediated by hyaluronan-LYVE1 (Johnson et al, 2017; Stanly et al, 2020), and, upon inflammation, integrin-ICAM/VCAM (Arasa et al, 2021b; Johnson et al, 2006; Johnson and Jackson, 2010; Teijeira et al, 2013), allow the DCs to arrest, exert force, and transmigrate. In this context, we propose that the use of two distinct exocytosis routes allows high-fidelity control over chemokine presentation and DC guidance to the site of transmigration.

Altogether, our results provide a platform for exploring spatiotemporal control of lymphatic endothelial paracrine functions. Interestingly, lymphatic endothelium also controls stem cell activity in the intestine (Niec et al, 2022; Palikuqi et al, 2022), heart (Liu et al, 2020), and hair follicles (Gur-Cohen et al, 2019; Pena-Jimenez et al, 2019), suggesting an instructive role for LEC exocytosis even beyond the LEC–leukocyte interactions. Furthermore, targeted exocytosis at the multicellular junctions may present a general mechanism relevant to multiple functions in endothelial and epithelial biology.

# Methods

## DNA constructs

Lentiviral pLenti6.3-CCL21-mCherry, pLenti6.3-CCL21ΔC-mCherry, and pLenti6.3-EB3-mCherry constructs have been published previously (Kopf et al, 2020; Vaahtomeri et al, 2017).

The novel constructs published in this article are available from the corresponding author upon a reasonable request. The novel constructs were sub-cloned as detailed below:

(i) For RAB1A, 2A, 3D, 5C, 6A, 7A, 10, 13, 27A, 35 and 37 the template cDNAs were ordered from the Orfeome or MGC libraries existing at the Gene Biology Unit (GBU), University of Helsinki. The EGFP (derived from FUW-mLef1v1-EGFP), linker (GGAGGAAGCGGTGGAAGTGGTGGTTCT) and the RAB-GTPase were assembled into pDONR221 (Gateway, 12536017), using restriction digestion sub-cloning or Gibson assembly, to yield pENTR221-EGFP-RAB constructs.

(ii) EGFP-RAB8A, EGFP-RAB8A T22N dominant negative (DN) mutant, and EGFP-RAB11A fusions were amplified via PCR from previously published constructs (Hattula et al, 2006). Gateway™ compatible Att- sites were added to the PCR product in the same amplification PCR, using appropriate primer pairs. Next, the amplified fragments were recombined with pDONR221 backbone, using Gateway™ BP clonase reaction (Invitrogen, 11789020) to yield pENTR221-EGFP-RAB constructs.

(iii) The mCherry-RAB27A constructs were created by using the Gibson assembly protocol. Primers were designed according to the manufacturer's instructions (New England Biolabs). We used EcoRI-BamHI restriction digestion to cut pENTR221 backbone from pENTR221-EcoRI-RAB2A-BamHI vector. Then, we used pENTR221-CCL21-mCherry to amplify mCherry and pENTR221-EGFP-RAB27A to amplify RAB27A. The PCR products were purified using gel electrophoresis and correct bands of DNA were extracted using gel purification kit (Machery-Nagel, 740609.50). The DNA fragments were assembled using Gibson assembly master mix (New England Biolabs, E2611S) to produce pENTR221-mCherry-RAB27A construct.

(iv) The signaling peptide (SP)-mCherry construct, was produced using pENTR-D-TOPO-CCL21-mCherry (Vaahtomeri et al, 2017), as a template. Here, we designed primers according to the manufacturer's instructions (Thermo Scientific, F-541) to delete the body of the CCL21 (base pairs 70–399, corresponding to aa 24–133), resulting in a construct, which encoded signaling peptide (SP; aa 1–23) of CCL21 fused to mCherry (pENTR-D-TOPO SP-mCherry).

Subsequent to the sub-cloning reactions described in (i–iv), the resulting plasmids were transformed into DH5α derived chemically competent *Escherichia coli* cells- NEB 5-alpha (New England Biolabs, C2987H) and grown overnight at 37 °C. Next day, the bacterial colonies were analyzed using colony PCR. Upon positive colony PCR result, mini-prep cultures were grown overnight, the DNA was extracted using DNA extraction kit (Machery Nagel, 740588.50) and analyzed using restriction digestion followed by gel electrophoresis. Subsequently, the plasmids, which displayed the correct restriction pattern, were further validated by sequencing the whole AttL1-cDNA-AttL2 cassette (Eurofins Genomics, Germany).

Next, the correct clones were transferred from pENTR-vectors to pLenti6.3 vector, in a Gateway™ LR-reaction according to the manufacturer's protocol (Invitrogen, 11791-020). Then, the LR product was transformed into HB101 derived chemically competent *Escherichia coli* cells-One Shot™ Stabl3™ (Thermo Scientific, C737303) and grown overnight at 37 °C. DNA was extracted from mini-prep cultures and analyzed using restriction digestion followed by gel electrophoresis. The correct clones were grown as maxi-cultures, and the DNA was isolated using maxi-prep DNA isolation kit (Machery Nagel, 740414.10) according to manufacturer's protocol.

## Virus production

All the lentiviruses were produced in LentiX-293 packaging cells (Chemicon, Darmstadt, Germany) according to the protocol provided by Functional Genomics Unit (FuGU), University of Helsinki (https://www2.helsinki.fi/en/infrastructures/genome-editing-function-and-stem-cell-platform/infrastructures/fugu-lentiviral-libraries/resources#section-91406).

## Mice

Adult wild-type female and male C57BL/6JRcc mice were produced locally at the Laboratory Animal Center of the University of Helsinki. All the mice were bred and handled according to the local ethical regulations. Animal housing was in conditions of 12-h light/dark cycle, 23 °C ambient temperature, and approximately 50–60%

humidity. The tissue collection was done at the age of 8–12 weeks. Experimental procedures were approved by the Project Authorization Board in Finland (animal licenses ESAVI/30523/2019, ESAVI/40857/2022, and ESAVI/27566/2023.

## Generation of bone marrow-derived dendritic cells

Activated bone marrow-derived DCs were generated, as before (Vaahtomeri et al, 2017), from control and femur tibias of wild-type and CCR7$^{-/-}$ C57BL6 mice (Forster et al, 1999). The CCR7$^{-/-}$ and corresponding control femurs and tibias were received as a generous gift from Dr. Michael Sixt laboratory (IST Austria). The extracted bone marrow was cultured in R10 culture medium, i.e., RPMI1640 (Corning, 15-040 or EuroLone, ECB9006L), supplemented with 10% fetal calf serum (Gibco, 10270-106), 1% L-Glutamine (Corning, 25-005-CI) and 1% Penicillin/Streptomycin mix (Lonza, DE17-603E), and 10% GM-CSF hybridoma supernatant (from cultured GM-CSF hybridoma cells). CSF hybridoma cells were a generous gift from Dr. Michael Sixt lab (IST, Austria). The media was changed on days 3 and 6 and on day 8 the DCs were frozen in freezing medium, consisting of 90% FCS and 10% DMSO. For experimentation, the DCs were thawn in R10 culture medium, centrifuged at 180 RCF and dissolved in R10 culture medium supplemented with 200 ng/ml lipolysaccharide from *Escherichia coli* O26:B6-LPS (Sigma L2654). After 21-h culture, the activation of DCs was confirmed by the presence of dendrites on the DC surface, observed by cell-culture phase contrast microscope (Motic, AE31E).

## Primary lymphatic endothelial cell (LEC) culture

All the experiments were conducted by using primary human male juvenile foreskin dermal lymphatic endothelial cells (LEC, Promocell, C-12216). LECs were expanded (passages 2–4) at 37 °C and 5% $CO_2$ in endothelial cell growth medium MV2 (Promocell, C-22022) containing all the provided supplements and, also, gentamicin and amphotericin (GA) (Lonza, cc-4083). Expanded LECs were frozen in Cryo-SFM freezing medium (Promocell, C-29910). After expansion, LEC identity was checked by staining for PROX1 (R&D Systems, AF2727). For experimentation, LECs were used at passage 5. Frozen LECs were directly plated on experimental wells coated with 0.1% gelatin. Depending on the experiment, a 10-well chambered slide with cover glass bottom (0.17 mm thickness) (Greiner, 543078), 18-well chambered slide with high optical glass bottom (0.15 mm thickness) (Ibidi, 81817), or 48-well cell culture plate (Thermo Scientific, 150687), were used. CCL21 is down-regulated in cell culture conditions (Wick et al, 2007). At p5 endogenous CCL21 is observed only at low amounts in Golgi (Johnson and Jackson, 2010). To overcome this, we lentivirally express CCL21-mCherry, which recapitulates the CCL21 in vivo localization and function (see Fig. 2J–L; Appendix Fig. S2A,F, and (Vaahtomeri et al, 2017)). To induce CCL2 expression in some of the experiments (Figs. 2G,I, 6G–J, 7H–K and EV4A; Appendix Fig. S4B), we treated the LECs with 20 ng/ml of tumor necrosis factor-α (TNF-α; R&D Systems, 210-TA-005), diluted in full MV2 (without GA) for 24 h at 37 °C and 5% $CO_2$. LECs were either imaged live or fixed with 4% paraformaldehyde (PFA; Histolab, HL96753) for 15′ followed by three washes with phosphate buffered saline (PBS, pH 7.4).

## Transfection for siRNA-mediated silencing

For siRNA-mediated silencing, we used the following smart pool oligos (Dharmacon™/Horizon): siRAB6 (catalog # L-008975-00-0005), and siERC1 (ELKS; catalog # LQ-010942-00-0002) along with a control pool oligos siCTRL (catalog # D-001810-10-05) (see Fig. EV5D and F,G). For the knockdown of CCL21 and ELKS mRNA, we also tested four individual oligos: siRNA-CCL21 (catalog # J-007833-05 to -08) and siRNA-ERC1 (catalog # J-010942-05 to -08). Of these, the two most efficient oligos were selected and used in the experiments (for siCCL21 catalog # J-007833-05 and J-007833-08, see Fig. EV1M; For siERC1 catalog # J-010942-06 and J-010942-08, see Fig. EV5E). As a control, non-targeting siRNA 1 (catalog # D-001810-01-05) and 2 (catalog # D-001810-02-05) oligos were used. All the oligos were dissolved to stock concentration of 4 μM in RNA dilution buffer (Dharmacon™/Horizon, catalog # B-002000-UB-100), according to the protocol provided by the manufacturer (Horizon). For the experimentation, 20 nM final concentration of oligos was used.

For silencing of the target genes, primary LECs were plated on gelatin-coated cell culture slides on day 0. On day 1, LECs were transfected with RNAiMax transfection reagent (Invitrogen, 13778100), according to the manufacturer's protocol. In short: transfection mix of oligos and RNAiMax was prepared in reduced serum Opti-mem (Gibco, 11058-021). Meanwhile, after washing once, 150 μl of "transfection media" (basal MV2 media, Promocell C-22221) supplemented with all MV2 supplement pack growth factors (Promocell, C-39221), but excluding FCS, was added on LECs. Then, 50 μl of transfection mix was added dropwise on the LEC cultures. On day 2, the 24-h transfection was terminated by replacing the transfection media with full MV2, which was replaced daily. Transfected LECs were used for experimentation on day 3–4.

## Lentiviral transduction

Lentiviral transduction was performed either 16 h after plating the primary LECs (day 1), or on day 2, subsequent to siRNA transfection of LECs. For latter, siRNA transfection media was replaced by full MV2 for 2 h, to let LECs recover, before starting the lentiviral transduction. For transductions, hexadimethrine bromide (polybrene; final concentration of 8 μg/ml, Sigma, H9268) and lentiviruses were added to full MV2 media (without GA antibiotics). The LECs were incubated with the transduction mix for 5–6 h, after which the media containing the lentiviruses was removed and cells were washed with PBS, and fresh full MV2 media (with GA antibiotics) was added on the LECs and replaced daily. The LECs were used for experimentation on day 3–4.

## Staining of LECs

In the end of the experimentation, the cultured LECs were washed once with PBS and fixed with 4% paraformaldehyde (PFA; Histolab, HL96753) for 15′ at room temperature. For ELKS staining, LECs were fixed with 1% PFA. After fixation the cells were washed three times with PBS. The fixed and PBS-washed samples were permeabilized with 0.15% Triton x-100 in PBS for 15′, washed three times with PBS and blocked with 3% bovine serum albumin (BSA) in PBS for 1 h. Primary antibodies were diluted in blocking buffer supplemented with 0.1% Triton x-100

and the samples were incubated overnight at 4 °C with the following antibodies: Purified mouse anti-human VE-cadherin (BD bioscience, 555661; dilution of 1:200) (Figs. 2A, 5I, 6C and EV5G; Appendix S4C); rabbit anti-human VE-cadherin (D87F2, Cell signaling, 2500S; 1:200) (Figs. 5A,E and EV5A, 6G, 7C,H; Appendix S4A,B); rabbit anti-human RAB6 (D37C7, Cell signaling, 9625S; 1:100) (Figs. 2H,I, 5I, 7G and EV5G; Appendix Fig. S4C); rabbit anti-human RAB7 (D95F2 XP, Cell signaling 9367TS; 1:200) (Fig. 3D–F); mouse anti-human CCL2 (24822, R&D Systems, MAB279; 1:50) (Figs. 2G,I, 6G, 7H and EV4A; Appendix Fig. S4B); mouse anti-human α-Tubulin (Santa-Cruz biotechnology, sc-32293; 1:200) (Fig. 4B,C); mouse anti-human ELKS (E-1, Santa-Cruz biotechnology, sc-365715; 1:200) (Figs. 5A and EV5A,D). After incubation with the primary antibodies, the cells were washed three times with PBS and fluorophore conjugated secondary antibodies (diluted 1:400 in blocking buffer), were added to the cells for 2 h. The following secondary antibodies were used: Alexa Fluor 488 donkey anti-rabbit (Invitrogen, A21206); Alexa Fluor 488 donkey anti-mouse (Invitrogen, A21202); Alexa Fluor 594 donkey anti-mouse (Invitrogen, A21203); Alexa Fluor 647 donkey anti-rabbit (Invitrogen, A31573); Alexa Fluor 647 donkey anti-mouse (Invitrogen, A31571). After the secondary antibody staining, the cells were washed three times with PBS and stored in PBS containing 0.05% sodium azide at 4 °C. Prior to imaging, the LECs were counterstained with DAPI (Tocris, 5748) for 5′ and washed 3 times with PBS.

For live microscopy, cultured LECs were stained with fluorophore-coupled non-blocking Alexa Fluor 647 conjugated mouse anti-human VE-cadherin (BD pharmingen, 561567, dilution of 1:1000) for 30′ at 37 °C and 5% CO$_2$. Upon replacing the staining media with fresh full MV2 media, the cells were immediately used for live imaging (Figs. 1A,E, 2B, 4A,D, and 7L; Movies EV1 and EV6–13).

## Ear pinna sheet staining

The ventral and dorsal ear pinnae sheets were prepared as before (Vaahtomeri et al, 2017) and fixed with 1%PFA for 1 h at room temperature. The fixed ear sheets were washed three times with PBS, permeabilized with 0.3% Triton x-100 in PBS for 30′, blocked with donkey immunomix blocking buffer ((DDIM; 5% donkey serum (Biowest, S2170), 0.2% BSA (Biowest, P6154), 0.05% sodium azide (Sigma-Aldrich, S2002) and 0.3% Triton X-100 (Fisher Bioreagents, BP151) in Dulbecco's PBS (Corning, 21-030-CVR), for 1 h at room temperature. Next, the ear sheets were incubated with primary antibodies (diluted in blocking buffer), for 2-to-3 days at +4 °C. The used primary antibodies were: a mix of rat anti-mouse VE-cadherin (Biolegend, 138003; dilution of 1:100) and rat anti-mouse VE-cadherin (Invitrogen, 14-1441-81; 1:100) (Figs. 2K,L and EV1A; Appendix Fig. S2A–F); goat anti-mouse CCL21/6Ckine biotinylated antibody (R&D BAF457; 1:300) (Fig. 2J–L and Appendix Fig. S2A–F); rabbit anti-RAB6 (D37C7, Cell signaling, 9625S; 1:100) (Fig. 2J–L; Appendix Fig. S2A–F); rat anti-mouse LYVE1 (R&D, MAB2125; 1:300) (Fig. 2J); rabbit anti-mouse LYVE1 (a kind gift from Dr. Kari Alitalo lab (Karkkainen et al, 2004); 1:1000) (Fig. EV1A). After incubation with primary antibodies, the ear sheets were washed three times with PBS followed by incubation with fluorophore-coupled secondary antibodies, diluted in blocking buffer, for 2 days at +4 °C. The used secondary antibodies were Alexa Fluor donkey anti-rat 594 (Invitrogen, A21209, dilution of 1:400); Alexa Fluor 488 donkey

anti-rabbit (Invitrogen, A21206; 1:400). Alexa Fluor 647 Streptavidin (Jackson Immuno Research, 016-600-084; 1:300) was used to detect the biotinylated anti-mouse CCL21. After staining, ear sheets were washed three times with PBS and stored in PBS containing 0.05% sodium azide at 4 °C. In Fig. 2J, prior to imaging, ear sheets were counter-stained with DAPI (Tocris, 5748) for 15′ and washed 3 times with PBS.

## Microscopy set-up used for live microscopy and imaging of fixed samples

(i) For images shown in Figs. 2A,D,G,I, 3D–F,I, 4C,D, 5A,E,I, 6C,G, 7C,G,H, EV2A–C, EV2H,I, EV3A,B, EV4A, EV5A, EV5D, EV5G, Appendix S4A–C and Movie EV10, we used Andor dragonfly spinning disc microscope equipped with a full-enclosure atmospheric incubator (37 °C, 90% humidity and 5% CO$_2$), Plan Apo λ 20X dry objective (NA-numerical aperture, 0.75), plan Apo VC 60X water objective (NA 1.2) or SR Apo TIRF 100X oil objective (NA 1.49). The signal was captured using the Andor Zyla 4.2 sCMOS, Andor Sona sCMOS, or Andor iXon 888 U3 EMCCD cameras and Fusion 2.0 software. The microscope was equipped with Lumencor Sola light engine with 4 solid-state light sources including DAPI-50606, GFP-30309, TRITC-B 05, mCherry and a white light source with TR-POL-DIC-FWL polarizer, for visual inspection of samples. The laser sources available on the system were 404, 488, 561, 642, and 730 nm.

(ii) For images shown in Figs. 2H,J and 4B, we used Zeiss LSM 880 inverted microscope equipped with a Plan-APOCHROMAT 63x oil objective (NA 1.4), PMT, GaAsP, AiryScan (fast) and PicoQuant FLIM detector and operated with Zeiss Zen 2 software. The LSM 880 microscope was equipped with 405 nm and 561 nm diode lasers, 458 nm, 488 nm, and 514 nm Argon lasers and a 633 nm Helium-Neon laser. System-recommended settings for AiryScan filters, main and secondary beam splitters were used.

(iii) For images shown in Fig. 2K,L and Appendix S2A–F, we used Leica Stellaris 8 FALCON/DLS inverted microscope (DMI8) equipped with HC PL APO CS2 63X oil objective (NA 1.40) and 5 sensitive HyD detectors (HyD S and HyD X). The laser sources available on the system were a 405 nm diode laser and a supercontinuum white-light laser. The microscope was also equipped with an Acusto Optical Beam Splitter (AOBS), SuperZ-galvanometric stage and motorized xy-scanning stage, AFC hardware autofocus, OKO lab full-enclosure incubator with temperature, CO$_2$, humidity control, and the Leica LAS X software (version 4.6.0).

(iv) For live microscopy shown in Figs. 1A,E, 2B, 4A, 7L, EV1J, and Movies EV1, EV6–9 and EV11–13, we used Nikon Ti-E widefield microscope equipped with a full-enclosure atmospheric incubator (37 °C, 90% humidity and 5% CO$_2$); Lumencor Spectra X light engine with 6 solid-state light sources, including LED-DAPI-A (392/23 nm excitation), LED-FITC-A (474/27 nm excitation), TxRed-4040C (562/40 nm excitation), LED-Cy5-A (635/18 nm excitation); and a lamp source for phase contrast supplemented with DIC polarizer. The signal was recorded using Hamamatsu Orca Flash 4.0 V2 B&W camera and NIS elements software. See details of the used objectives below.

(v) For live microscopy of the dermal explants, shown in Fig. 1C, Appendix Fig. S1A–C and Movies EV2–5, we used Leica SP8 fully motorized upright confocal microscope equipped with 405, 488, 552, and 638 nm lasers; HC Fluotar L 25x (NA0.95) dipping objective; three PMT detectors; LAS-X software; and a full-enclosure atmospheric incubator (37 °C, 90% humidity and 5% $CO_2$).

## Live imaging and analyses of CCL21ΔC-mCherry exocytosis and EGFP-RAB6A vesicle trafficking

Because of the thin nature of LECs, we were able to use epifluorescence microscopy for recording exocytosis events and vesicle trafficking. This approach allowed short frame interval and longer recordings before bleaching of the sample. For live imaging of CCL21ΔC-mCherry exocytosis (Figs. 2B,C, 5C,D, 6A,B, 7A, EV1P and Movie EV7) and EGFP-RAB6A+ vesicle trafficking (Fig. 4A and Movie EV9), we used Nikon Ti-E widefield microscope (see above for details). Imaging of CCL21ΔC-mCherry exocytosis or EGFP-RAB6A+ vesicle trafficking was done using 100X Plan Apo VC (NA 1.4) oil objective. Live recordings were carried out with a frame interval of 400 ms for a total movie length of 1′30″–2′. Movies of CCL21ΔC-mCherry expressing LECs were captured using the 555 nm LED (20% intensity and 200 ms exposure) and of EGFP-RAB6 expressing LECs using the 470 nm LED (40% intensity and 80 ms exposure). VE-cadherin-647 signal was captured only in the first and last frames of the recordings (640 nm LED, 10% intensity, and 20 ms exposure).

To visualize the EGFP-RAB6 vesicle tracks (Fig. 4A), maximum projections (in time) of the recorded movies (Movie EV9) were carried out with ImageJ (v. 2.0.0-rc-69/1.52n). Similarly, CCL21ΔC-mCherry exocytosis events were manually analyzed by observing the sudden disappearance of CCL21ΔC-mCherry vesicles (Figs. 2B,C, 5C,D, 6A,B, 7A, EV1P, and Movie EV7). To do this, using ImageJ (v. 2.0.0-rc-69/1.52n), we drew a 200-pixel wide line, with the VE-cadherin-stained cell junction in the middle of this line. The thickness of the line corresponded to 14 μm, thus, providing 7 μm area on either side of the proper LEC cell junction midline. The number of disappearing vesicles, within this 7 μm distance of the cell junctions, for each chosen cell, were manually counted (Fig. 2B). Next, all the multicellular junctions, present in the same cell, were marked using the multi-point tool in ImageJ. The distance of each detected exocytosis event, from the nearest multicellular junction, was then measured by drawing straight lines along the length of bicellular junction and reaching to the center point of the multicellular junction. Data from two or three independent experiments were then plotted showing the total number of exocytosis events in all the analyzed cells as a function of distance from the nearest multicellular junction (Figs. 2C, 5D, 6B). Using this dataset, also, the total number of CCL21ΔC-mCherry exocytosis events upon treatment with siELKS1, siRAB6, or EGFP-RAB8A DN were determined and plotted using GraphPad Prism (v.9.2.0) (Figs. 5C, 6A, and 7A).

To investigate calcium dependency of the CCL21ΔC-mCherry exocytosis, on the day of live imaging (Day 4 after plating), the cells were incubated with 10 μm BAPTA-AM (Invitrogen, B6769) or the same volume of DMSO (control) and 1:1000 diluted

anti-VE-cadherin-647 non-blocking antibody in MV2 LEC culture media for 1 h. Then LECs were washed once with MV2, fresh MV2 media was added, and the cells were imaged immediately using Nikon Ti-E widefield microscope (see above for details).

## Live imaging analyses of EB3-mCherry positive microtubule + ends

LECs expressing EB3-mCherry were stained with VE-cadherin 647 non-blocking antibody as described above. Immediately after staining, EB3-mCherry live imaging was done using the Andor dragonfly spinning disc microscope and 100X objective (see above for details). The 561 nm laser was used for live imaging of the EB3-mCherry, and 634 nm laser was used to separately capture VE-cadherin signal as a single image. To avoid phototoxicity and allow frequent frame interval for EB3-mCherry imaging, we used a 2-μm-thick z-stack (5 slices, step size of 0.5 μm). Due to the thin nature of LECs, the stack covered the LEC entirely in z-direction, i.e., from basal to apical surface, in most part of the LEC, including the junctions. We used frame interval of 638 ms, which allowed tracking of the EB3+ microtubule ends and a total movie length of 1′–2′ before fluorophore bleaching.

For the analysis, maximum intensity projections of the movie Z-stack were made using ImageJ (v. 2.0.0-rc-69/1.52n). Next, the corresponding images showing the VE-cadherin-stained cell junctions were overlayed with the movie. On the merged movie, circular ROIs with a 10 μm diameter were placed in the middle of bicellular or at multicellular junctions. The EB3-mCherry comets were classified as (i) moving through or (ii) arresting at the circular ROI. The results (Fig. 4D,E) were tabulated as percentages of total number of comets and plotted using GraphPad Prism (v.9.2.0).

## Live imaging of the DC-LEC co-cultures

Prior to live imaging, the VE-cadherin junctions of LEC monolayers were stained as described above. To label the nuclei of wild-type and CCR7−/− DCs, the cells were collected by centrifugation at 180 RCF and then incubated with R10 culture medium, containing the NucBlue live cell stain (Invitrogen, R37605), for 20′ in the cell culture incubator (37 °C and 5% $CO_2$). The labeled DCs were centrifuged at 180 RCF for 6′, dissolved in MV2 media, and allowed to recover in the cell culture incubator (37 °C and 5% $CO_2$) for a few minutes.

To record the DC migration on the LEC monolayer (Figs. 1E–G and EV1B,J, Movie EV6) and the transmigration across the monolayer (Figs. 1A,B, 6K, 7L,M, and Movies EV1 and EV11–13), we used the 20X Plan Apo λ (NA 0.75) dry objective with Ph2 phase contrast. We, first, checked that the area displayed uniform CCL21-mCherry or CCL21ΔC-mCherry expression (TxRed-4040C, 562 nm excitation) and confluency (VE-cadherin channel; LED-Cy5-A, 635 nm excitation). For experimentation shown in Fig. 7L,M and Movie EV13, we also checked that a great majority of LECs expressed EGFP or EGFP-RAB8A DN fluorescence. Then, LPS-activated labeled DCs were loaded on top of the anti-VE-cadherin stained confluent LEC monolayer and the recording was started immediately with a frame interval of 1′30″ and total acquisition length of 90′. For fluorescence recording, LED-DAPI-A (392 nm excitation) and LED-Cy5-A (635 nm

excitation) channels were used. Also, LED-FITC-A (474 nm excitation) channel was used for experiments involving EGFP or EGFP-RAB8A DN. Phase contrast channel was simultaneously acquired, with 10% lamp power and 5 ms exposure.

We used the ImageJ (v. 2.0.0-rc-69/1.52n) to analyze the recorded movies for the DC transmigration events. For analysis of the transmigration site, we classified the transmigration events to take place at the bicellular or multicellular junctions. The number of events were then expressed as a percentage of the total number of events and depicted in (Figs. 1B and EV1D,O). To analyze the effect of lymphatic endothelial RAB6 silencing, or EGFP-RAB8A DN expression on transmigration, we calculated the efficiency of DC transmigration in 446 × 446 or 664 × 663 μm-sized images. Here, we first calculated the percentage of transmigrating DCs and then, normalized the result to the average of control samples, which was set at 1, in each experiment (Figs. 6K and 7M).

To analyze the site of wild-type or CCR7$^{-/-}$ arrest on the LEC monolayer (Fig. 1E,G) we, first, defined the arrest. Wild-type DCs stayed arrested at the transmigration site, on average, for 450″ (mean, $n = 43$) prior to transmigration. We used this 450″ time-interval as definition for arrested DCs. Also, the wild-type DCs that transmigrated already before the 450″ were counted as arrested. Then, we analyzed the location of the DC arrest on the LEC monolayer (Fig. 1G), we classified the locations into three types: at bicellular junction, at multicellular junction, or non-junctional. Only the locations where the quality of the VE-cadherin staining was good enough to classify the junctions, were counted in the analysis (Fig. 1G).

To track the migration of labeled (Hoechst) wild-type and CCR7$^{-/-}$ DCs on the LEC monolayer (Fig. 1E,F), we used the IMARIS tracking tool. To analyze only actively migrating DCs, we filtered all the tracks corresponding to DCs that were not migrating on the monolayer but, rather, were flown along the media convection. To study the straightness of the drawn tracks, the positional data (x, y) of each track at each timepoint was exported to Microsoft Excel (v 16.71). MATLAB (MatLab & Simulink, version r2021b, MathWorks) was used to calculate the length and displacement of each track. The straightness of the tracks was calculated by dividing the displacement (which is the straightest possible route to the stopping site), by the actual length of the track. Therefore value 1 represented the straightest possible track (Fig. 1F).

To compare the morphology of wild-type and CCR7$^{-/-}$ DCs (Fig. EV1J), after the LPS-mediated activation, we labeled the wild-type DCs with Deep Red Cell tracker (Invitrogen, C34565) and CCR7$^{-/-}$ DCs with Green cell tracker (Invitrogen, C7025), in serum-free R10 culture medium for 30′ in the cell culture incubator (37 °C and 5% $CO_2$). Next, we centrifuged the cells at 180 RCF for 6′, removed the supernatant, and resuspended the cell pellet in MV2 medium. The labeled wild-type and CCR7$^{-/-}$ DCs were pooled and loaded onto a non-transduced and unstained LEC monolayer. The images were captured with Plan Fluor 40X objective (NA 0.75) with Ph2 phase contrast from the areas that displayed wild-type and CCR7$^{-/-}$ DCs.

To investigate if CCL21ΔC-mCherry exocytosis is increased by DC contact (Movie EV8), first, exocytosis events at the multicellular junction, prior to DC addition, were captured as described above. Next, activated and labeled DCs were loaded onto the LEC

monolayer, and then, we imaged areas where DC was located on top of multicellular junctions.

## Live imaging and analysis of the transmigration events in dermal explants

For live microscopy of mouse dermal explants (Fig. 1C; Appendix Fig. S1A–C and Movies EV2–5), we modified the earlier protocol (Weber et al, 2013). In short: Mice were sacrificed, and the ear pinna were cut off. The ear pinna sheets were separated, and the dorsal sheets were stained with anti-mouse CD31-FITC antibody (Thermo Scientific, clone 390, 11-0311-82, 1:100 dilution) in R10 media for 45 min. After washing the ear pinna sheets 3 times with R10, the sheets were mounted onto a cell culture dish (exposed dermis up) by attaching a washer on top of the explant by paraffin. The activated DCs (see above), which had been labeled with Deep Red Cell Tracker (Invitrogen, C34565) according to the manufacturer's protocol, were loaded on the exposed dermis. After 15 min, the mounted ear pinna sheet was washed, the plate was filled with R10, and placed on a Leica SP8 upright confocal microscope stage holder (see details above). For imaging, we selected locations, which showed pre-collector-like phenotype and strong-enough α-CD31-FITC signal. For the benefit of the α-CD31-FITC and DEEP Red Cell Tracker intensity, and to avoid bleaching, we used pinhole of 2, step size of 1.5 μm, zoom-factor 3, and resolution of 512 × 512 px/inch.

In three independent experiments and altogether 6 ear sheet samples, representing 6 mice, we captured, altogether, 33 transmigration events, where we were able to determine the location of the transmigration site. By using FiJi image analysis software, we first classified the transmigration events as "at the multicellular junction" or "at bicellular junction". Then, regarding the latter, we measured the distance of the transmigration site, along the junction, to the closest multicellular junction (Fig. 1D).

## Colocalization analysis of cell culture data

To quantify the colocalization of CCL21ΔC-mCherry with EGFP-RAB-GTPases we used Imaris (v9.5.0, Bitplane, Oxford instruments). The vesicles expressing CCL21ΔC-mCherry and EGFP-RAB-GTPase, were automatedly detected with the spot detection tool. The detection diameter of the smallest vesicle was set at 0.2 μm. Next, the spots corresponding to the Golgi area, were cleared by manual deletion. To do this, the brightness intensity of CCL21ΔC-mCherry channel was first set to minimum level at which most of the vesicles near the Golgi region were clearly seen. At this level of brightness, the continuous, non-vesicular structure of the Golgi membranes was visible. Colocalization between the detected EGFP- and mCherry-expressing spots was determined automatically for spots closer than 0.4 μm, using the colocalization tool in IMARIS. The colocalization was presented as percentage of CCL21ΔC-mCherry spots that colocalized with EGFP-RAB-GTPase spots in the whole LEC area (Figs. 2E, EV2D–F and EV3C,D). Since EGFP-RAB8A also localized to the plasma membrane, signal intensity of EGFP-RAB8A vesicle-like signal did not stand-out (Fig. EV2H,I). Therefore, hampering the detection with IMARIS automated spot detection tool. To address this, we manually added the spots to mark the EGFP-RAB8A dim vesicle-like signals that colocalized with CCL21ΔC-mCherry. Next,

colocalization tool in IMARIS was used to determine the number of CCL21ΔC-mCherry vesicles which colocalized with EGFP-RAB8A (as described above). A comparison of automatic and manual detection of the colocalization result of EGFP-RAB8A and CCL21ΔC-mCherry is shown in Fig. EV2J,K.

To measure the distance of colocalized CCL21ΔC-mCherry or CCL21-mCherry vesicles from the nearest multicellular junction (Figs. 2F, EV2G, EV2K and EV3E) we, first, marked the multicellular junctions using the surface tool in IMARIS and then ran distance transformation protocol. The results were exported and plotted, using GraphPad Prism (v.9.2.0) and are shown as a percentage of vesicles found within a binned distance of 5μm from the nearest multicellular junction. The analysis method of the colocalization area (Fig. 5J) is described below.

## Colocalization analysis of in vivo data

To quantify the colocalization percentage of endogenous CCL21 and RAB6 in mouse ear pinna dermal lymphatic vessel endothelium, we, first, used a multipoint tool to mark RAB6 colocalizing or non-colocalizing CCL21 vesicles as separate sets in ImageJ (v. 2.0.0-rc-69/1.52n). Then, the multicellular junctions or discontinuous VE-cadherin junctions were marked with a freehand line tool or multipoint tool, respectively. Then two scripts (see below) were employed to draw lines from each marked vesicle to the nearest multicellular junction or discontinuous VE-cadherin junction. The length of the lines was then measured in ImageJ. The results were exported and plotted, using GraphPad Prism (v.9.2.0) and are shown as a percentage of colocalized and non-colocalized CCL21 vesicles in the whole LEC area (Fig. 2M,O) and as a function of distance from multicellular junctions within a binned distance of 5 μm (continuous junctions, Fig. 2N) or 2 μm (discontinuous junctions, Fig. 2P). The scripts used for this analysis are written by Ved Sharma and are available at the following URL: https://github.com/ved-sharma/Shortest_distance_between_objects (accessed on 31.01.2024 at 1100 h).

## Transmission electron microscopy

For electron microscopy of immuno-gold labeled samples (Fig. 3A–C; and Appendix S3A,B), LECs were cultured on 0.1% gelatin-coated 13 mm (No.1) glass coverslips. On day 1, LECs were transduced with CCL21-mCherry, CCL21ΔC-mCherry, EGFP-RAB6, or EGFP-RAB27A encoding lentiviruses and cultured as described above. On the day 3, the cultures were fixed with PLP fixative containing 2% PFA (paraformaldehyde), 0.01 M sodium periodate, in 0.1 M lysine HCl-100mM sodium phosphate buffer-pH 7.4 (protocol provided by Electron microscopy Unit, University of Helsinki). Next, the cells were washed three times for 5′ each with 0.1 M sodium phosphate $NaPO_4$ buffer (pH 7.4) and permeabilized with buffer A (0.01% saponin in 0.1% BSA solution in sodium phosphate buffer (100 mM, pH 7.4)) for 8′. Next, for the detection of CCL21-mCherry, CCL21ΔC-mCherry or EGFP-RAB-GTPases, samples were incubated in a moist chamber for 2 h at room temperature with primary antibodies; rabbit anti-RFP (Rockland, 600-401-379) or rabbit anti-EGFP (Abcam, ab290), diluted 1:500 in permeabilization buffer. To control for the antibody staining, non-transduced LECs were incubated in a similar way. The samples were washed three times (for 5′ each)

with buffer A. After washing, the cells were incubated in a moist chamber at room temperature for 1 h with anti-rabbit nanogold particles (diluted 1:50 in buffer A). Next the cells were washed 3 times (for 5′ each) with buffer A and 3 times (for 5′ each) with sodium phosphate buffer. Post-fixation of samples was done with 1% glutaraldehyde (GA) in 100 mM sodium phosphate buffer. Samples were then incubated in 50 mM glycine solution in sodium phosphate buffer for 5′, for the quenching of free aldehyde groups, washed with sodium phosphate buffer and, finally, left in water overnight at 4 °C. Nanogold particles were amplified with silver using the HQ Silver™ Enhancement kit (Nanoprobes Inc.), in a dark room, for 4–5′ at room temperature using manufacturer's protocol. The samples were then washed with distilled water and subjected to gold toning; First the samples were incubated with 2% sodium acetate for 5′ each (3 times in total) at room temperature. Next, they were incubated in 0.05% tetra chloroauric acid for 10′ on ice and finally for 10′ (two times each) in 0.3% sodium thiosulphate. After gold toning, the samples were preserved in water. The following steps were performed at the electron microscopy unit. Samples were first treated with 1% osmium tetroxide in sodium cacodylate buffer (NaCac, pH 7.4) supplemented with 15 mg/ml tetra potassium ferrocyanide (60′ at room temperature in the dark). Next, the samples were dehydrated subsequently with 70%, 96% and absolute ethanol and then soaked in acetone on aluminum plates. The samples were covered with drops of epon with a beam capsule placed on top of the epon drops. Next, the samples were incubated at room temperature for 2 h and baked at 60 °C for 14 h. Finally, samples were dropped in liquid nitrogen to remove the coverslips. Serial ultrathin sections (60 nm) were cut with an ultramicrotome (ultracut UCT7, Leica Biosystems) and collected on pioloform-coated copper single slot grids. The samples were observed with Jeol JEM-1400 operated at 80 kV and equipped with Gatan Orius SC 1000B bottom-mounted CCD-camera at Electron microscopy unit (EMBI), University of Helsinki.

## ELKS intensity analysis

For the quantification of ELKS at LEC junctions (Figs. 5A,B and EV5A–C), ImageJ (v. 2.0.0-rc-69/1.52n) was used. In the first step, we created average projections of the images using the z-project tool. Next, based on the VE-cadherin staining, the multicellular junctions were marked with dots using the overlay brush tool. Then, using the freehand line tool, a 60-pixel wide line (corresponding to 6 μm) was drawn along the midline of the VE-cadherin-stained cell junctions, starting from one multicellular junction and ending in the next multicellular junction. The width of the drawn line covered the cell junction and extended approximately 3 μm into the LECs on both sides. To obtain the ELKS intensity data along these drawn lines, the multi-plot tool was used to get intensity profiles for each line in one image. The intensity profiles consisted of ELKS intensity values plotted against the respective distance values, so that each intensity value represented the average intensity for each 60-pixel wide row in a drawn line. The first and last intensity values were defined as the intensity at the multicellular junction, and the midpoint value as the point furthest away from the multicellular junction. Mean intensity at each distance for all lines within one image, and then within one experiment, were calculated. The mean intensity values for each experiment at distance 0 μm (i.e., at multicellular junction) was

then set at 1. Finally, the combined mean and standard deviation for all three experiments were calculated and a graph (Fig. 5B) was made using Prism GraphPad (v.9.2.0). Similarly, changes in ELKS intensity were measured upon treatment with PI3K inhibitor (PI-103). For that analysis, the mean of controls in each of 5 experiments at distance 0 μm was set as 1 (Fig. EV5B,C) and the data was plotted using Prism GraphPad (v.9.2.0).

## CCL21ΔC-mCherry and CCL2 intensity analysis

The CCL21ΔC-mCherry and CCL2 levels in the whole LECs were analyzed with ImageJ (v. 2.0.0-rc-69/1.52n) as follows: We, first, created average projections of 205.80 μm × 205.80 μm sized spinning disc confocal microscopy images. Next, for siRNA-treated and CCL2-stained samples, we measured the average intensity per pixel for the whole image as endogenous CCL2 was expressed in all the LECs upon TNF-α treatment (Fig. 6J). Conversely, in LECs treated with siRNA and transduced with CCL21ΔC-mCherry, only the cells showing the mCherry signal were segmented using the polygon tool in ImageJ. The total intensity of the CCL21ΔC-mCherry signal was then measured in cells expressing CCL21ΔC-mCherry and similarly in non-transduced cells (i.e., the background). Next, the background was subtracted, and the resulting values are reported in the quantifications shown in (Figs. 5H and 6F). Similarly, in experiments studying the effect of EGFP-RAB8A DN on CCL21ΔC-mCherry (Fig. 7F) and CCL2 intensity (Fig. 7K), only the cells expressing both EGFP, EGFP-RAB8A WT, or EGFP-RAB8A DN and CCL21ΔC-mCherry, or endogenous CCL2 (which was expressed in all the cells) were segmented. The average intensity per pixel of CCL21ΔC-mCherry or CCL2 signal was measured in each segmented LEC using the intensity measurement tool in ImageJ. Then, background signal was measured in non-CCL21ΔC-mCherry transduced cells or cells, where CCL2 primary antibody was excluded from the staining procedure. The background signal was subtracted from the measured CCL21ΔC-mCherry or CCL2 values.

For analyses of siELKS, siRAB6, or EGFP-RAB8-T22N, effect on CCL21ΔC-mCherry and CCL2 intensity at the multicellular junctions (Figs. 5F,G, 6D,E, 6H,I, 7D,E and 7I,J), 5 LECs/sample, showing highest chemokine expression in the whole cell analysis, were selected. Next, for each LEC, multicellular corners were marked by using the spot tool and distance transformation tool was used to create a distance map in regard of the marked spots (in IMARIS v9.8.2). The distance transformed channel was saved as a separate file. In ImageJ (v. 2.0.0-rc-69/1.52n), a minimum projection of the distance map was then created, and thresholding tool was used to show pixels only up to 5 μm from the center of the spot, thus, resulting in 5 μm radius circles. Center points of the circles represented the multicellular junctions. Next, a mask was created from the thresholded minimum projection. The created masks were added to the average projection of the original image, using the image calculator tool, to confirm that the spot represents the multicellular junction. Then we either (i), for the measurement of CCL2 intensity at multicellular junctions in siRNA-treated samples, used the masks to create ROIs, which were used to measure the mean intensity of CCL2 from the optical slice of the hyperstack, showing the greatest chemokine signal. The quantification method is also shown in Appendix Fig. S4B. Whereas (ii), in case of transduced LECs, the cell junctions of the selected cells (see above) were drawn by using the polygon tool and saved as ROIs. These ROIs were used to segment the cells and to create a new

mask. The cell masks were then multiplied with the multicellular junction masks, resulting in a new mask showing only the multicellular corners (in the form of pieces of a pie with a radius of 5 μm—as shown in Appendix Fig. S4A). Finally, these masks were used to create ROIs, which were in turn used to measure the mean intensity of CCL21ΔC-mCherry or CCL2 from the optical slice of the hyperstack showing the brightest chemokine signal. Background signal was subtracted from the measured values in (i) and (ii).

## Analysis of colocalization area at multicellular junctions

To measure the colocalization of the CCL21ΔC-mCherry with endogenous RAB6 (Fig. 5I,J) we measured the colocalization area within a 5 μm distance from the multicellular junction vs. the rest of the LEC. For this we first segmented these areas as follows: First, all steps from the CCL21ΔC-mCherry intensity in the multicellular junction analysis (as explained in the section above) were conducted on the image, until we possessed (1) a mask and ROIs representing only the area within 5 μm from multicellular junctions, and (2) a mask and ROI representing the segmented cell of interest. After this, the mask (1) was reduced from mask (2), using XOR-function from the image calculator, to create a new mask and ROI representing the LEC excluding the areas within 5 μm from multicellular junctions.

Next, to exclude the nucleus and Golgi from the quantified areas, we used the average projection of the original image to segment the nucleus based on DAPI staining and the Golgi apparatus based on RAB6 depot close to the nucleus. Segmentation was done using the polygonal tool in ImageJ and the resulting ROIs were saved.

Next, to segment the RAB6 and CCL21ΔC-mCherry colocalization areas, the corresponding channels were thresholded separately. As RAB6 was endogenously stained, the thresholding limits for RAB6-channel were kept constant for all images throughout each individual experiment, whereas CCL21ΔC-mCherry limits varied between images/cells due to differences in CCL21ΔC-mCherry overexpression. The thresholds were masked, and the resulting CCL21ΔC-mCherry and RAB6 masks were multiplied to create a mask, showing only colocalizing CCL21ΔC-mCherry and RAB6 vesicles (done separately for each 5 μm multicellular junction area and the rest of the cell area). Then, the colocalization and total areas were measured. All results were exported to Microsoft Excel and colocalization percentages per area were calculated.

## Flow cytometry analysis

To analyze the DC marker expression and size (Fig. EV1E–I), we flushed wild-type or CCR7[−/−] DCs from the cell culture plates, centrifuged the cells for 6′ at 300 RCF, resuspended the cells in 1:1 mix of R10 media and 4% PFA. The samples were fixed on ice for 15′, washed twice with PBS, washed once with FACS buffer (PBS + 2 mM EDTA), and then blocked 10′ at room temperature in FACS buffer + 1% BSA + 1:100 diluted Rat α-mouse CD16/32 (BD Pharmingen Cat #: 553142). The α-CD11b-PE (M1/70, BioLegend, 101208, diluted 1:400), α-CD11c-PE-Cy7 (HL3, BD Pharmingen, 121421, 1:300), α-CD86-FITC (GL1, BD Pharmingen, 561962, 1:300), and α-MHCII-eFluo450 (MS/114.15.2, eBioscience, 48-5321-82, 1:800) were diluted in blocking buffer and incubated on the cells for 1 h at room temperature, followed by a wash with

FACS buffer and a resuspension to 300 µl of FACS buffer. As a negative control, we used non-stained DCs. For compensation, we used compensation beads (Biolegend, 424602) that were incubated with individual antibodies. At least 277,000 single cells/sample were analyzed using the Novocyte Quanteon Flow Cytometer and Novoexpress software.

## Western blot analysis

For Western blot analysis analysis (Fig. EV1K), DCs were lyzed with SDS boiling buffer (2.5% SDS, 0.25 M Tris-HCl, 50 mM NaF, 10 mM glycerol-2-phosphate, 0.5 mM DTT, 0.5 mM PMSF, 2.5 µg/ml aprotinin and 1 µg/ml leupeptin), needled (25-gauge) and centrifuged at 18,000 RCF for 10 min at 4 °C. Protein concentrations of the supernatants were measured with Pierce™ BCA Protein Assay Kit (Thermo Fisher Scientific, 23227). 12 µg of protein were used per sample, run in SDS-PAGE gel, and blotted using the wet transfer method. The membranes were, first, blocked with TBS including 0.05% Tween 20 and 5% BSA, and, then, probed overnight at 4 °C with the following primary antibodies diluted in the blocking buffer: α-b-actin (clone AC-15, Sigma, A1978, 1:2500), α-non-muscle heavy chain IIA (Poly19098, Biolegend, 909801, 1:1000), α-HSC70 (clone B6, Santa Cruz Biotechnology, sc-7298, 1:2000). The membranes were washed three times with TBS-T, incubated with the secondary antibodies (IRDye 680RD anti-mouse (1:5000, LI-COR) and IRDye 800CW anti-rabbit (1:5000, LI-COR)), diluted in the blocking buffer, for 2 h at room temperature, and washed two times with TBS-T and once with TBS. The blots were imaged on Azure 500 imaging system (Azure biosystems) and analyzed with AzureSpot Pro software (Azure biosystems, 1.9.0.0406).

## Quantitative PCR

For analyses of siRNA-mediated mRNA silencing (Figs. EV1M and EV5E,F), on day 0, LECs were plated on 48-well plates (Thermo Scientific, 150687), coated with 0.1% gelatin for 60′ in a cell-culture incubator (37 °C and 5% CO$_2$). On day 1, siRNA transfection was done using the individual or smart pool oligos (Dharmacon™/Horizon; see the section on transfection protocol above). On day 3, 48 h subsequent to the start of transfection, the LECs were detached using Trypsin 0.25% EDTA (Gibco, 25200056) for 1–2′ at 37 °C. Trypsinization was stopped by adding fresh MV2 media and the cells were centrifuged at 180 RCF for 6′. The cells were immediately subjected to lysis and RNA extraction using RNeasy RNA extraction kit using manufacturer's instructions (Qiagen, 74104). The concentration of extracted RNA was measured using a NanoDrop™ spectrophotometer (NanoDrop-ND-100), and the RNA was stored at −80 °C. Next, the extracted RNA, from all samples in one experiment, was diluted to the same concentration using RNA-free water and was converted to cDNA using the high-capacity cDNA reverse transcription kit and manufacturer's protocol (Applied Biosystems, Cat: 4368814).

The qPCR primers were designed with Primer3web (version 4.1.0) (available at https://primer3.ut.ee). The following qPCR primers were then ordered from Sigma-Aldrich/Merck, Germany.

HPRT: ATTATGGACAGGACTGAACGTCTTG and TGTAA TCCAGCAGGTCAGCAAAG. RAB6A: AGGAGGGAGAGAGGA AAGCC and CTTTCCATTCCCGGCAAAGC. ERC1 (ELKS): AT

GGCTGAAGAGAAGGGGAC and GTGTCAGTGTTGGTGGT GTC. CCL21: TCCTTATCCTGGTTCTGGCC and GCAAGAA-CAGGATAGCTGGG. The qPCR reactions were set-up in a 96-well skirted PCR plate (4titude, 4ti-0740) with a final volume of 20 µl/well constituting of 4 µl cDNA and 16 µl of master-mix (1 µl of 10 µM primer pair mix of forward and reverse qPCR primers), 5 µl of water (Lonza, 11548015), and 10 µl FastStart universal SYBR green master (Sigma/Merck, 4913850001). The sealed plate (IntelliXseal™ SA, Azenta life science) was centrifuged at 1000 rpm for 2′ at 4 °C and loaded to the qPCR machine (BioRad, C-1000 touch, CFx96 real-time qPCR). The qPCR program was as follows:

| Program | Temp | Time | Cycle |
|---------|------|------|-------|
| Step 1 | 95 °C | 3′ | 1 |
| Step 2 | 95 °C | 15″ | 39 |
| Step 3 | 60 °C | 1′ | |
| Step 4 | To step 2 | To step 2 | |
| Step 5 | 65 °C | 5″ | 1 |

First, we calculated the average of Cq values of technical duplicates. Then, for each primer pair, the resulting Cq value of siCCL21, siRAB6 or siELKS samples was substracted by Cq value of siControl samples to yield ΔCt value. Next, ΔΔCt values were calculated by subtracting the ΔCt of RAB6, ELKS or CCL21 primer pairs by ΔCt of HPRT primer pair. Then, $2^{-\Delta\Delta Ct}$ was calculated by calculating the square of the inversed ΔΔCt values. The $2^{-\Delta\Delta Ct}$ values were finally multiplied by 100 to give % expression of each mRNA and plotted using Prism GraphPad (v.9.2.0).

## Measurement of mCherry signal in the culture supernatant

Primary LECs were plated on day 0, on 0.1% gelatin-coated 48-well cell culture plates (Nunc, Thermo, 150687) in full MV2 media without GA. On day 1, LECs were co-transduced with lentivirus encoding CCL21ΔC-mCherry and EGFP (control) or EGFP-RAB8A DN. Full MV2 media was replaced daily and, finally, collected for analysis on day 4 (i.e., 72 h after transduction and 24 h subsequent to last media change). Collected supernatant was centrifuged at 300 RCF for 6′ and 200 µl of the supernatant was transferred to a black 96-well plate (Perkin Elmer, 6005270). Fluorescence was measured with EnSight multimode plate reader (Perkin Elmer, excitation 587 nm and emission 610 nm) and Kaleido software version 3.0. Background fluorescence intensity was measured from the supernatant collected from non-transduced LECs. To analyze the CCL21ΔC-mCherry fluorescence intensity, we first reduced the background fluorescence intensity and then, normalized the remaining intensity to the average of control samples, which was set at 1, in each experiment. The results were tabulated using GraphPad Prism (v.9.2.0) (Fig. 7B).

## Study design

We determined sample sizes based on the previous experimental observations and similar published experiments, in which statistically significant differences were observed. For data collection,

analysis, and quantifications no blinding was done because the collected data was quantitative and not influenced by the investigator's bias. Further, most of the analysis/quantifications were done in a semi-automated or automated manner.

## Statistics

Prism GraphPad software (v.9.2.0) was used for visualization and statistical analysis of the data. We tested the normality of the data with Shapiro–Wilk test, except for Fig. 7I, for which we used Kolmogorov–Smirnov test. Unpaired two-tailed Student's t-test with Welch's correction was used for normally distributed data. For non-normal data, Mann–Whitney U test was used. For categorical data Chi-squared test or, if the sample size was small, Fisher's exact test was used. For straightness of the migration tracks, the data was binned with 0.1 intervals and analyzed with Chi-squared test. The grouped data in Fig. 2E was analyzed using ordinary ANOVA test. The corresponding statistical tests and sample sizes ($n$) are indicated in the figure legends. The $p$-values are shown in figures.

## Graphics

Graphics for the synopsis image were created with BioRender.com.

# Data availability

Source data accompanies this article. No large data sets were produced in this study. This study includes no data deposited in external repositories.

The source data of this paper are collected in the following database record: biostudies:S-SCDT-10_1038-S44318-024-00129-x.

# Peer review information

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

## Acknowledgements

We hereby acknowledge University of Helsinki core facilities: Genome Biology Unit (GBU) Electron Microscopy Unit (EMBI), Helsinki Bioimaging (HBI) research infrastructure platform, Biomedicum Imaging Unit (BIU), Light Microscopy Unit (LMU), HiLife Flow Cytometry Unit, and Laboratory Animal Center. We also acknowledge Seppo Kaijalainen for plasmid constructs and insight on cloning techniques, and Dr. Michael Sixt for critical reading of the manuscript. This work was supported by Sigrid Juselius Foundation young group leader grant (KV), Academy of Finland Research Fellow grant (KV, #315710), Research Council of Finland Project Grant (KV, #356849), Wihuri Reasearch Institute (KV), and Doctoral Program in Integrative Life Science (IL). Open access funded by Helsinki University Library and Wihuri Research Institute.

## Author contributions

**Inam Liaqat**: Formal analysis; Funding acquisition; Validation; Investigation; Methodology; Writing—original draft; Writing—review and editing. **Ida Hilska**: Formal analysis; Investigation. **Maria Saario**: Formal analysis; Investigation. **Emma Jakobsson**: Formal analysis. **Marko Crivaro**: Formal analysis. **Johan Peränen**: Resources. **Kari Vaahtomeri**: Conceptualization; Resources; Formal analysis; Supervision; Funding acquisition; Validation; Investigation; Methodology; Writing—original draft; Project administration; Writing—review and editing.

Source data underlying figure panels in this paper may have individual authorship assigned. Where available, figure panel/source data authorship is listed in the following database record: biostudies:S-SCDT-10_1038-S44318-024-00129-x.

## Disclosure and competing interests statement

The authors declare no competing interests.

# Expanded View Figures

**Figure EV1.** (i) Anchoring incapable CCL21ΔC-mCherry supports DC transmigration at multicellular junctions and (ii) CCR7$^{-/-}$ are phenotypically normal.

(A) VE-cadherin (red) and LYVE1 (green) staining of mouse ear pinna dermis. A flattened overview image and zoom-in of a single optical slice are shown. The images represent $n = 3$ mice. (B) The dot plot shows mean ± SD percentage of wild-type or CCR7$^{-/-}$ DCs that detached, subsequent to the arrest at multicellular junction. The data point represents $n = 6$ wild-type and $n = 5$ CCR7$^{-/-}$ biological replicates from three independent experiments, altogether, representing 401 wild-type and 140 CCR7$^{-/-}$ DCs. The data is related to Fig. 1E–G. (C) Schematic shows mCherry tagged full-length CCL21 and CCL21ΔC-mCherry that lacks the charged C-terminus (Hirose et al, 2002). (D) Quantification of wild-type DC transmigration sites in CCL21ΔC-mCherry expressing LEC cultures. The stacked bar graph shows transmigration sites as a percentage of all events from 15 biological replicates in three independent experiments and, altogether, $n = 128$ transmigration events. (E–I) The bar graphs show the mean percentage ± SD of DCs positive for (E) CD86, (F) CD11b, (G) CD11c, and (H) MHCII in wild-type versus CCR7$^{-/-}$ DCs. In (I) the bar graph shows mean cell size ± SD of wild-type vs CCR7$^{-/-}$ DCs normalized to the average of wild-type DCs, which was set at 1, in each experiment. In (E, I) The data points represent $n = 4$ biological replicates/genotype in two independent experiments. The number of analyzed DC singlets was at least 277000/sample. (J) A capture of a phase contrast/immunofluorescence microscopy showing wild-type (magenta) and CCR7$^{-/-}$ DC (green), on a LEC monolayer. The images represent $n = 3$ biological replicates. (K, L) Western blot panel of non-muscle myosin heavy chain 2 A (NMH-IIA), actin, and HSC70 (for loading control) in wild-type versus CCR7$^{-/-}$ DCs. Quantification of the band intensities is shown in the western blot data shown in (L). The bar graph in (L) shows, mean intensity ± SD. Data was normalized to the average of wild-type DCs, which was set at 1. The data in (K, L) represent $n = 4$ biological replicates/genotype and two independent experiments. (M) Quantification of human CCL21 mRNA levels in human specific siCCL21 transfected and mouse CCL21ΔC-mCherry expressing LECs. The dot plot shows the mean hCCL21 mRNA level ± SD normalized to the average of siControl samples, which was set at 1 (green line), in each experiment. Data points represent $n = 4$ biological replicates/ siRNA oligo in 2 independent experiments. *P*-values show the comparison to controls. (N) Images show mouse CCL21ΔC-mCherry expression in siControl or human CCL21-specific siCCL21 oligo transfection. Images represent $n = 4$ biological replicates in 2 independent experiments. (O) Quantification of transmigration sites of wild-type DCs in siControl or siCCL21 transfected and CCL21-mCherry or CCL21ΔC-mCherry expressing LEC cultures. Transmigration sites are shown as a percentage of all events from 2 independent experiments, consisting of siCTRL01 $n = 209$ transmigration events (3 biological replicates), siCTRL02 $n = 187$ (3 biological replicates), siCCL21-05 $n = 137$ (3 biological replicates), or siCCL21-08 $n = 249$ (4 biological replicates) transfected and CCL21-mCherry expressing LEC cultures and of siCTRL01 $n = 96$ (4 biological replicates), siCTRL02 $n = 68$ (4 biological replicates), siCCL21-05 $n = 69$ (4 biological replicates), or siCCL21-08 $n = 223$ (8 biological replicates) transfected and CCL21ΔC-mCherry expressing LEC cultures. (P) The bar graph shows a mean number of observed CCL21ΔC-mCherry exocytosis events/cell ± SD in control or BAPTA-AM treated LECs. The data points represent $n = 37$ DMSO control and $n = 33$ BAPTA-AM treated cells in 11 biological replicates across two independent experiments. Data information: In (A), yellow arrows indicate the site of zoom-in image and white arrowheads indicate the multicellular junctions. In (J), the white arrowheads indicate wild-type and yellow arrows CCR7$^{-/-}$ DC dendrites. The *p*-values in (B), (E–I), and (M) were calculated using a parametric T-test with Welch's correction, whereas in (L) and (P) *p*-values were calculated using the Mann–Whitney test. In (O), Chi-square test was used to test the significance of data. The scale bar is 20 μm in overview images (A and J), 5 μm in zoom-in images (A), and 50 μm in (N).

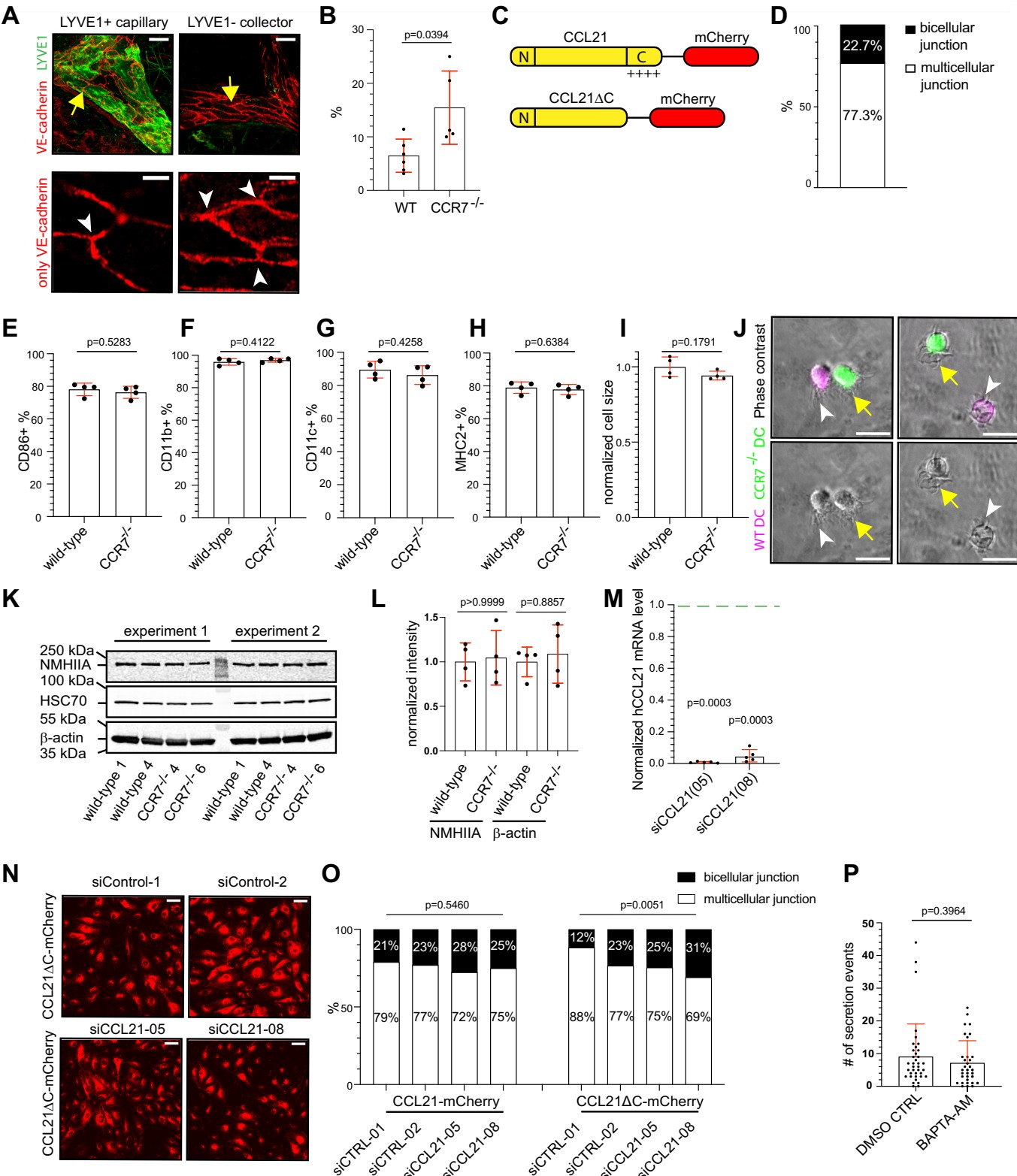

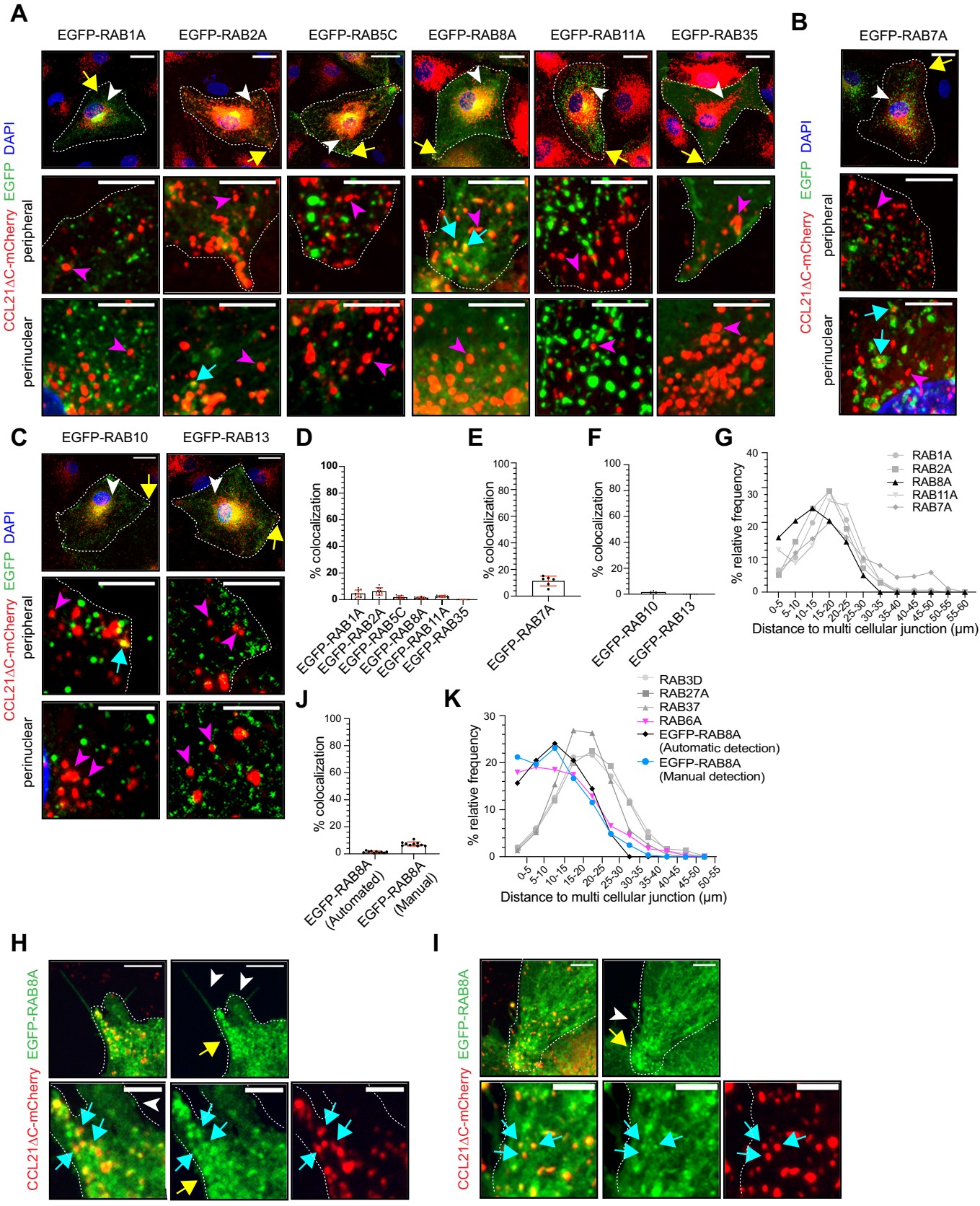

◀  **Figure EV2.  Colocalization of CCL21ΔC-mCherry with EGFP-RAB GTPases.**

(**A–G**) Immunofluorescence images in (**A–C**) show LECs expressing chemokine CCL21ΔC-mCherry (red) and the indicated EGFP-tagged RAB-GTPase (green). The nuclei were stained with DAPI (blue). In (**A**) images represent $n = 3$ biological replicates from 3 independent experiments and in (**B**, **C**) $n = 2$ biological replicates in 2 experiments. (**D–G**) Quantification of CCL21ΔC-mCherry+ vesicle colocalization with the indicated EGFP-RAB GTPases in the whole LEC area. The dot plots in (**D–F**), show the mean percentage ± SD. Each data point represents a single analyzed cell. In (**A** and **D**) $n = 11$ in EGFP-RAB1A, $n = 10$ in EGFP-RAB2A, $n = 9$ in EGFP-RAB5C, $n = 10$ in EGFP-RAB8A, $n = 9$ in EGFP-RAB11A, $n = 9$ in EGFP-RAB35, all representing three independent experiments. In (**B**, **C**, **E**, and **F**) $n = 6$ in EGFP-RAB7, EGFP-RAB10, and EGFP-RAB13 all representing two independent experiments. (**G**) The histogram shows the distribution (mean percentage) of CCL21ΔC-mCherry and the indicated EGFP-RAB-GTPase colocalized vesicles as a function of distance from a multicellular junction. The number of cells and independent experiments is the same as in (**A–E**). (**H–K**) Immunofluorescence images of LECs expressing EGFP-RAB8A and CCL21ΔC-mCherry. Quantification of colocalization in the whole LEC area is shown in (**J**). The dot plot shows the mean percentage ± SD. Each data point represents a single analyzed cell. $n = 10$ cells, from three independent experiments. (**K**) The histogram shows the distribution (mean percentage) of CCL21ΔC-mCherry and EGFP-RAB8A colocalized vesicles, as a function of distance from a multicellular junction and is projected on the histogram shown in Fig. 2F. The number of samples represented is the same as in (**J**). Data information: In (**A–C**), yellow arrows and white arrowheads indicate the site of the peripheral and perinuclear areas, respectively, shown in the zoom-in images. In the zoom-in images, cyan arrows indicate the colocalizing, and the magenta arrowheads examples of the more abundant non-colocalizing CCL21ΔC-mCherry+ vesicles. In (**H**, **I**), white arrowheads indicate plasma membrane localization of the EGFP-RAB8A and yellow arrows high-intensity areas in the vicinity of multicellular junctions. The cyan arrows highlight examples of colocalizing vesicles that were missed in the semi-automated analysis but included in the manual detection. In (**A–C**) and (**H**, **I**) cell borders are indicated with white dotted lines. Scale bars are 20 µm in overview images; 5 µm in zoom-in images in (**A**, **C** and **H**, **I**), and 3 µm in (**B**).

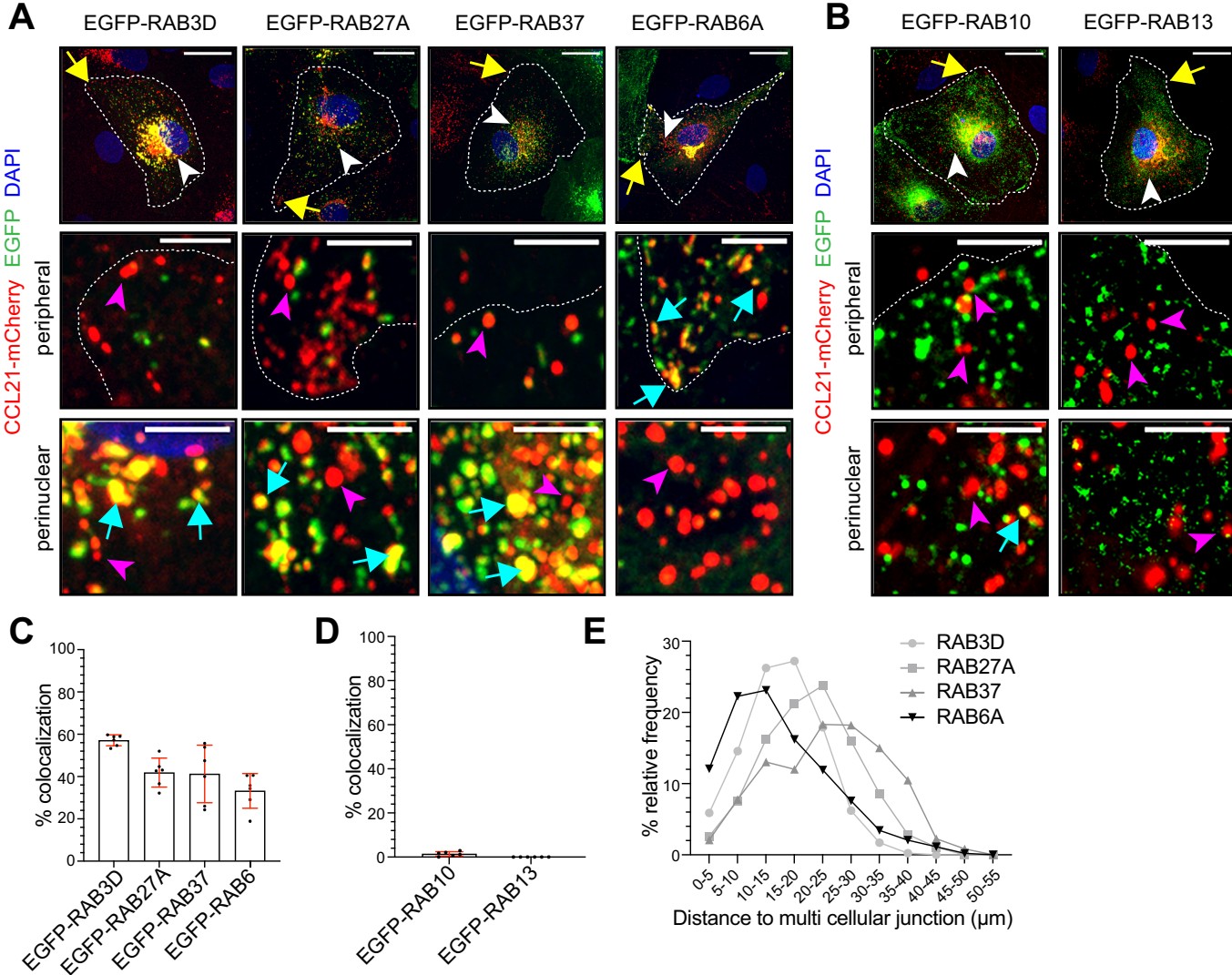

**Figure EV3. Colocalization of CCL21-mCherry with EGFP-RAB GTPases.**

(A–E) Shows colocalization of full-length CCL21-mCherry (red) with the indicated EGFP-tagged RAB-GTPase (green). The nuclei were stained with DAPI (blue). (C, D) Quantification of CCL21-mCherry+ vesicle colocalization with the indicated EGFP-RAB GTPases in the whole LEC area. The dot plots show the mean percentage ± SD. Each data point represents a single analyzed cell with $n = 6$ cells for each of EGFP-RAB3D, EGFP-RAB27A, EGFP-RAB37, EGFP-RAB6, EGFP-RAB10, and EGFP-RAB13 representing two independent experiments. (E) The histogram shows the distribution (mean percentage) of CCL21ΔC-mCherry and the indicated EGFP-RAB-GTPase colocalized vesicles as a function of distance from a multicellular junction. The number of cells and independent experiments is the same as in (C). Data information: In (A, B), the perinuclear and peripheral areas (shown in the zoom-in images below the overviews) are indicated with white arrowheads and yellow arrows, respectively. Colocalization is indicated with cyan arrows and non-colocalizing CCL21-mCherry vesicles are indicated with magenta arrowheads. The cell borders are indicated with white dotted lines. Scale bars are 20 μm in the overview images; 3 μm in zoom-in images.

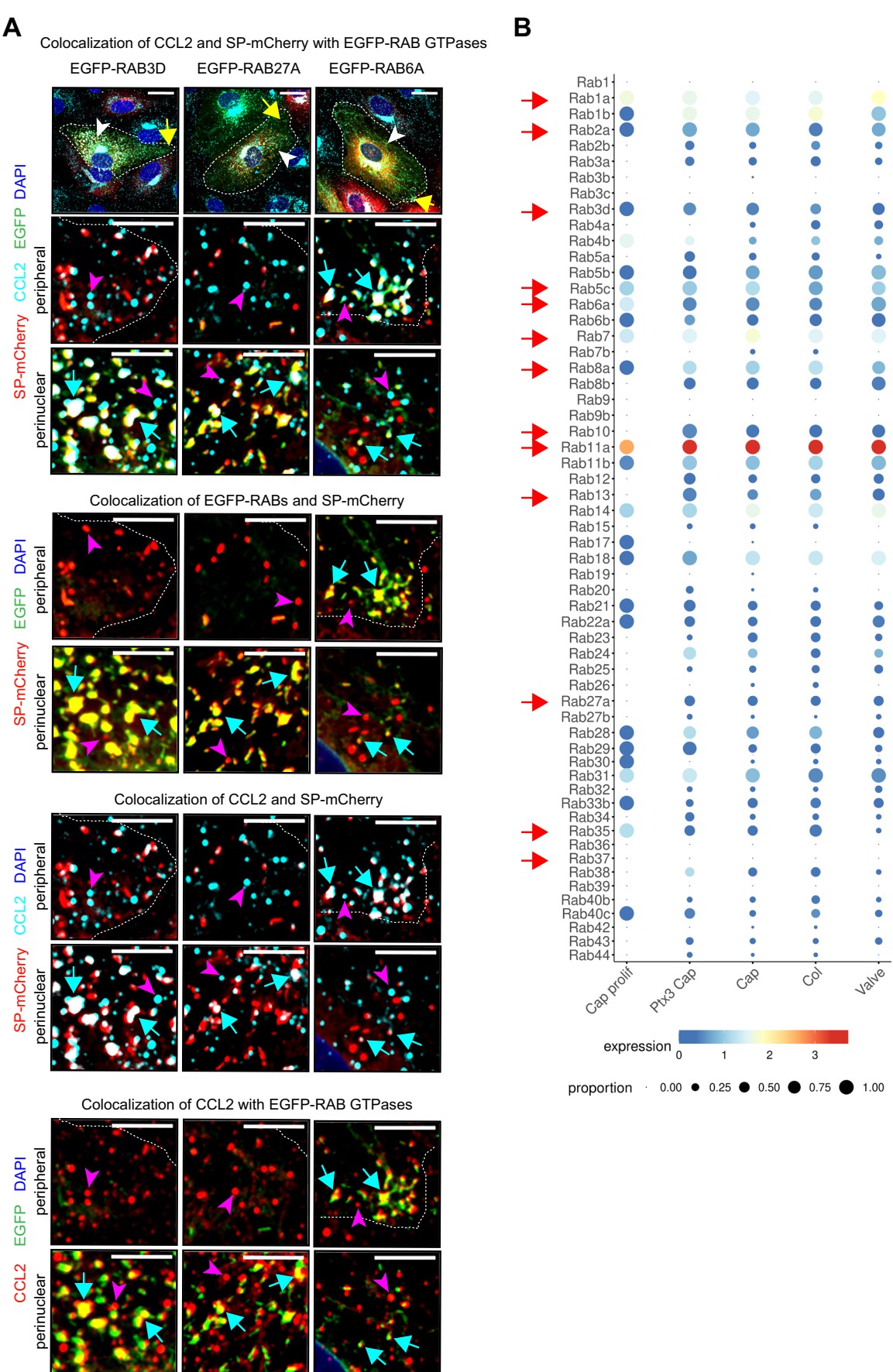

◀ **Figure EV4.  (i) Colocalization of chemokine CCL2 and general secretory marker SP-mCherry with EGFP-RAB-GTPases, (ii) expression levels of RAB-GTPases in dermal LECs.**

(**A**) TNF-α treated LECs expressing signal peptide tagged with mCherry (SP-mCherry; red) together with the indicated EGFP-RAB GTPases (green) and stained for endogenous CCL2 (cyan) and nuclei (DAPI, blue). The overview images show a merge of CCL2, SP-mCherry, and EGFP-RAB signals. The top panel shows a merge of all three channels and the lower panels show a merge of the indicated 2 channels. The images represent $n = 3$ independent experiments. (**B**) Re-analyses of single-cell mRNA sequencing data shows RAB-GTPase expression in mouse dermal LECs in vivo. Cap prolif = proliferative capillary LECs, Ptx3 Cap = Ptx3+ capillary LECs, Cap = capillary LECs, Col = collector LECs, and Valve = valve LECs. The size of the dots indicates the proportion of LECs expressing the indicated RAB-GTPases and the color indicates the level of expression. The data was compiled using the database interface published at https://makinenlab.shinyapps.io/DermaLymphaticEndothelialCells/, which is related to a recent study reported by Petkova et al (data ref: Gene Expression Omnibus, GSE201916) (Petkova et al, 2023). The RAB-GTPases used in this study are indicated with red arrows. Data information: In (**A**), yellow arrows and white arrowheads indicate the site of the peripheral and perinuclear areas, respectively, shown in the zoom-in images. In the zoom-in images, examples of triple positive vesicles (SP-mCherry, EGFP-RAB, and CCL2) are shown with cyan arrows. Magenta arrowheads indicate non-colocalizing CCL2+ or SP-mCherry vesicles. Cell borders are indicated with white dotted lines. Scale bars, 20 μm in overview images; 5 μm in zoom-in images.

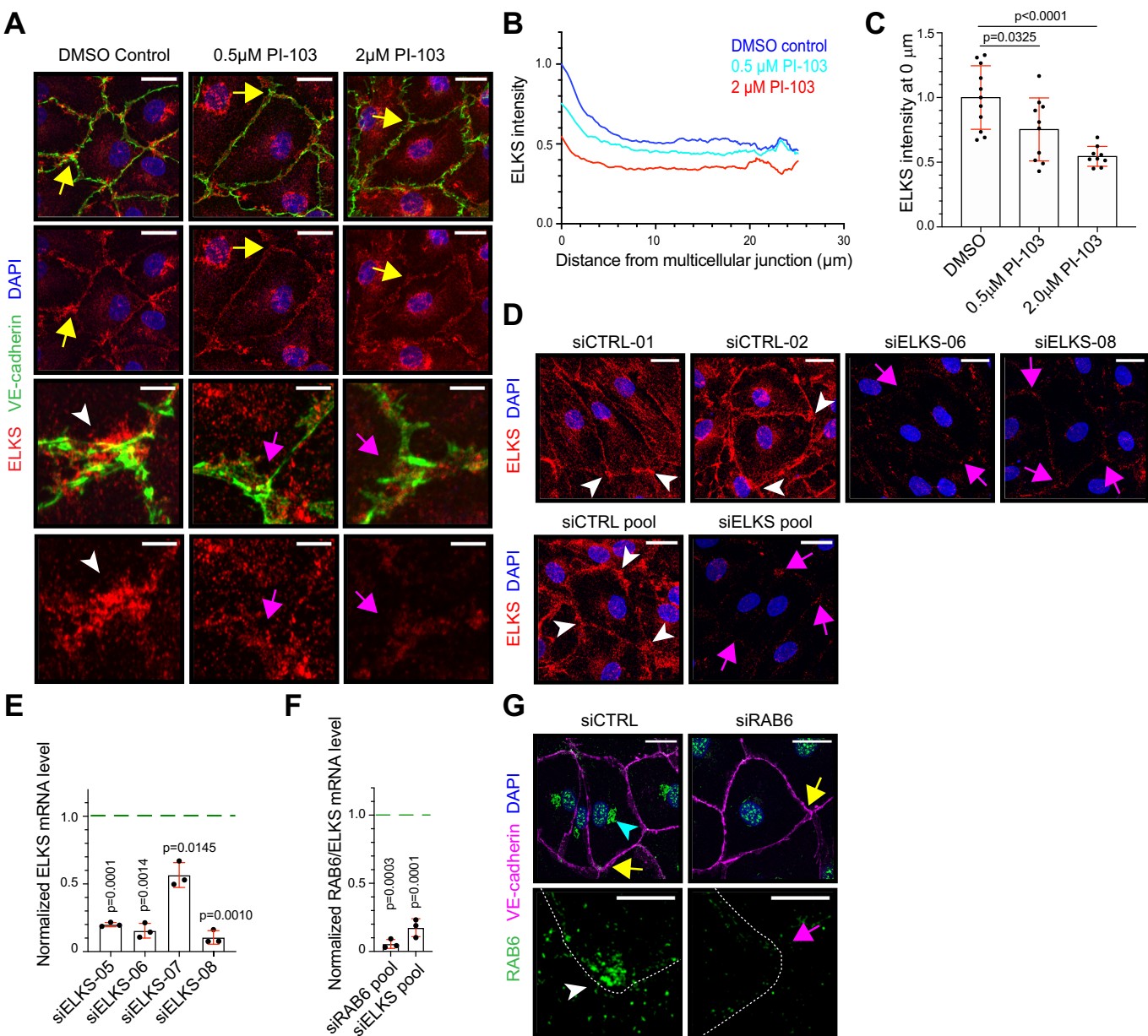

**Figure EV5.** **(i) Inhibition of the PI3K attenuates ELKS localization at the LEC multicellular junctions, and (ii) confirmation of the silencing efficiency of the used siRNA oligos.**

(A–C) LEC monolayer was treated with DMSO (control), 0.5 μM P1-103, or 2 μM PI-103 for 1 h. Fixed samples were stained for ELKS (red), VE-cadherin (green), and nuclei (DAPI, blue). (B, C) Quantification of ELKS staining at the LEC junctions in control (DMSO) or 0.5 μM, or 2 μM PI-103 treated LEC monolayers. The graph in (B) shows mean ELKS intensity as a function of distance from the nearest multicellular junction. In (C), the dot plot shows mean ELKS intensity ± SD at the multicellular junction (distance 0 μm). The results were normalized to the average of controls (set at 1) in each experiment. In (A–C), the data represent $n = 11$ (DMSO), $n = 10$ (0.5 μM PI-103), and $n = 9$ (2 μM PI-103) biological replicates, representing, altogether, 399 (DMSO control), 318 (0.5 μM PI-103), and 286 (2 μM PI-103) junctions in 5 independent experiments. (D) Images show the effect of siRNA-mediated knockdown of ELKS. Monolayers were stained for ELKS (red) and nuclei (DAPI, blue). Images are representative of at least $n = 3$ independent experiments. (E, F) Quantification of ELKS and RAB6 mRNA levels upon siELKS or siRAB6 treatments, respectively. The dot plots show ELKS or RAB6 mean mRNA level ± SD normalized to the average of siControl samples (set at 1, green dashed line) in each experiment. *P*-values show the comparison to controls. Data points represent $n = 3$ independent experiments derived, altogether, from 6 (siELKS-05), 8 (siELKS06), 6 (siELKS-07), 8 (siELKS08), 12 (siControl 01/02), 7 (siELKS pool), 5 (siRAB6 pool), or 12 (siControl pool) biological replicates. (G) Images show the effect of siRAB6 on RAB6 protein level. LEC monolayer was treated with the indicated oligos and stained for RAB6 (green), VE-cadherin (magenta), and nuclei (DAPI, blue). Images are representative of at least $n = 3$ independent experiments. Data information: In (A) and (G), yellow arrows indicate the site of the zoom-in images shown below. In (A), (D), and (G) white arrowheads indicate accumulation at the multicellular junctions and the magenta arrows lack of accumulation. In (G), the cyan arrow in the siCTRL-treated sample shows RAB6 signal in the Golgi apparatus, which is not seen upon knock down in siRAB6 treated sample. In (G), LEC borders are marked with white dotted line. The *p*-values in (C) and (E, F) were calculated using parametric t-test with Welch's correction. Scale bars, 20 μm in overview images; 5 μm in zoom-in images.

