## [Peer Review File · The EMBO Journal]

Spatially targeted chemokine exocytosis guides transmigration at lymphatic endothelial multicellular junctions

Inam Liaqat, Ida Hilska, Maria Saario, Emma Jakobsson, Marko Crivaro, Johan Peränen, and Kari Vaahtomeri

Corresponding author: Kari Vaahtomeri (kari.vaahtomeri@helsinki.fi)

Review Timeline:

Submission Date:	26th Jul 23
Editorial Decision:	14th Sep 23
Revision Received:	27th Mar 24
Editorial Decision:	17th Apr 24
Revision Received:	24th Apr 24
Accepted:	29th Apr 24

Editor: Kelly Anderson

Transaction Report:

Dear Dr. Vaahomeri,

Thank you for submitting your manuscript for consideration by the EMBO Journal. It has now been seen by three referees whose comments are shown below.

Given the referees' positive recommendations, I would like to invite you to submit a revised version of the manuscript, addressing the comments of all three reviewers. I should add that it is EMBO Journal policy to allow only a single round of revision, and acceptance of your manuscript will therefore depend on the completeness of your responses in this revised version. I think it would be good to discuss your plan to address the referee concerns and I am available to do so via email or zoom in the coming weeks. I have also attached a guide for revisions for your convenience.

Thank you for the opportunity to consider your work for publication. I look forward to your revision.

Yours sincerely,

Kelly M Anderson, PhD
Editor, The EMBO Journal
k.anderson@embojournal.org

We realize that it is difficult to revise to a specific deadline. In the interest of protecting the conceptual advance provided by the work, we recommend a revision within 3 months (13th Dec 2023). Please discuss the revision progress ahead of this time with the editor if you require more time to complete the revisions.

Referee #1:

The study by Liaqat et al describes an intriguing cellular mechanism that regulated dendritic cell transmigration through lymphatic endothelial cell monolayers. Specifically, they show that the vertices where multiple cells meet is a location with increased exocytosis of CCL21. This guides dendritic cells via CCR7 and promotes transmigration specifically at these sites. Furthermore, the authors show that RAB6 and its regulators are important in this localised exocytosis and for dendritic cell transmigration.

The lymphatic vasculature represents an important trafficking system for adaptive and innate immune cells in tissues and importantly at lymph nodes. As such understanding how immune cells interact with lymphatic vessels will provide important insights in vascular biology and immunology. While a general role for CCL21 and CCR7 was already established, this work could improve understanding in how immune cells migrating along endothelia find the point for transmigration regulated by this pathway. The idea of spatially regulated exocytosis is especially interesting.

The work is generally very well performed and the paper is well written. However, at many points representative data is used with no careful quantification and this makes the robustness of the experiments hard to judge. In addition, it is important and it should be relatively easy to demonstrate if CCL21 and RAB6 colocalisation in vivo is specifically enriched at junctional vertices in support of this mechanism occurring in real tissues. Finally, the demonstration of altered transmigration of DCs upon altered exosome activity at the junctional vertices relies on only RAB8A DN treatment. This is promising but use of a DN construct could have non-specific effects. Demonstrating the same effect with inhibition of secretion in more than one way will be important.

Specific concerns are outlined below.

1. Were the CCR7^{-/-} DCs that migrated abnormally in Figure 1 and Movie 1 & 2 otherwise phenotypically normal? Did they generally express normal cell surface receptors and display normal marker expression for contractile proteins, normal actin distribution, normal size and shape? It is important to know that the cells are overall quite normal and the behavioural defects were therefore primary.
2. The paper presents generally convincing data in cultured cells but little correlation in real lymphatic vessels in vivo. High quality imaging of lymphatics is well established in mice. Was it possible to quantify the % of CCL21 and RAB6 colocalisation in peripheral/junctional versus perinuclear regions in the in vivo stains performed (eg. Fig 2I)? Imaging of mouse skin vessels should provide sufficient resolution for this and showing the same relationship clearly and formally in vivo (more than just representative images) would significantly enhance the paper.
3. The quantification of exocytosis events based on distance to multicellular junctions in Figure 2B and 2C appears to present a very nice result. However, it is stated that it is based on 241 secretion events from 22 LECs. The data should therefore be presented in a way that makes variance between cells more clear. There needs to be an indication of standard deviation presented as an error bar to have a feel for how consistent this observation is between cells.
4. It is stated that the colocalisation of CCL21-mCherry and RAB6A was especially observed at multicellular junctions. Quantification is needed as in Fig 2C where the distance of the co-localised expression relative to the multicellular junction is indicated. The quantification in Fig 2E is also a little confusing, is this at the junctions or perinuclear or the combination of both? Clearer and more detailed quantification is needed.
5. The % co-localisation in perinuclear and peripheral regions in Fig S2 needs to be shown. Just saying events are "rare" or "more abundant" but not providing data beyond representative images makes the robustness of these observations difficult to judge.
6. Some quantification would improve clarity for statements like on page 9, 191 - "... (Fig. S4C-D), albeit to a much lower extent than the EGFP-RAB6A, RAB3D, RAB27A, and RAB37" . In general there is too heavy a reliance on representative images.
7. In Figure 5 H it appears that ELKS knockdown arrests exocytosis because there is an increase in RAB6-CCL21 coexpressing vesicles in the periphery. Can the authors confirm this is specific to the peripheral regions at the junctions and not just generally an increase in colocalisation throughout all regions of the cell? If it is specifically a defect due to ELKS regulation of exosome release then the increased colocalisation should be very local to the multicellular junctional regions. Quantification in different regions of the cell is needed and would make the result stronger and more convincing.
8. The data showing RAB8A DN reduces DC migration is promising but relying on only one approach could lead to misleading data. Can other methods the reduce exosome release at teh junctional vertices be used to support this finding? Does the ELKS siRNA inhibit transmigration of DCs? Or other equivalent treatments? This would provide further evidence alongside the data in Figure 6 where a dominant negative RAB8A is used and would improve confidence in the findings.

Minor -

- Panel D label is missing in Figure 2

Referee #2:

This manuscript describes targeted secretion of chemokines at multicellular junctions as an important spatial guide that contributes to the preferred transmigration of DC through primary lymphatic monolayers at multicellular junctions. Analyzing 13 different RABs, the authors show that RAB6⁺ vesicles contain the chemokine CCL21 close to multicellular junctions. The RAB6 docking factor ELKS is required for CCL21 exocytosis at junctions. Another component of such a docking complex (as known from other cell types) was needed in LEC for CCL21 secretion, as shown by overexpression of a dominant negative version (RAB8A-DN). This version of RAB8A also inhibited transmigration of DC through the monolayer of LEC.

This study highlights the importance of chemokine targeting to junctions as transmigration guide. It does a very good job in identifying important components of the molecular targeting/trafficking machinery for chemokines and in showing that this machinery is important for the leukocyte transmigration process. Collectively the presented results reveal a mechanism that favors multicellular junctions between LEC as the preferred site of transmigration. Importantly, also CCL21 Δ C induced mainly multicellular transmigration, excluding cell surface chemokine anchoring as a major mechanism for the multicellular diapedesis process. The study seems to be very thoroughly and carefully performed.

The manuscript may benefit from addressing the following points:

Major concerns:

1) Given this study focusses on intracellular molecular mechanisms of chemokine trafficking, it is understandable that most of the experiments are done in vitro. However, figure 2I gives some insight into the in vivo situation. Here, co-staining for CCL21 and RAB6 are shown in the lymphatic vessel of the ear pinna dermis. Instead of using Lyve1 staining, however, it would perhaps be more informative to show VE-cadherin or PECAM-1 in order to know the position of cell junctions. It would be interesting to see whether the Rab6 vesicles are also mainly located in the cell periphery, as is well documented in many of the in vitro images.

2) Since RAB8A was found to be important for the secretion of CCL21deltaC via a RAB6-dependent mechanism (Fig. 6), why then was there no or only modest colocalization found with CCL21deltaC as reported on page 7 (line142) and illustrated in Fig. S2A?

3) Since RAB8A seemed to be needed for optimal DC transmigration, it would be good to see whether ELKS is likewise important for diapedesis.

Minor concerns

1) I guess it is difficult to avoid, but it was sometimes cumbersome to follow all the data since jumping between figures and supplemental figures was not straightforward. There was also back and forth jumping between different supplemental figures. Scrolling up and down was a lot more hectic than in other papers.

2) Letter D (not the panel, just the letter) is missing in Fig. 2.

Referee #3:

- general summary and opinion about the principle significance of the study, its questions and findings

Dendritic cells recruitment to and transmigration across the lymphatic endothelium is an initiating and fundamental process to induce adaptive immune responses. That DC CCR7 and LEC expressed CCL21 is a key for driving these steps is well established. However, the details of how these directional cell movements across the endothelium are coordinated and regulated still lacks detailed understanding. Earlier publications by the last author (i.e. Vaahromeri et al., 2017) ref 4 in manuscript, in which they also use overlapping tools as in the current manuscript i.e. mCherry labelled CCL21) has shown that DC transmigration require CCL21 secretion from the endothelium and that this step is connected to Ca flux in LECs triggered by DC-LEC physical interaction, occurring at junctions. Vesicular CCL21 transport along microtubules to the membrane was also demonstrated. The current manuscript can be seen as building on and extending those findings by:

1. Carefully mapping the transmigration sites specifically to multicellular junctions, which has previously not been demonstrated.
2. Showing that DC expressed CCR7 promote identification of multicellular junction sites
3. Elegantly elucidate the vesicular pathway that transport CCL21 containing vesicles to multicellular junctions - RAB6+ vesicles and req docking factor ELKS. This was previously also not known.
4. Show that transport of CCL21 cargo along microtubuli will allow accumulation to multicellular junctions

From a more general interest perspective (beyond DC transmigration) the paper also illustrates underlying mechanisms that can drive cell transmigration specifically to occur at multicellular vertices.

The image analysis of the cell cultures in this manuscript is generally well presented and adequate description of strategy for image analysis has been included. Videos are also illustrative and of good quality. Data is well documented.

- specific major concerns essential to be addressed to support the conclusions

1. Translation in vivo

Major point: The experimental tools/system used in the paper is mainly based on cell culture with some support from staining of lymphatic vessels ear skin. Strength and major findings of this paper come from the analysis of the vesicular trafficking in LECs that has not been described in this detail before. A weakness is that the paper contains little translational experiments in vivo. If authors can show that DC entry in vivo preferentially occur at multicellular junctions it would further strengthen the paper. This can e.g. be evaluated using skin explants or in vivo imaging ear skin. Similarly, the study should be strengthened on the level of image analysis of CCL21 RAB6 in lymphatic vessels in vivo further discussed point 3, see below.

2. Data in relation to earlier work

Questions:

The paper deepens previous understanding but questions remain. Further discussion regarding the current working model of the sequence of events in DC transmigration across the lymphatic endothelium by the authors in relation to their own and others previous data and the new data is required. This is also important in relation of the news value of the manuscript. Previous findings (Vaahtomeri et al., 2017) indicated a need of a physical interaction DC-LEC before a burst in vesicular exocytosis at junctions. The new findings in this manuscript indicate inherent (DC independent) exocytosis at multicellular junctions (Fig 2, Fig 6) guiding DC recruitment. Are these mechanisms cooperating or independent? Or should the previous model (DC-LEC interaction induced release of CCL21) be redefined?

a) Does DCs co-culture further increase CCL21 release and preferentially at multicellular junctions? Licensing transmigration to occur at these sites preferentially? This should be possible to test experimentally. The authors show that they can measure CCL21 in the media.

b) Authors express truncated mCherry labelled CCL21 that do not attach to the cell or tissue dish surface. Still, can endogenously expressed full length CCL21 from the LECs be fully excluded to contribute to patterning the sites for entry?

c) The authors show and discuss that also CCR7KO DCs halt at multicellular junctions albeit in less frequency and with shorter duration (Fig 1E, S1) and that this suggest additional molecular/biophysical determinants beside CCL21. If DC-LEC interaction further increase baseline secretion (see previous question a), will these short interactions still induce LEC to secrete CCL21 or is initial interaction CCR7-CCL21 required (prolonging interaction DC-LEC).

3. Data/Figures Specific comments

Fig 2I

As mentioned earlier, it is important to provide data to further support translation to the in vivo situation. This picture is not convincing, could VE-cadherin be included to further define the junctions and provide more examples high mag? Quantification. Peripheral part of cell body versus central but also by better defining vessel type. Would also strengthen the statement done by the authors line 334-339. Add and image initial (defined not only by LYVE-1 but by selecting the end of the blind ended vessel where button junctions are most frequent) versus analysis of pre-collectors (similar to the vessel area shown) would add further insight and refine analysis. Can the multicellular association of vesicles be seen in both vessels with button junctions and in pre-collectors, with mainly zipper-like junctions? Are there differences?

- minor concerns that should be addressed*

4. Accessibility to evaluate data for all audience:

A major part the illustrations of co-localization is displayed using green and red. For color blind this becomes a major problem and make it hard to be able to evaluate the data. <https://www.ascb.org/science-news/how-to-make-scientific-figures-accessible-to-readers-with-color-blindness/>. Different shapes arrows can be used instead of using different colors, reducing numbers of colors needed in the images. *Decision for the Journal, if these things should be adjusted or not.

REVIEWERS COMMENTS

Referee #1:

The study by Liaqat et al describes an intriguing cellular mechanism that regulated dendritic cell transmigration through lymphatic endothelial cell monolayers. Specifically, they show that the vertices where multiple cells meet is a location with increased exocytosis of CCL21. This guides dendritic cells via CCR7 and promotes transmigration specifically at these sites. Furthermore, the authors show that RAB6 and its regulators are important in this localised exocytosis and for dendritic cell transmigration.

The lymphatic vasculature represents an important trafficking system for adaptive and innate immune cells in tissues and importantly at lymph nodes. As such understanding how immune cells interact with lymphatic vessels will provide important insights in vascular biology and immunology. While a general role for CCL21 and CCR7 was already established, this work could improve understanding in how immune cells migrating along endothelia find the point for transmigration regulated by this pathway. The idea of spatially regulated exocytosis is especially interesting.

The work is generally very well performed and the paper is well written. However, at many points representative data is used with no careful quantification and this makes the robustness of the experiments hard to judge. In addition, it is important and it should be relatively easy to demonstrate if CCL21 and RAB6 colocalisation in vivo is specifically enriched at junctional vertices in support of this mechanism occurring in real tissues. Finally, the demonstration of altered transmigration of DCs upon altered exosome activity at the junctional vertices relies on only RAB8A DN treatment. This is promising but use of a DN construct could have non-specific effects. Demonstrating the same effect with inhibition of secretion in more than one way will be important.

We are glad that the reviewer finds our manuscript well-performed and written. We now provide new experimental data and quantification to address the individual comments and concerns as detailed below.

Specific concerns are outlined below.

1. Were the CCR7^{-/-} DCs that migrated abnormally in Figure 1 and Movie 1 & 2 otherwise phenotypically normal? Did they generally express normal cell surface receptors and display normal marker expression for contractile proteins, normal actin distribution, normal size and shape? It is important to know that the cells are overall quite normal and the behavioural defects were therefore primary.

We have now compared the phenotype of the wild-type and CCR7^{-/-} DCs. The analysis shows that wild-type and CCR7^{-/-} DCs are of the same size and morphology (new Fig. EV1 I and J). Also, the wild-type and CCR7^{-/-} DCs were of the same identity, as shown by cell surface receptor CD86, CD11b, CD11c, and MHC class 2 expression (new Fig. EV1E-

H). Further, both wild-type and CCR7^{-/-} DCs express identical levels of contractile machinery proteins beta-actin and non-muscle myosin heavy chain IIA (new Fig. EV1 K-L). We think that the actin distribution, similar to migration itself, would be expected to be sensitive to CCL21 cues on the lymphatic endothelium and, thus, would not be indicative of non-desired secondary effects. In conclusion, these results show that CCR7^{-/-} DCs are phenotypically normal, thus, warranting our conclusion that the altered migration path of CCR7^{-/-} DCs, is primary to the inability to sense CCL21 (p. 7, lines 136-140).

2. The paper presents generally convincing data in cultured cells but little correlation in real lymphatic vessels in vivo. High quality imaging of lymphatics is well established in mice. Was it possible to quantify the % of CCL21 and RAB6 colocalisation in peripheral/junctional versus perinuclear regions in the in vivo stains performed (eg. Fig 2I)? Imaging of mouse skin vessels should provide sufficient resolution for this and showing the same relationship clearly and formally in vivo (more than just representative images) would significantly enhance the paper.

We thank the referee for this suggestion, which led to rewarding experimentation and quantification. Based on the suggestions of all the three referees, we have now produced new sample sets and provide multiple images of CCL21, RAB6, and VE-cadherin-stained mouse dermis. Using this data set, we show that 39% of CCL21 vesicles co-localized with RAB6 *in vivo*, in LECs with continuous junctions (new Fig. 2K and M and more examples in Appendix Fig. S2A-D). Importantly, analysis of vesicle location as a function of distance from multicellular junctions, indicates that CCL21+ RAB6+ double positive vesicles are enriched at the LEC multicellular junctions *in vivo* in comparison to RAB6 negative CCL21 vesicles (new Fig. 2N). These new experiments solidify the notion of CCL21 delivery to the LEC periphery by RAB6+ vesicles.

In response to question #3 by referee 3, we have now also quantified the CCL21 and RAB6 colocalization in lymphatic capillary LECs that show discontinuous LEC junctions. Also, in these LECs 44% of CCL21 vesicles colocalized with RAB6 and, importantly, these were enriched at the VE-cadherin junctions (new Fig. 2L and O-P, and more examples in Appendix Fig S2E-F), thus, indicating that RAB6 vesicles deliver CCL21 to the LEC-junctions also in the capillary LECs. These colocalization results are presented on p. 11 lines 234-244.

These new results, together with the data showing that DCs preferentially transmigrate lymphatic endothelial multicellular junctions also in dermal explants *ex vivo* (see referee 3, question #1 and new Fig. 1C-D, Appendix Fig. S1A-C, and Videos 2-5), strengthen our notion of targeted exocytosis in driving localized transmigration of DCs, across the lymphatic endothelium.

3. The quantification of exocytosis events based on distance to multicellular junctions in Figure 2B and 2C appears to present a very nice result. However, it is stated that it is based on 241 secretion events from 22 LECs. The data should therefore be presented in a way

that makes variance between cells more clear. There needs to be an indication of standard deviation presented as an error bar to have a feel for how consistent this observation is between cells.

We now present the SD (with error bar) of the exocytosis events in Fig. 2C. We have now added SD (with error bars) also in the line graphs shown in Fig. 2N, 2P, 5D, and 6B.

Additionally, while replotting the line graphs, we noticed that there was a mistake in the binning of the values: the first data point represented the 0-2.5 μ m distance, whereas all the other bins represented 5 μ m distance. We have now corrected this (bin width in all cases is 5 μ m) and the corrected result emphasizes, even further, the association of exocytosis events at the multicellular junctions.

4. It is stated that the colocalisation of CCL21-mCherry and RAB6A was especially observed at multicellular junctions. Quantification is needed as in Fig 2C where the distance of the co-localised expression relative to the multicellular junction is indicated. The quantification in Fig 2E is also a little confusing, is this at the junctions or perinuclear or the combination of both? Clearer and more detailed quantification is needed.

First, we now clarify that the quantification in Fig. 2E shows colocalization as a percentage of CCL21 Δ C-mCherry vesicles in the whole LEC area (p. 9 line 195, Figure legend 2E p. 64 line 1610, and Methods section p. 39 line 919).

Second, we have now analyzed the same data set for the percentage of colocalizing vesicles as a function of distance to multicellular junctions (new Fig. 2F). These analyses highlight the EGFP-RAB6+ CCL21 Δ C-mCherry+ vesicle enrichment at the vicinity of multicellular junctions whereas, the CCL21 Δ C-mCherry vesicles that co-localize with EGFP-RAB3D, -RAB27A, and -RAB37 show a distinct profile, i.e., are enriched in the perinuclear area (p.9, lines 193-197).

5. The % co-localisation in perinuclear and peripheral regions in Fig S2 needs to be shown. Just saying events are "rare" or "more abundant" but not providing data beyond representative images makes the robustness of these observations difficult to judge.

We have now analyzed the colocalization of the CCL21 Δ C-mCherry with EGFP-RAB1A (4.6%), -RAB2A (6.3%), -RAB5C (1.9%), -RAB8A (7%), -RAB11A (2.5%), -RAB35 (0.1%), -RAB7A (11.4%), -RAB10 (1.6%), and -RAB13 (0.0%) in an identical manner to Fig. 2. (point #4 above): we show both % of co-localization in the whole LEC area and also location of colocalizing vesicles as a function of distance from multicellular junctions (new Fig. EV2D-G). These new analyses show that CCL21 Δ C-mCherry exhibits colocalization with EGFP-RAB1A, -RAB2A, -RAB7A, and RAB11A mostly in the perinuclear area, whereas -EGFP-RAB8A colocalizes with CCL21 Δ C-mCherry in the vicinity of multicellular junctions, similar to EGFP-RAB6A.

EGFP-RAB8A differed from other RAB-GTPases by localization both to the plasma membrane and vesicles. This unique localization made the semi-automated Imaris co-localization analyses difficult due to the lower contrast between the vesicle and the surroundings. Thus, we show both semi-automated co-localization analysis (1.5% for the brighter EGFP-RAB8A vesicle) and manual analysis (7.3%, including the moderate EGFP-RAB8A intensity vesicles). The manual analysis is shown in new Fig. EV2 H-K (p. 9 lines 201-205).

We have, also, analyzed the colocalization percentage (whole LEC area) and distance of colocalizing vesicles from multicellular junctions for the full-length CCL21-mCherry with the EGFP-RAB3D (57.1%), -RAB27A (41.9%), -RAB37 (41.2%), -RAB6A (33.2%), -RAB10 (1.4%), and RAB13 (0.0%) (new Fig. EV3C-E). These results recapitulate the co-localization result shown for CCL21 Δ C-mCherry. Altogether, these quantitation strengthen our notion that the EGFP-RAB6A, -RAB3D, -RAB27A, and RAB37 display most robust colocalization with CCL21 Δ C-mCherry and CCL21-mCherry, whereas other studied RAB-GTPases do colocalize with CCL21, but at lower frequency.

6. Some quantification would improve clarity for statements like on page 9, 191 - "... (Fig. S4C-D), albeit to a much lower extent than the EGFP-RAB6A, RAB3D, RAB27A, and RAB37". In general there is too heavy a reliance on representative images.

We have now quantified the co-localization of CCL21 Δ C-mCherry with endogenous RAB7, CD63 (LAMP3), and LAMP1. 5.6 to 7% of CCL21 Δ C-mCherry vesicles colocalized with these endogenous markers of the late endosome/lysosome route (the original images and the new quantification are moved to the main Fig. 3D-H). These results are in line with the 11.4% co-localization of CCL21 Δ C-mCherry with overexpressed EGFP-RAB7A (new Fig. EV2E).

7. In Figure 5 H it appears that ELKS knockdown arrests exocytosis because there is an increase in RAB6-CCL21 coexpressing vesicles in the periphery. Can the authors confirm this is specific to the peripheral regions at the junctions and not just generally an increase in colocalisation throughout all regions of the cell? If it is specifically a defect due to ELKS regulation of exosome release then the increased colocalisation should be very local to the multicellular junctional regions. Quantification in different regions of the cell is needed and would make the result stronger and more convincing.

We have now analyzed the colocalization of CCL21 Δ C-mCherry and endogenous RAB6 in siControl and siELKS samples within a 5 μ m distance from multicellular junctions vs. the rest of the LECs. This analysis shows that siELKS increases the co-expressing vesicles at multicellular junctions (8.4-fold increase, p=0.0035), whereas in the rest of the LEC co-expressing vesicles are not significantly increased (1.97%-fold increase, p=0.1606) (new Fig. 5J, p.15 lines 333-336). These results solidify our notion of preferential CCL21 Δ C-mCherry+ RAB6+ secretion at the LEC multi-cellular junctions.

Further, we now show that also silencing of siRAB6 (see point #8 below) results in blunting of CCL21 Δ C-mCherry exocytosis at multicellular junctions and accumulation in the periphery, whereas whole LEC levels are largely not affected (new Fig. 6A-F). Thus, siRAB6 phenocopies siELKS effect on CCL21 Δ C-mCherry exocytosis and, therefore, further strengthens our notion that RAB6 and ELKS mediate CCL21 exocytosis especially at multicellular junctions.

8. The data showing RAB8A DN reduces DC migration is promising but relying on only one approach could lead to misleading data. Can other methods the reduce exosome release at teh junctional vertices be used to support this finding? Does the ELKS siRNA inhibit transmigration of DCs? Or other equivalent treatments? This would provide further evidence alongside the data in Figure 6 where a dominant negative RAB8A is used and would improve confidence in the findings.

We thank the reviewer for raising this important point. Now, in addition to RAB8A DN, we use a second independent method to blunt the CCL21 Δ C-mCherry secretion at the multicellular junctions and attenuate DC transmigration. Here, we chose to use siRAB6, which was more efficient in silencing the target mRNA than the siELKS (Fig. EV5F). We confirmed that siRAB6 caused almost a total abolishment of CCL21 Δ C-mCherry secretion at the multicellular junctions and, thus, accumulation of the CCL21 Δ C-mCherry (new Fig. 6A-E, p. 15 lines 340-345).

Importantly, RAB6 silencing resulted in attenuated DC transmigration across CCL21 Δ C-mCherry expressing monolayers (by 29%, $p=0.0003$) (new Fig. 6K, Video 11, p.16 lines 364-366). Silencing of RAB6 in monolayers expressing full-length CCL21-mCherry did not show attenuated transmigration (new Fig. 6K, Video 12, p.15 lines 366-368). This is expected, since charged full-length CCL21 accumulates in non-physiological amounts to the cell culture dish surface and, thus, we presume, that anchoring incapable CCL21 Δ C-mCherry mimics the *in vivo* condition better (Vaahtomeri K. et al. 2017 Cell Reports). Indeed, transmigration is less affected upon attenuation of full-length CCL21 than CCL21 Δ C-mCherry also in other contexts (Fig. 7M and Vaahtomeri K. et al. 2017 Cell reports). In conclusion, siRAB6 provides a second independent method to inhibit RAB6-ELKS-mediated CCL21 exocytosis and DC transmigration at the lymphatic endothelial multicellular junctions.

*Minor -
- Panel D label is missing in Figure 2*

We have now added "D" in Fig. 2.

Referee #2:

This manuscript describes targeted secretion of chemokines at multicellular junctions as an important spatial guide that contributes to the preferred transmigration of DC through

primary lymphatic monolayers at multicellular junctions. Analyzing 13 different RABs, the authors show that RAB6+ vesicles contain the chemokine CCL21 close to multicellular junctions. The RAB6 docking factor ELKS is required for CCL21 exocytosis at junctions. Another component of such a docking complex (as known from other cell types) was needed in LEC for CCL21 secretion, as shown by overexpression of a dominant negative version (RAB8A-DN). This version of RAB8A also inhibited transmigration of DC through the monolayer of LEC.

This study highlights the importance of chemokine targeting to junctions as transmigration guide. It does a very good job in identifying important components of the molecular targeting/trafficking machinery for chemokines and in showing that this machinery is important for the leukocyte transmigration process. Collectively the presented results reveal a mechanism that favors multicellular junctions between LEC as the preferred site of transmigration. Importantly, also CCL21deltaC induced mainly multicellular transmigration, excluding cell surface chemokine anchoring as a major mechanism for the multicellular diapedesis process. The study seems to be very thoroughly and carefully performed.

We thank the reviewer for their assessment of our work and for their constructive suggestions, which we have addressed below.

The manuscript may benefit from addressing the following points:

Major concerns:

1) Given this study focusses on intracellular molecular mechanisms of chemokine trafficking, it is understandable that most of the experiments are done in vitro. However, figure 2I gives some insight into the in vivo situation. Here, co-staining for CCL21 and RAB6 are shown in the lymphatic vessel of the ear pinna dermis. Instead of using Lyve1 staining, however, it would perhaps be more informative to show VE-cadherin or PECAM-1 in order to know the position of cell junctions. It would be interesting to see whether the Rab6 vesicles are also mainly located in the cell periphery, as is well documented in many of the in vitro images.

We have now performed CCL21, RAB6, and VE-cadherin triple staining on a new set of mouse ear pinna samples (three independent experiments). The quantification shows that 39% of the CCL21 vesicles are RAB6 positive in LECs displaying continuous junctions *in vivo* (new Fig. 2K and M and more examples in Appendix Fig. S2A-D, p.11 lines 234-237). We, further, analyzed the location of these double-positive vesicles as a function of distance from multicellular junctions. These quantifications show that CCL21+ RAB6+ double positive vesicles are enriched in the vicinity of multicellular junctions, whereas RAB6 negative CCL21 containing vesicles are more abundant further away from multicellular junctions (new Fig. 2N, p. 11 lines 237-239).

In response to referee 3 question #3, we now also show that similarly to pre-collector/collector LECs discussed above, also in lymphatic capillary LECs, RAB6+ CCL21+ double positive vesicles localize to LEC-periphery, i.e., discontinuous LEC junctions, *in vivo* (new Fig, 2L and O-P, and Appendix Fig. S2E-F, p. 11 lines 239-244). These results emphasize the general significance of RAB6 vesicles in the delivery of CCL21 to the site of transmigration.

In response to referee 3 question #1, we now also show that the multicellular junctions and the immediate surroundings are the preferential locations of DC transmigration across lymphatic vessel endothelium in mouse dermal explants (new Figures 1C-D, Appendix Fig. S1A-C, and Videos 2-5, p. 5 lines 104-119). Altogether, these experiments solidify our notion that targeted RAB6 vesicle-mediated CCL21 delivery drives DC transmigration at multicellular junctions.

2) Since RAB8A was found to be important for the secretion of CCL21deltaC via a RAB6-dependent mechanism (Fig. 6), why then was there no or only modest colocalization found with CCL21deltaC as reported on page 7 (line142) and illustrated in Fig. S2A?

We thank the referee for raising this point, which resulted in a fruitful re-analysis of the dataset. The analysis indicated that EGFP-RAB8A differed from other studied RAB-GTPases by localizing both on vesicles and plasma membrane and forming high-intensity areas in the vicinity of multi-cellular junctions, which we now highlight in Fig. EV2H-I. We conducted a semi-automated colocalization analysis, which indicated EGFP-RAB8A colocalization with 1.5% of CCL21 Δ C-mCherry positive vesicles (new Fig. EV2D). Although this detection method is suitable for other RAB-GTPases, in the case of EGFP-RAB8A, it captures only the brightest vesicles, which clearly stand out from the plasma membrane EGFP-RAB8A signal. To complement the analysis, we manually detected low-intensity EGFP-RAB8A vesicle-like punctae, which were not captured by automated analysis but, still, colocalized with CCL21 Δ C-mCherry. This analysis shows that EGFP-RAB8A co-localizes with 7.3% of CCL21 Δ C-mCherry vesicles (new Fig. EV2J). The thin nature of the LEC periphery combined with the limited Z-resolution of spinning disc/confocal microscopy did not allow us to conclude, whether the EGFP-RAB8A signal decorated the CCL21 Δ C-mCherry vesicles or presented clustered EGFP-RAB8A at the site of vesicle docking complexes at multicellular junctions (Fig. EV2H-I).

We also analyzed the EGFP-RAB-GTPase and CCL21 Δ C-mCherry colocalized vesicles as a function of distance from multicellular junctions. The results indicate that the EGFP-RAB6A and EGFP-RAB8A are the only RAB-GTPases, that mostly colocalize with CCL21 Δ C-mCherry in the vicinity of the multicellular junctions, whereas other studied RAB-GTPases colocalize with CCL21 Δ C-mCherry in the perinuclear area (new Fig. EV2G and K, p. 9 lines 204-205).

Previously, RAB8A has been indicated to promote plasma membrane fusion, i.e., exocytosis, of RAB6 vesicles (Grigoriev 2011, *Current Biology*; Shibata 2016, *J. Cell Science*). Those studies suggest that RAB8A promotes RAB6 docking complex formation,

at least in part, on RAB6 vesicles (Grigoriev 2011, Current Biology) but also possibly at the plasma membrane (Shibata 2016, J. Cell Science). Our results show that, in LECs, EGFP-RAB8A localizes to some CCL21 Δ C-mCherry vesicles in the LEC periphery and, in addition, forms high-intensity areas at multicellular junctions (new Fig. EV2H-K). Thus, it is conceivable that RAB8A dynamically colocalizes with CCL21 Δ C-mCherry vesicles at the multicellular junctions. Together with inhibition of CCL21-RAB6 vesicle exocytosis and DC transmigration by dominant negative EGFP-RAB8A (Fig. 7A-G and M), these results indicate that RAB8A is a constituent of the RAB6 - ELKS exocytosis complex in LECs (p. 20, lines 466-472) and, thus, plays a critical role in CCL21 exocytosis and DC transmigration.

3) Since RAB8A seemed to be needed for optimal DC transmigration, it would be good to see whether ELKS is likewise important for diapédesis.

We thank the reviewer for this comment, which was also brought up by the referee 1 (point #8), and which resulted in rewarding experimentation and significant strengthening of the manuscript. In addition to the RAB8A dominant negative approach, we have now used a second means to blunt the RAB6+ vesicle-mediated CCL21 exocytosis in our transmigration assay. We reasoned that the most suitable tool for this is siRAB6 since siRAB6 is more efficient in silencing the target mRNA than siELKS (Fig. EV5F). First, we confirmed that siRAB6-mediated silencing of RAB6 resulted in the abolishment of CCL21 Δ C-mCherry exocytosis at LEC junctions and accumulation especially seen at the multicellular junctions (new Fig. 6A-F, p. 15 lines 340-345). Thus, siRAB6 phenocopied siELKS effect, further strengthening the notion that RAB6 and ELKS are the critical molecules mediating CCL21 exocytosis at multicellular junctions.

Next, we addressed, whether RAB6 is essential for DC transmigration across the LEC monolayers. Our results indicate that siRAB6 resulted in a 29% reduction in DC transmigration across CCL21 Δ C-mCherry expressing LEC monolayers (new Fig. 6K, Video 11, p. 16 lines 364-366). siRAB6 did not attenuate transmigration across CCL21 full length-mCherry expressing monolayers (0.8% reduction). This is expected, since the full-length CCL21-mCherry accumulates on the cell culture dish surface in non-physiological amounts (Vaahtomeri K. et al. 2017, Cell Reports). Thus, we presume that CCL21 Δ C-mCherry recapitulates *in vivo* context better. Indeed, attenuation of the CCL21 exocytosis on full-length CCL21-mCherry expressing monolayers shows a milder effect on DC transmigration also in other contexts (Fig. 7M and Vaahtomeri K. et al. 2017, Cell Reports). The more moderate effect of siRAB6 on transmigration, in comparison to EGFP-RAB8A DN, is in line with more moderate accumulation of the chemokine (compare quantifications in Fig. 6C-F and 7C-F). In conclusion, here we used a second tool to show that inhibition of RAB6-mediated exocytosis attenuates DC transmigration, thus, supporting our conclusion of targeted CCL21 exocytosis at multicellular junctions drives DC transmigration.

Minor concerns

1) I guess it is difficult to avoid, but it was sometimes cumbersome to follow all the data since jumping between figures and supplemental figures was not straightforward. There was also back and forth jumping between different supplemental figures. Scrolling up and down was a lot more hectic than in other papers.

We thank the referee for this notion. The space in the main figures is limiting and we have aimed at having the images and graphs at sufficient size to allow evaluation of results even from a printed version. We have now translocated some of the supplemental images/graphs to the main figures, when the available space has allowed this. For this purpose, we have, for example, split the former Fig. 5 to Fig. 5 and 6 to accommodate the new results and some of the former supplements (current Fig. 3D-F, 5G, 6H, 7E, and J).

In the text, we tend to refer to the results mentioned in earlier figures. Here, our purpose has been to highlight the cohesiveness of the story. We have now critically read through these internal references and removed the ones that were unnecessary.

We hope that these changes have improved the readability of the manuscript.

2) Letter D (not the panel, just the letter) is missing in Fig. 2.

We have now added "D" in Fig. 2.

Referee #3:

- general summary and opinion about the principle significance of the study, its questions and findings

Dendritic cells recruitment to and transmigration across the lymphatic endothelium is an initiating and fundamental process to induce adaptive immune responses. That DC CCR7 and LEC expressed CCL21 is a key for driving these steps is well established. However, the details of how these directional cell movements across the endothelium are coordinated and regulated still lacks detailed understanding. Earlier publications by the last author (i.e. Vaahhtomeri et al., 2017) ref 4 in manuscript, in which they also use overlapping tools as in the current manuscript i.e. mCherry labelled CCL21) has shown that DC transmigration require CCL21 secretion from the endothelium and that this step is connected to Ca flux in LECs triggered by DC-LEC physical interaction, occurring at junctions. Vesicular CCL21 transport along microtubules to the membrane was also demonstrated. The current manuscript can be seen as building on and extending those findings by:

- 1. Carefully mapping the transmigration sites specifically to multicellular junctions, which has previously not been demonstrated.*
- 2. Showing that DC expressed CCR7 promote identification of multicellular junction sites*
- 3. Elegantly elucidate the vesicular pathway that transport CCL21 containing vesicles to multicellular junctions - RAB6+ vesicles and req docking factor ELKS. This was previously also not known.*

4. Show that transport of CCL21 cargo along microtubuli will allow accumulation to multicellular junctions

From a more general interest perspective (beyond DC transmigration) the paper also illustrates underlying mechanisms that can drive cell transmigration specifically to occur at multicellular vertices.

The image analysis of the cell cultures in this manuscript is generally well presented and adequate description of strategy for image analysis has been included. Videos are also illustrative and of good quality. Data is well documented.

We thank the reviewer for their insightful review of our work and for their constructive comments, which we have addressed below.

- specific major concerns essential to be addressed to support the conclusions

1. Translation in vivo

Major point: The experimental tools/system used in the paper is mainly based on cell culture with some support from staining of lymphatic vessels ear skin. Strength and major findings of this paper come from the analysis of the vesicular trafficking in LECs that has not been described in this detail before. A weakness is that the paper contains little translational experiments in vivo. If authors can show that DC entry in vivo preferentially occur at multicellular junctions it would further strengthen the paper. This can e.g. be evaluated using skin explants or in vivo imaging ear skin. Similarly, the study should be strengthen on the level of image analysis of CCL21 RAB6 in lymphatic vessels in vivo further discussed point 3, see below.

We thank the reviewer for these suggestions, which resulted in successful experimentation and marked strengthening of the manuscript. We have now carried out the suggested skin explant DC transmigration assays. Upon loading the LPS-activated DCs on the exposed dermis, we observe robust transmigration of dendritic cells into the pre-collector lymphatic vessels (as shown by Arasa J. et al. 2021, J. Exp. Med.). In this assay context, we now labelled the cell junctions with α -CD31-FITC antibody.

We recorded, altogether, 33 DC transmigration events (6 biological replicates in 3 independent experiments) at the LEC junctions. Importantly, 48.5% of transmigration events occurred at the multicellular junction and 24.2% in the immediate vicinity, i.e., within 5 μ m of the multicellular junctions. The rest, 27.3%, occurred at 5-18 μ m distance from the multicellular junctions (new Fig. 1C-D, Appendix Fig. S1A-C, and Videos 2-5, p. 5 lines 104-119). These results show that the multicellular junctions and the immediate surroundings present the preferred site of dendritic cell transmigration also in the tissue context. Together with the primary cell culture studies (Fig. 1A-B), these results show that multi-cellular junctions are the preferential site of the DC transmigration across lymphatic endothelium. We agree with the referee that intravital *in vivo* imaging would

be ideal for future studies in this research area but setting it up is out of the scope of this study.

In point #3 below, we show that CCL21 and RAB6 colocalize at the multicellular junctions, also, *in vivo*. Together, these results show that CCL21 at multicellular junctions determines the preferential site of DC transmigration across lymphatic endothelium, also, in the tissue context.

2. Data in relation to earlier work

Questions:

The paper deepens previous understanding but questions remain. Further discussion regarding the current working model of the sequence of events in DC transmigration across the lymphatic endothelium by the authors in relation to their own and others previous data and the new data is required. This is also important in relation of the news value of the manuscript. Previous findings (Vaatmeri et al., 2017) indicated a need of a physical interaction DC-LEC before a burst in vesicular exocytosis at junctions. The new findings in this manuscript indicate inherent (DC independent) exocytosis at multicellular junctions (Fig 2, Fig 6) guiding DC recruitment. Are these mechanisms cooperating or independent? Or should the previous model (DC-LEC interaction induced release of CCL21) be redefined?

We thank the referee for these notions and have now performed new experiments (see below for details) to address the raised points. We have also extended the discussion of the current findings in the context of previous publications of our own and others (see below and p. 22 lines 499-518, and p. 22 lines 522-535).

a) Does DCs co-culture further increase CCL21 release and preferentially at multicellular junctions? Licensing transmigration to occur at these sites preferentially? This should be possible to test experimentally. The authors show that they can measure CCL21 in the media.

We thank the reviewer for bringing up this point that is in the heart of our research but has proven to be complicated to address directly.

As the referee points out, we are able to measure the bulk CCL21-mCherry signal in the media. However, this assay does not make a distinction between the two CCL21 storage vesicle populations (RAB6 or RAB27/3), since all the CCL21-mCherry in the media is measured. Another complication is caused by the co-cultured DCs, which sequester the CCL21-mCherry (or CCL21 Δ C-mCherry), thus, reducing the mCherry signal in the media (Fig. 1A-C response to referees below). This sequestration of CCL21-mCherry might occur via the binding of CCL21 on CCR7 or charged glycans, and by subsequent internalization, as reported for CCL19, a closely related ligand of CCR7 (Bardi G. et al.

2001, EJI; Alanko J. et al. 2023, Sci. Imm.). Thus, bulk CCL21-mCherry levels in the media do not reflect the amount of exocytosed CCL21.

Figure for reviewers removed

As an alternative approach, we conducted live imaging to directly monitor multicellular junction CCL21 exocytosis events during the DC arrest and transmigration of multicellular junctions. This approach proved to be difficult, since capture of transient arrest and transmigration events with high magnification objective is rare. One of the

other challenges is that the transmigrating DC alters the focus plane of the LECs, making focusing on the CCL21 vesicles difficult. In the captured high-quality videos of multicellular junctions, with an arrested DC on top of the multicellular junction (n=17 DCs/videos, in 12 biological replicates in 2 experiments), we did not observe an increase in local exocytosis in relation to no-DC controls (n=21 videos, in 15 samples in 2 experiments). We now provide representative videos of CCL21 Δ C-mCherry dynamics in control and "arrested DC on top" multicellular junctions (new Video 8, p. 8 lines 173-177).

Conditional secretion is calcium-dependent. We have previously shown that Ca⁺⁺ chelation with BAPTA-AM decreases total CCL21 Δ C-mCherry amounts in the media (Vaahtomeri K. et al. 2017, Cell Reports). Now, to address, whether CCL21 exocytosis, specifically, at multicellular junctions is conditional, we chelated LEC intracellular calcium with BAPTA-AM. We recorded CCL21 vesicle dynamics with live microscopy and quantified the vesicle release at multicellular and bicellular LEC junctions, i.e., the population, which is sensitive to ELKS and RAB6 silencing (see Fig. 5C-D and new Fig. 6A-B). Calcium chelation did not affect CCL21 exocytosis at LEC junctions (new Fig. EV1P, p. 8 lines 177-180). Based on these results, we now write on p. 8 lines 180-183: "Although we cannot fully exclude conditional CCL21 exocytosis during some point of transmigration, these results, altogether, suggest that CCL21 exocytosis at primary LEC multicellular and bicellular junctions is constitutive." This would be in line with previous reports, showing constitutive exocytosis of Golgi-derived RAB6 secretory vesicles in other cell types (Grigoriev I. et al. 2006, Current Biology, Fourriere L. 2019, JCB).

Previously, we have shown that, in primary LEC cultures, some of the CCL21 storage vesicles are calcium sensitive and DCs induce calcium fluxes in LECs. Further, in dermal explants, DC transmigration into lymphatic vessels results in the mobilization of intra-LEC CCL21 storage vesicles and an increase in the extracellular CCL21, thus, indicating that a part of the CCL21 secretion is conditional (Vaahtomeri K. et al. 2017, Cell reports). Based on our current results, it is conceivable that the conditional arm of the CCL21 secretion is carried out by RAB3/27+ secretory granules, which are, including blood endothelial Weibel Palade-bodies (Naß J. et al. 2021, Front Cell Dev Biol), conditional and calcium-sensitive (Burgoyne R.D. et al. 2003, Physiological reviews). Based on our current and, also, our and others' previous results, we have now modified the model of CCL21-driven transmigration, as presented on page 22 (lines 505-518): "...This conditional arm of CCL21 exocytosis may be accounted for the RAB3/27 positive CCL21 storage granules in primary cell cultures and RAB6 negative CCL21 storage granules *in vivo*. The RAB6 negative CCL21 vesicles are found also at the LEC junctions *in vivo*, although, more dispersed throughout the LEC than the RAB6+ positive CCL21+ vesicles, which are more restricted to multicellular junctions and LEC borders (Fig. 2J-P and Appendix. Fig S2A-F). Based on the previous and current results, we propose a model, where RAB6+ vesicle-mediated CCL21 exocytosis at the LEC multicellular junctions and LEC borders forms a constitutive cue that guides DCs to the preferred transmigration sites. Whereas, a conditional burst of RAB6 negative CCL21+ vesicles, upon DC-LEC contact, may mark the arrival of a DC to the lymphatic vessel, thus, slowing down the fast interstitial migration

[21, 23, 24], and/or mark the sites of successful transmigration [4, 23]. Thus, the two mechanisms of CCL21 exocytosis would be cooperating." And on page 22 (lines 522-535) "...The spatially controlled exocytosis, as characterized here, is expected to synergize with other reported control mechanisms of CCL21 presentation in space, including anchoring of CCL21 [25, 47, 56-58], cleavage by DCs [59], tissue fluid and lymph flow [26, 60], and decoy receptor-mediated sequestration [61, 62]. We envision that spatio-temporal control of CCL21 presentation and, possibly, semaphorin-plexin driven DC migration [63], guide the DCs to multicellular junctions, where CCL21 together with direct DC-LEC contacts mediated by hyaluronan-LYVE1 [64, 65], and, upon inflammation, integrin-ICAM/VCAM [19, 27, 43, 66], allow the DCs to arrest, exert force, and transmigrate. In this context, we propose that the use of two distinct exocytosis routes allows high-fidelity control over chemokine presentation and DC guidance to the site of transmigration."

In the longer term, the identification of critical gatekeepers of each of the LEC secretory routes (this and future studies) will allow the use of genetic models to further dissect the role of each of the secretory routes *in vivo*.

b) Authors express truncated mCherry labelled CCL21 that do not attach to the cell or tissue dish surface. Still, can endogenously expressed full length CCL21 from the LECs be fully excluded to contribute to patterning the sites for entry?

The expression of endogenous CCL21 is downregulated upon LEC culture (Wick N. et al. 2007, *Physiol. Genomics*). Accordingly, in our setup, low levels of endogenous CCL21 are not sufficient to drive the transmigration of DCs. However, in a transwell-assay DCs transmigrate LEC monolayer in an endogenous CCL21-dependent manner (Johnson L.A. et al. 2010, *Int Immunol.*). Thus, we addressed, whether endogenous CCL21 would contribute to the patterning of the transmigration sites in our assay conditions.

First, to directly address whether endogenous CCL21 is localized to multicellular junctions, we stained the endogenous CCL21 with commercially available antibodies, but no CCL21-specific signal was observed (data not shown). Also, in earlier publication (Johnson L.A. et al. 2010, *Int. Immunol.*) the staining of the endogenous CCL21 in primary LEC culture was restricted to perinuclear/Golgi area and did not recapitulate the full punctate peripheral pattern of CCL21 observed *in vivo* or upon overexpression of CCL21-mCherry *in vitro* (Johnson L.A. et al. 2010, *Int. Immunol*; Weber M. et al. 2013, *Science*; Vaahtomeri K. et al. 2017, *Cell Reports*; and current manuscript), possibly, reflecting the reduced CCL21 expression levels. In summary, we were unable to determine whether the low amounts of endogenous CCL21 localize to multicellular junctions.

Instead, to functionally test the significance of endogenous CCL21 in patterning the LEC multicellular junctions, we took advantage of our assay set-up, where overexpression of mouse CCL21-mCherry drives DC transmigration across the human lymphatic endothelial cells (Vaahtomeri K. et al. 2017 and the current manuscript). Thus, we silenced endogenous hCCL21 in LECs overexpressing mouse CCL21 Δ C-mCherry or full-length

CCL21-mCherry. We used qPCR to confirm that hCCL21 was efficiently silenced in this assay context (new Fig. EV1M), while overexpressed mouse CCL21 Δ C-mCherry was not affected (new Fig. EV1N). Thus, we were able to address, whether endogenous hCCL21 has a role in patterning the transmigration sites.

In these experiments, overexpression of mCCL21 Δ C-mCherry or mCCL21-mCherry was sufficient to drive transmigration even in the absence of endogenous hCCL21. Analysis of the transmigration sites showed that hCCL21 silencing did not have a marked effect on the percentage of DCs transmigrating the multicellular junctions of mCCL21 Δ C-mCherry expressing (siControl-01 88%, siControl-02 77%, siCCL21-05 75%, siCCL21-08 69%) or full-length mCCL21-mCherry expressing (siControl-01 79%, siControl-02 77%, siCCL21-05 72%, siCCL21-08 75%) LEC monolayers (new Fig. EV1O and p. 7 lines 152-158). These results indicate that the localized CCL21 exocytosis is sufficient to drive DC transmigration at LEC multicellular junctions. However, we discuss that it is conceivable that CCL21 anchoring contributes, but is not necessary, for the patterning of the transmigration sites (see discussion on p. 20 lines 455-462 and p. 22, lines 522-527).

c) The authors show and discuss that also CCR7KO DCs halt at multicellular junctions albeit in less frequency and with shorter duration (Fig 1E, S1) and that this suggest additional molecular/biophysical determinants beside CCL21. If DC-LEC interaction further increase baseline secretion (see previous question a), will these short interactions still induce LEC to secrete CCL21 or is initial interaction CCR7-CCL21 required (prolonging interaction DC-LEC).

As mentioned in point #2A, we do not detect DC-induced CCL21 exocytosis at multicellular corners and, thus, the RAB6-mediated CCL21 exocytosis is considered to be constitutive. We hypothesize that the additional determinants may include, for example, hyaluronan-LYVE1 (Johnson L.A. et al. 2017, Nat. Immunol.; Stanly, T.A. et al. 2020, J. Biol. Chem.) and integrin-ICAM/VCAM interactions (Johnson, L.A. et al. 2006, J. Exp. Med.; Johnson L.A. et al. 2010, Int. Immunol.; Teijeira A. et al. 2013, J. Invest. Dermatol.; Arasa J. et al. 2021, J. Exp. Med), similar to integrin ICAM interactions in blood endothelium (Sumagin R. 2010, J. Immunol.; Gronloh M.L.B. 2023, EMBO Reports). These interactions may allow DC to arrest and exert force, whereas localized CCL21 release contributes to the arrest and guides DC transmigration to and across the multicellular junctions (discussion on p. 22 lines 527-533).

3. Data/Figures Specific comments

Fig 2I

As mentioned earlier, it is important to provide data to further support translation to the in vivo situation. This picture is not convincing, could VE-cadherin be included to further define the junctions and provide more examples high mag? Quantification. Peripheral part of cell body versus central but also by better defining vessel type. Would also strengthen the statement done by the authors line 334-339. Add and image initial (defined not only by LYVE-1 but by selecting the end of the blind ended vessel where button junctions are

most frequent) versus analysis of pre-collectors (similar to the vessel area shown) would add further insight and refine analysis. Can the multicellular association of vesicles be seen in both vessels with button junctions and in pre-collectors, with mainly zipper-like junctions? Are there differences?

We have now carried out new sets of experiments and included VE-cadherin staining alongside CCL21 and RAB6 staining. We provide multiple exemplary high-magnification images (new Fig. 2K and Appendix Fig. S2A-D) and have quantified both the colocalization percentage of CCL21 and RAB6 and the location of colocalized and non-colocalized CCL21 vesicles as a function of distance from LEC multicellular junctions. These results show, in line with cell culture experiments, that 39% of CCL21 vesicles are RAB6 positive in dermal LECs showing continuous junctions *in vivo* (new Fig. 2M). Importantly, RAB6 positive CCL21-containing vesicles are more enriched at the multicellular junctions than RAB6 negative CCL21 vesicles (new Fig. 2N, p. 11 lines 237-239). These *in vivo* results solidify our notion of RAB6 vesicle-mediated delivery of CCL21 to multicellular junction.

We have, now, also analyzed CCL21 colocalization with RAB6 in the context of blind-ended lymphatic capillaries, which show discontinuous VE-cadherin junctions. We have a limitation of 3-color staining (CCL21, RAB6, VE-cadherin) and, thus, cannot include LYVE1 staining here. However, for this analysis, we imaged only the tip segments of the lymphatic vessels, which, together with discontinuous VE-cadherin junctions, guarantee the capillary identity of the LECs. This analysis showed that 44% of CCL21 vesicles were positive for RAB6 (new Fig. 2L and O, Appendix Fig. S2E-F). Due to the discontinuous pattern of capillary LEC VE-cadherin, and the overlapping of neighboring LECs, we cannot determine the location of the vesicles in regard to the multicellular junctions. However, we have analyzed the location of RAB6 positive vs. RAB6 negative CCL21 vesicles as a function of distance from discontinuous VE-cadherin positive junction segments. These results indicate that CCL21-RAB6 vesicles are enriched at the VE-cadherin junctions of lymphatic capillary LECs. The location of RAB6 negative CCL21 vesicles is more dispersed throughout the LECs (both at the junctions and LEC body) (new Fig. 2P, p. 11 lines 239-244).

Altogether, these results show that RAB6 vesicles deliver CCL21 to multicellular LEC junctions, *in vivo*, both in LECs with continuous (zipper-like) and discontinuous junctions, thus, highlighting the general significance of our finding.

*- minor concerns that should be addressed**

4. Accessibility to evaluate data for all audience:

A major part the illustrations of co-localization is displayed using green and red. For color blind this becomes a major problem and make it hard to be able to evaluate the data. <https://www.ascb.org/science-news/how-to-make-scientific-figures-accessible-to-readers-with-color-blindness/>. Different shapes arrows can be used instead of using

*different colors, reducing numbers of colors needed in the images. *Decision for the Journal, if these things should be adjusted or not.*

We have discussed this with the editor and decided to provide additional versions of the images. The color combination of green, red, and yellow is very powerful for showing the colocalization for non-color-blind persons and, thus, we still provide the data with these colors in the main images. Now, we also provide all the main figure images in the Appendix with alternative colors that are suitable for most of the color-blind persons (Appendix Fig. S5-10). We refer to these figure panels in the main figure legends.

We have, also, transformed some of the arrows into arrowheads and have selected colors for the benefit of most of the color-blind persons (concerning all the figures).

We expect that these changes, suggested by the referee, considerably improve the data accessibility for color-blind persons.

Dear Kari,

Congratulations on a great revision! Overall, the referees have been positive. However, there remain several editorial items that we ask you to address in a revised version. When you submit your revision, please add the following in a new point-by-point:

1. Please review our new policy on conflicts of interests on our author website and rename this section to: "Disclosure statement and competing interests".
2. Please remove the author contribution section from the main manuscript.
3. We do not allow data not shown in our publications, please correct these references or add the figures as supplemental on p76, twice on p81, and p82.
4. In the figure legends, please correct so that all the main figures are listed first, followed by EV figures.
5. The file names and callouts should be "Movie EV1", etc. instead of "Video 1", etc. Movie legends need to be removed from the ms file: each legend should be provided as a readme.txt file and then it should be zipped up together with its corresponding movie so that we have one movie folder uplidd per one movie.
6. Please note that a separate 'Data Information' section is required in the legends of figures 1a, c, e; 2a-b, d, g-l, m-n, o-p; 3a-f, i; 4a-d; 5a, c, e-j; 6a, c-k; 7a-k; EV 1a-b, e-j, l-n, p; EV 2a-c, h-i; EV 5a, c-g.
7. Please define error bars in the legend of fig EV1l.
8. Please note that the measure of center for the error bars needs to be defined in the legends of figures 2c, n, p.
9. Please add a scale bar and its definition for fig 7l.
10. Please define the cyan arrowheads in the legend of fig EV5g.
11. Please clarify: the data callouts in the text for Petkova et al., 2023 data citation includes both "Data ref." and ""DATASET"" as a prefix. The data citation Petkova et al., 2023, is not tagged with the label "DATASET" in the reference list, however the following data citation prefixed as ""DATASET Gene Expression Omnibus, GSE201916, <https://makinenlab.shinyapps.io/DermaLymphaticEndothelialCells/>"" is provided in the list.
12. Please provide the URL for Petkova et al., 2023.

Thank you for the opportunity to consider your work for publication, I look forward to your revision.

Warm wishes,
Kelly

Kelly M Anderson, PhD
Editor, The EMBO Journal
k.anderson@embojournal.org

Referee #2:

The authors have done a very good job in addressing all the points that were raised.

Referee #3:

- general summary and opinion about the principle significance of the study, its questions and findings

Thank you to the authors for addressing the concerns raised in the previous reviews. The authors have performed a very well performed revision of the manuscript which has both strengthened and expanded the manuscript findings. I find that the authors addressed all important points raised by the 3 reviewers.

The main concerns of translational value of the findings are now addressed using explant cultures and by further analysis of dermal lymphatic vessels. The authors clearly show preference for DC transmigration through multicellular junctions under the conditions of dermal explants. I agree further analysis in vivo is out of the scope of this current paper. They also performed imaging and quantification of CCL21 positive versus negative RAB6 vesicles as a function of distance from LEC multicellular junctions, which support the previous data retrieved from cell cultures. The authors have also added experiments to further address induced/conditional versus constitutive CCL21 exocytosis convincingly demonstrating that the exocytosis is mainly constitutive. They also elegantly show that the endogenous expression of full length CCL21 in their experimental setup does not contribute to the DC migrational pattern. With the also added discussion of an interesting plausible model based on the current data, their paper provides a valuable base for continuing research to understand these regulatory steps.

- specific major concerns essential to be addressed to support the conclusions

No further changes are needed.

- minor concerns that should be addressed

No minor concerns.

The authors addressed the remaining editorial issues.

Dear Kari,

Congratulations on an excellent manuscript, I am pleased to inform you that your manuscript has been accepted for publication in the EMBO Journal. Thank you for your comprehensive response to the referee concerns and for providing detailed source data. It has been a pleasure to work with you to get this to the acceptance stage.

I will begin the final checks on your manuscript before submitting to the publisher next week. Once at the publisher, it will take about three weeks for your manuscript to be published online. As a reminder, the entire review process, including referee comments and your point-by-point response, will be available to readers.

I will be in touch throughout the final editorial process until publication. In the meantime, I hope you find time to celebrate!

Warm wishes,
Kelly

Yours sincerely,

Kelly M Anderson, PhD
Editor, The EMBO Journal
k.anderson@embojournal.org
